# Online Experimental Design With Estimation-Regret Trade-off Under Network Interference

**Zhiheng Zhang**[*][†]
School of Statistics and Data Science, Shanghai University of Finance and Economics,
Shanghai 200433, P.R. China
Institute of Data Science and Statistics, Shanghai University of Finance and Economics,
Shanghai 200433, P.R. China

**Zichen Wang**[*]
Department of ECE and CSL
UIUC

## Abstract

Network interference has attracted significant attention in the field of causal inference, encapsulating various sociological behaviors in which the treatment assigned to one individual within a network may affect the outcomes of others, such as their neighbors. A key challenge in this setting is that standard causal inference methods often assume independent treatment effects among individuals, which may not hold in networked environments. To estimate interference-aware causal effects, a traditional approach is to inherit the independent settings, where practitioners randomly assign experimental participants to different groups and compare their outcomes. Although effective in offline settings, this strategy becomes problematic in sequential experiments, where suboptimal decisions persist, leading to substantial regret. To address this issue, we introduce a unified interference-aware framework for online experimental design. Compared to existing studies, we extend the definition of arm space using the statistical concept of exposure mapping, which allows for a more flexible and context-aware representation of treatment effects in network settings. Crucially, we establish a Pareto-optimal trade-off between estimation accuracy and regret under the network concerning both time period and arm space, which remains superior to baseline models even without network interference. Furthermore, we propose an algorithmic implementation and discuss its generalization in different learning settings and network topology.

## 1 Introduction

Network interference has attracted significant attention in the fields of causal inference [Leung, 2022a,b, 2023, Ma and Tresp, 2021] and online statistical learning theory [Agarwal et al., 2024, Jia et al., 2024], due to its capability to capture more complex real-world interactions. Unlike the Stable Unit Treatment Value Assumption (SUTVA) assumption [Imbens, 2024], which posits that the treatment assignment and outcomes are isolated to individuals, network interference acknowledges the influences that treatments received by one individual may have on the outcomes of others within a network. This model has found extensive application in economics [Arpino and Mattei, 2016, Munro et al., 2021] and social sciences [Bandiera et al., 2009, Bond et al., 2012, Paluck et al., 2016, Imbens, 2024], where understanding such interconnected dynamics is crucial.

---

[*]Equal contribution.
[†]Corresponding author. Email: `zhangzhiheng@mail.shufe.edu.cn`.

39th Conference on Neural Information Processing Systems (NeurIPS 2025).

To successfully identify causal effect under network interference, one straightforward way is to conduct randomized experiments and use the difference in means type estimators to estimate causal effect based on the experimental data [Leung, 2022a,b, 2023, Gao and Ding, 2023]. Such design is related to many applications [Ciotti et al., 2020, Cai et al., 2015]. For instance, Ciotti et al. [2020] suggested the randomized experiment on a group of volunteering patients to investigate the therapeutic average treatment effects of various drugs for influenza, e.g., COVID-19, where each individual's status of cure is influenced by the treatment assignment of their neighboring individuals. In practice, an experiment may consist of multiple rounds, and researchers may wish to use the experimental data from the previous rounds to enhance the social welfare of the experimental participants by minimizing the regret of the future rounds [Mok et al., 2021]. This requires us to consider the trade-off between the *estimation accuracy* of the causal effect and the *cumulative regret* of the experiment. Apparently, such an online experiment represents a more complex design than offline. For example, if experimental designers directly borrow the Bernoulli sampling in offline design [Leung, 2022a], they would empirically result in a regret linear to round time due to the lack of optimal strategy exploration. This motivates us to design a sequential policy that theoretically guarantees the optimal trade-off between the two objectives under interference. Besides, such sequential policy is also relevant to multi-armed bandits with network interference literature [Jia et al., 2024, Agarwal et al., 2024], which focuses primarily on minimizing regret rather than improving estimation accuracy.

To reiterate, it is crucial to recognize that estimation efficiency and regret might not be optimized simultaneously, necessitating a careful consideration of the trade-off between these two objectives. Optimal estimation efficiency, such as the Bernoulli design above, generally requires that the sampling probability of each arm remains strictly greater than zero, where the sub-optimal decision persists, leading to substantial regret. Conversely, optimal algorithms, such as the Upper Confidence Bound (UCB) [Auer et al., 2002] and its variants, employ probability-vanishing exploration strategies for sub-optimal arms, potentially violating the overlap assumption in causal inference [D'Amour et al., 2021]. This violation limits the estimator's precision, as the overlap assumption is critical for ensuring valid causal inferences by maintaining sufficient data across all arms Sekhon [2009].

Existing works that explore the estimation-regret trade-off often overlook the presence of network interference, effectively assuming a scenario where only a single individual is considered throughout the experiment. Perspectives include empirical algorithm design [Liang and Bojinov, 2023], theoretical bi-objective optimization [Simchi-Levi and Wang, 2024], and analyses of the interaction between trade-offs and exogenous model assumptions [Duan et al., 2024]. In comparison, our work extends such a trade-off in the context of network interference. Integrating the aforementioned perspectives requires an elevated viewpoint to construct a challenging yet more universally applicable framework. Specifically, we introduce a unified online network interference-based experimental design setting, referred to as `MAB-N`. This setting extends the definition of arm space in the multi-armed bandit (MAB) literature by employing the statistical concept of exposure mapping [Leung, 2022a, Aronow and Samii, 2017]. We derive the theoretical optimal estimation-regret trade-off within it and provide an algorithmic implementation capable of achieving this optimal balance. Our contributions are summarized as follows: 1) We establish a unified setting for online experimental design with network interference, referred to as `MAB-N`, which leverages the statistical concept of exposure mapping. 2) We bridge the multi-objective minimax trade-off, achieving Pareto-optimality between treatment effect estimation and regret efficiency under network interference. Additionally, we propose criteria for a MAB algorithm to achieve Pareto-optimality. 3) We propose the `UCB-TSN` algorithm to achieve the aforementioned Pareto trade-off by constructing an upper bound for both the Average treatment effect (ATE) estimation error and regret, which is also validated by experiments. Our `UCB-TSN` algorithm outperforms the elegant preliminary work in (i) the degenerated single-unit case without interference and (ii) the extended adversarial bandit setting. The simulation results are provided in the Appendix E to validate its effectiveness.

Our paper is organized as follows: Section 2 provides a brief literature review. Section 3 introduces our general `MAB-N` setting and discusses Pareto-optimality to illustrate the estimation-regret trade-off. Section 4 provides a general lower bound for the joint performance of regret and estimation, followed by the criteria for any algorithm to achieve Pareto optimality. Section 5 proposes the Pareto-optimal algorithmic implementation and includes a comparison with the baseline. Section 6 extends `MAB-N` to adversarial cases. Finally, Section 7 concludes the paper with further discussion.

## 2 Related Work

Our results primarily bridge two lines of research: (i) extending bandit modeling scenarios by integrating interference settings from the statistical community [Agarwal et al., 2024, Jia et al., 2024], and (ii) exploring the trade-off between estimation and regret in online learning without network interference [Simchi-Levi and Wang, 2024, Duan et al., 2024], as detailed in Table 2 in Appendix C. In the first line of research, the insightful work of Agarwal et al. [2024] creatively utilizes Fourier analysis to reformulate interference-aware bandits as sparse linear stochastic bandits. This innovative approach, however, focuses on interference among first-order neighbors and incorporates a sparsity assumption to limit the number of neighbors each node can have. Complementing this, the meticulous study by Jia et al. [2024] advances the understanding of bandits under interference by forgoing such assumptions, though their methodology requires a switchback design. This design insists that all nodes adopt the same arm synchronously, potentially overlooking scenarios where the optimal arm varies across nodes or subgroups. Turning to the second line of research, we commend Simchi-Levi and Wang [2024] for pioneering a rigorous trade-off between regret and estimation error. Additionally, Duan et al. [2024] contribute significantly by proposing enhancements to this Pareto-optimality, suggesting that both regret and estimation error might simultaneously reach their optimal levels under the thoughtful assumption of covariate diversity. We invite readers to explore further details on these related works in Appendix C.

## 3 Framework

**Classic MAB under network interference.** We introduce our setting following Agarwal et al. [2024], which generalizes Auer et al. [2002], Simchi-Levi and Wang [2024] to the network interference. We focus on the stochastic bandit problem involving a $K$-armed set $\mathcal{K} = \{k\}_{k=0}^{K-1}$, an $N$-unit set $\mathcal{U} = \{i\}_{i=1}^{N}$, and the time horizon $t \in [T]$. The relationship between units is encoded in the adjacency matrix $\mathbb{H} := \{h_{ij}\}_{i,j \in \mathcal{U}} \in \{0,1\}^{N \times N}$[3], where $h_{i,j} = 1$ signifies that units $i$ and $j$ are neighbors, whereas $h_{i,j} = 0$ otherwise. $K, N, \mathbb{H}$ are predefined. At each round, unit interactions induce interference effects. The *original super arm* is represented by an $N$-dimension vector $A_t := (a_{1,t}, ..., a_{N,t}) \in \mathcal{K}^{\mathcal{U}}$. To bridge this formulation to causal inference, we start by notating the so-called potential outcome in statistics [Rubin, 2005] (expected reward in the bandit community [Auer et al., 2002]) as $\{Y_i(A_t)\}_{i \in \mathcal{U}} = \{Y_i(a_{1,t}, a_{2,t}, ...a_{N,t})\}_{i \in \mathcal{U}}$ for unit $i$ in time $t$[4]. Without loss of generality, we set $\forall i \in \mathcal{U}, A \in \mathcal{K}^{\mathcal{U}}, Y_i(A) \in [0, 1]$. In this sense, the *single-unit reward* of unit $i$ upon time $t$ is given by $r_{i,t}(A_t) = Y_i(A_t) + \eta_{i,t}$, where $r_{i,t}(.)$ represents the reward function of unit $i \in \mathcal{U}$, and $\eta_{i,t}$ is zero-mean i.i.d. 1-sub Gaussian noise for each unit. Finally, we define instance $\nu$ as any legitimate choice of $\{\mathcal{D}(Y_i(A))\}_{i \in \mathcal{U}, A \in \mathcal{K}^{\mathcal{U}}}$, where $\mathcal{D}(Y_i(A))$ denotes the reward distribution of unit $i$ if super arm $A$ is pulled; and then denote $\mathcal{E}_0$ as the set of all feasible $\nu$. Our primary interest is designing a learning policy $\pi := (\pi_1, ..., \pi_T)$. In round $t$, the agent observes the history $\mathcal{H}_{t-1} = \{A_1, \{r_{i,1}(A_1)\}_{i \in \mathcal{U}}, ..., A_{t-1}, \{r_{i,t-1}(A_{t-1})\}_{i \in \mathcal{U}}\}$, where each term is an $N$-dimensional vector. The policy $\pi_t$ is a probabilistic map from $\mathcal{H}_{t-1}$ to the next action $A_t$. We denote $\pi_t(A) = \mathbb{P}_\pi(A_t = A \mid \mathcal{H}_{t-1})$ indicating the probability that a super arm $A$ is selected in round $t$.

**Additional notation.** We define $e_i$ as the standard basis vector whose $i$-th element is 1 and all other elements are 0. For any $Q \in \mathbb{N}^+$, we use the shorthand notation $[Q] := \{1, 2, \ldots, Q\}$. We define the operations: $a \vee b := \max\{a, b\}, \quad a \wedge b := \min\{a, b\}$. For sequences of positive numbers $\{a_n\}_{n \in \mathbb{N}^+}$ and $\{b_n\}_{n \in \mathbb{N}^+}$, we adopt the following asymptotic notations: $a_n = O(b_n)$ if there exists a constant $C > 0$ such that for all sufficiently large $n$, $a_n \leq Cb_n$.; $a_n = \Omega(b_n)$ if there exists a constant $C > 0$ such that for all sufficiently large $n$, $a_n \geq Cb_n$.; $a_n = \Theta(b_n)$ if both $a_n = O(b_n)$ and $a_n = \Omega(b_n)$ hold. Finally, $a_n = \tilde{O}(b_n)$ if there exist constants $C > 0$ and $k \in \mathbb{N}^+ \cup \{0\}$ such that $a_n \leq Cb_n(\log b_n)^k$.

---

[3]It does not mean we must get all information about $\mathbb{H}$; instead, it depends on our detailed design.

[4]Unit $i$'s potential outcome is only related to the treatments of the total population via a fixed function, as is standard in interference-based causality [Leung, 2022a,b, 2023]. This setting relaxes the traditional "Stable Unit Treatment Value Assumption" (SUTVA) [Rubin, 1980], which assumes that one unit's outcome is unaffected by others' treatments.

## 3.1 Motivation: the hardness of classic MAB under interference

In this framework, referring to the concept of cumulative regret in traditional MAB problems [Lattimore and Szepesvári, 2020b], the performance metric of policy $\pi$ could be identified as

$$\mathcal{R}^{naive}(T, \pi) := \frac{T}{N} \sum_{i \in \mathcal{U}} Y_i(A^*) - \mathbb{E}_\pi \left[ \frac{1}{N} \sum_{t \in [T]} \sum_{i \in \mathcal{U}} r_{i,t}(A_t) \right], \; A^* := \arg \max_{A \in \mathcal{K}^{\mathcal{U}}} \frac{1}{N} \sum_{i \in \mathcal{U}} Y_i(A).$$
(1)

Foreseeably, a fundamental challenge in this setting is that the original super arm suffers from an exponentially large action space ($|\mathcal{K}^{\mathcal{U}}| = K^N$), making direct optimization infeasible. Given this computational burden, we first establish a *negative result* to illustrate that directly pursuing the policy $\pi$ using the original super arm is computational *impractical*.

**Proposition 1** *Given a priori $N, K, \mathbb{H}$. For any policy $\pi$, there exists a hard instance $\nu \in \mathcal{E}_0$ such that $\mathcal{R}_\nu^{naive}(T, \pi) = \Omega\big(\frac{1}{\sqrt{N}}(T \wedge \sqrt{K^N T})\big)$.*

Proposition 1 reveals that the regret convergence rate is influenced by the relative size of the time period compared to the arm space, resulting in a two-piece function. Specifically, when $T \leq K^N$ under interference, the regret $\mathcal{R}_\nu^{naive}(T, \pi)$ increases linearly with $T$. Conversely, otherwise, although the rate degenerates to a square root relative to $T$, it is adversely affected by an exponentially large parameter ($\sqrt{K^N/N}$). This negative result, from a counter perspective, substantiates why Agarwal et al. [2024] and Jia et al. [2024] respectively relaxed the model from the network topology and action space: Agarwal et al. [2024] prudently considers interference only from first-order neighbors and incorporates sparsity assumptions, while Jia et al. [2024] restrict the action space to the all one and all zero $N$-dimensional vector. Without such considerations, obtaining meaningful regret bounds would be unfeasible.

Further, it manifests more insights upon the triple of concepts (i) time, (ii) regret, and (iii) arm space, than lower bound analysis in classic MAB [Lattimore and Szepesvári, 2020b]. It is because researchers tend to preemptively judge that "time period $\gg$ arm numbers", e.g., force $N = 1$ in the single-unit setting and then $T \gg K$ holds by default. However, this oversimplification consideration of arm space can be detrimental under the interference scenario. For instance, even if we just choose $K = 2, N = 30$, any algorithm under interference-based MAB setting would potentially be cursed by an impractical regret. In sum, these insights motivate us to develop a general statistical framework to allow for a more reasonable reduction in the action space dimension without imposing excessive assumptions on the network topology, which is the so-called `MAB-N`, illustrated as follows.

## 3.2 Setting: `MAB-N`

We introduce the concept of *exposure mapping* developed by Leung [2022a], Aronow and Samii [2017]. We define the pre-specified function mapping from the original super arm space ($\mathcal{K}^{\mathcal{N}}$) to a $d_s$-cardinality discrete values ($d_s \ll K^N$) taking advantage of the network structure. For clarity, we consider the discrete function case:

$$s_i := \mathbf{S}(i, A, \mathbb{H}), \text{ where } \boldsymbol{S} : \mathcal{U} \times \mathcal{K}^{\mathcal{U}} \times \{0, 1\}^{N \times N} \to \mathcal{U}_s, \; |\mathcal{U}_s| = d_s.$$
(2)

Here $\mathcal{U}_s$ is called as exposure arm set. We set $S = \{\mathbf{S}(i, A, \mathbb{H})\}_{i \in \mathcal{U}} \equiv (s_1, \dots, s_N)$ as the *exposure super arm*, and then we can decompose the policy $\pi_t(\cdot)$ and define the exposure-based reward:

$$\pi_t(A) := \mathbb{P}(A_t = A \mid \mathcal{H}_{t-1}) = \mathbb{P}(A_t = A \mid S_t)\mathbb{P}(S_t \mid \mathcal{H}_{t-1}),$$
$$[\tilde{Y}_i(S_t), \tilde{r}_{i,t}(S_t)]^\top := \sum_{A \in \mathcal{K}^{\mathcal{U}}} [Y_i(A), r_{i,t}(A)]^\top \mathbb{P}(A_t = A \mid S_t),$$
(3)

The second line of Eq (3) generalizes the framework of Leung [2022a] by incorporating a broader class of exposure mappings. Specifically, while the original formulation assumes a fixed exposure structure, our approach allows for a more flexible characterization of treatment assignments under network interference. Detailed derivations are deferred to Appendix F. To formalize in practice, we could define $\mathbb{P}(A_t = A \mid S)$ as a *predefined, time-invariant* sampling rule, which the learner specifies before the learning process begins. For example, in the case of uniform sampling (by default), we have: $\mathbb{P}_\pi(A_t = A \mid S) = \sum_{A \in \mathcal{K}^{\mathcal{U}}} \delta\{\mathbb{A}\}/|\mathbb{A}|$, where $\delta(\cdot)$ is an indicator function, and

$\mathbb{A} := \{A : \{\mathbf{S}(i, A, \mathbb{H})\}_{i \in \mathcal{U}} = S\}$ denotes the set of all assignments that result in the observed exposure state $S_t$. This formulation ensures that if $S$ does not match the set $\{\mathbf{S}(i, A, \mathbb{H})\}_{i \in \mathcal{U}}$, the probability of selecting $A_t = A$ given $S$ is zero. Conversely, if $S$ corresponds to this set, then $A$ is chosen with strictly positive probability, i.e., $\mathbb{P}(A_t = A \mid S) > 0$. Under this framework, the observed outcome $\tilde{Y}_i(S_t)$ in Eq (3) depends solely on the network topology $\mathbb{H}$ and the exposure state $S_t$, independent of the specific arm assignment $A_t$. This highlights a key property of exposure mapping: it abstracts away individual-level treatment assignments while preserving the structural dependencies induced by network interference. To further quantify decision-making performance under network interference, we introduce the exposure reward $\tilde{r}_{i,t}(S_t)$, which serves as a proxy for the expected reward in the exposure space[5]. Building on this exposure-based representation, we now define the regret function, which quantifies the performance gap between the optimal and chosen policies under exposure mapping.

**Regret based on exposure mapping.** According to the action space reduction in Eq (3), we provide a more general and realistic regret compared to Jia et al. [2024], Simchi-Levi and Wang [2024], Agarwal et al. [2024] (refer to Example 1-4). We define the clustering set $\mathcal{C} := \{\mathcal{C}_q\}_{q \in [C]}, C = |\mathcal{C}|$ where $\forall i \neq j, i, j \in [C], \mathcal{C}_i \cap \mathcal{C}_j = \varnothing, \cup \{\mathcal{C}_q\}_{q \in [C]} = \mathcal{U}$. For brevity, we denote $\mathcal{C}^{-1}(i)$ as the cluster of node $i$. We define the exposure-based regret:

$$\mathcal{R}_\nu(T, \pi) = \frac{T}{N} \sum_{i \in \mathcal{U}} \tilde{Y}_i(S^*) - \frac{1}{N} \mathbb{E}_\pi \left[ \sum_{t \in [T]} \sum_{i \in \mathcal{U}} \tilde{r}_{i,t}(S_t) \right], \quad S^* = \arg\max_{S \in \mathcal{U}_\mathcal{E}} \sum_{i \in \mathcal{U}} \tilde{Y}_i(S), \quad (4)$$

where exposure arm space $\mathcal{U}_\mathcal{E} := \mathcal{U}_\mathcal{C} \cap \mathcal{U}_\mathcal{O}$ with $\mathcal{U}_\mathcal{C} := \{S : \forall i, j \in \mathcal{U}, \mathcal{C}^{-1}(i) = \mathcal{C}^{-1}(j) \text{ implies } Se_i = Se_j\}$ and $\mathcal{U}_\mathcal{O} := \{\{\mathbf{S}(i, A, \mathbb{H})\}_{i \in \mathcal{U}} : A \in \mathcal{K}^\mathcal{U}\}$. Here, $\mathcal{U}_\mathcal{C}$ denotes all kinds of ideally cluster-wise switchback exposure super arm. For instance, if $\mathcal{U}_s \in \{0, 1\}, N = 4, \mathcal{C}_1 = \{1, 2\}, \mathcal{C}_2 = \{3, 4\}$, then $\mathcal{U}_\mathcal{C} = \{(k_1, k_1, k_2, k_2) : k_1, k_2 \in \{0, 1\}\}$. Moreover, $\mathcal{U}_\mathcal{O}$ includes all exposure arm sets compatible with the original arm set. It induces that $|\mathcal{U}_\mathcal{E}| \leq |d_s|^\mathcal{C}$. Essentially, during the exposure mapping process, we efficiently reduce the action space by condensing the original arm information in a structured manner, thereby achieving a controlled enhancement of regret efficiency. According to Proposition 1, this balance between sacrifice and gain emerges naturally and inevitably. Such cluster-wise exposure mapping structures have appeared in multiple prior works. We illustrate how our framework can surrogate previous settings as special cases. By assigning specific parameter values, we can (i) flexibly transition between these cases (the following examples), (ii) allow for an adaptive balance in different scenarios (Table 1 in Appendix C), and (iii) even characterize new and more general real-world scenarios (experiments in Appendix E) where existing methods would fundamentally fail.

**Comparison with previous literature.** For the comparison of regret, **Example (i)** Classic MAB [Auer et al., 2002, Simchi-Levi and Wang, 2024] considered the case $N = 1$, i.e., single unit without network, and $\mathbf{S}(1, A, \mathbb{H}) := A, A \in \mathcal{K}$. **Example (ii)** Agarwal et al. [2024] chooses $\mathbf{S}(i, A, \mathbb{H}) := Ae_i$ and $C = N$ (each unit is assigned to a separate cluster). **Example (iii)** On the other hand, Jia et al. [2024] chooses $\mathbf{S}(i, A, \mathbb{H}) := Ae_i$ and $C = 1$ (all units are in one cluster), which denotes the global proportion of treatment in each time $t$. Additionally, the exposure mapping and clustering technique could also be traced back to the offline setting. **Example (iv)** Suppose $\forall j \in \mathcal{U}, \sum_j h_{ij} > 0$. We can choose $\mathbf{S}(i, A, \mathbb{H}) := \mathbf{1}\{\sum_{j \in \mathcal{U}} h_{ij} a_j / \sum_{j \in \mathcal{U}} h_{ij} \in [0, \frac{1}{2})\}$ inherited from the literature of offline causality [Leung, 2022a, Gao and Ding, 2023]. They require approximate neighborhood interference and their objective is to explore the influence of the treatment assignment proportion among all neighborhoods of each unit, which is still under-explored in the online learning scenario (we refer readers to experiments in Appendix E). **Example (v)** For a supplement, we point out that the clustering strategy could also be traced back to the offline setting, which is also our special case: Viviano et al. [2023], Zhang and Imai [2023] considered the clustering-based setting $\mathbf{S}(i, A, \mathbb{H}) := Ae_i$, in which only considers the set of the exposure arm $\{0, 1\}^\mathcal{C}$. Specifically, Viviano et al. [2023] focuses on the Bernoulli design in clusters, while Zhang and Imai [2023] further assumes that interference occurs only within clusters rather than across clusters.

---

[5]Notably, the difference between $\tilde{Y}_i(S_t)$ and the empirically observed reward $r_{i,t}(A_t)$ arises from two distinct noise components: (i) sampling noise, where practitioners approximate $\tilde{r}_{i,t}(S_t)$ using samples of $r_{i,t}(A_t)$, and (ii) endogenous noise, inherited from the original variability $\eta_{i,t}$ in the observed reward. A detailed discussion on noise rescaling is provided in Appendix F.

In these examples, they all satisfy $\mathcal{U}_\mathcal{E} = \mathcal{U}_\mathcal{C} \cap \mathcal{U}_\mathcal{O} \neq \emptyset$. We provide more justification for it in the next section and Appendix N.

## 3.3 Goal: estimation-regret trade-off

We introduce the trade-off between regret efficiency and statistical power of reward gap estimation. ATE between exposure super arm $S_i$ and $S_j$ is defined as the reward gap [Simchi-Levi and Wang, 2024]: $\Delta^{(i,j)} := \frac{1}{N} \sum_{i' \in \mathcal{U}} \left( \tilde{Y}_{i'}(S_i) - \tilde{Y}_{i'}(S_j) \right)$, where $S_i, S_j \in \mathcal{U}_\mathcal{E}$. It is a generalized definition compared with the most relevant literature [Jia et al., 2024, Agarwal et al., 2024, Simchi-Levi and Wang, 2024] when considering ATE (specifying the exposure mapping function as in Table 1 of Appendix C). We use $\hat{\Delta}^{(i,j)} := \{\hat{\Delta}_t^{(i,j)}\}_{t \geq 1}, \hat{\Delta} := \{\hat{\Delta}^{(i,j)}\}_{S_i, S_j \in \mathcal{U}_\mathcal{E}}$ to identify a sequence of adaptive admissible estimates of $\Delta^{(i,j)}$. The total design of an MAB experiment could be represented by the vector $\{\pi, \hat{\Delta}\}$. Our final goal is to portray the mini-max trade-off:

$$\min_{\{\pi, \hat{\Delta}\}} \max_{\nu \in \mathcal{E}_0} \left( \mathcal{R}_\nu(T, \pi), e_\nu(T, \hat{\Delta}) \right), \text{ where } e_\nu(T, \hat{\Delta}) := \max_{S_i, S_j \in \mathcal{U}_\mathcal{E}} \mathbb{E}\left[ \left| \Delta^{(i,j)} - \hat{\Delta}_T^{(i,j)} \right| \right]. \quad (5)$$

Given any feasible $\nu$, $\mathcal{R}_\nu(T, \pi)$ is associated with $\pi$, while $e_\nu(T, \hat{\Delta})$ is associated with $\hat{\Delta}$. Due to the complicated relation between $\pi$ and $\hat{\Delta}$ w.r.t. the history $\mathcal{H}_t, t \in [T]$, especially in the network interference setting, this multi-objective optimization is quite challenging. For preparation, we define what is the "best" pair of $\{\pi, \hat{\Delta}\}$ via the following definition of *front*:

**Definition 1 (Front and Pareto-dominate)** *For a given pair of $\{\pi, \hat{\Delta}\}$, we call a set of pairs $(\mathcal{R}, e)$ as a front of $\{\pi, \hat{\Delta}\}$, denoted by $\mathcal{F}(\pi, \hat{\Delta})$, if and only if (i) [Feasible instances exists] $\mathcal{V}_0 := \left\{ \nu_0 \in \mathcal{E}_0 : \left( \sqrt{\mathcal{R}_{\nu_0}(T, \pi)}, e_{\nu_0}(T, \hat{\Delta}) \right) = (\mathcal{R}, e) \right\} \neq \emptyset$, and (ii)[instances in $\mathcal{V}_0$ are the best] $\nexists \nu \in \mathcal{E}/\mathcal{V}_0, s.t. \exists \otimes \in \{K, T\}, (R, e) \preccurlyeq_\otimes \left( \sqrt{\mathcal{R}_\nu(T, \pi)}, e_\nu(T, \hat{\Delta}) \right)$. We claim $\{\pi, \hat{\Delta}\}$ Pareto-dominate another solution $\{\pi', \hat{\Delta}'\}$ if $\forall (\mathcal{R}, e) \in \mathcal{F}(\pi, \hat{\Delta}), \exists (\mathcal{R}', e') \in \mathcal{F}(\pi', \hat{\Delta}')$, such that $\forall \otimes \in \{K, T\}$, either (i) $\mathcal{R} \preccurlyeq_\otimes \mathcal{R}', e \prec_\otimes e'$ or (ii) $\mathcal{R} \prec_\otimes \mathcal{R}', e \preccurlyeq_\otimes e'$[6].*

We formalize the definition of *front* in the symbol of order $\preccurlyeq_\otimes, \prec_\otimes$. e.g., $(a, b) \preccurlyeq_\otimes (c, d), e \prec_\otimes f, g \preccurlyeq_\otimes h$ denotes $(a \leq c, b \leq d), e < f, g \leq h$ when we only consider the parameter with respect to $\otimes \in \{K, T\}$ sufficiently large and omit any other parameter. Finally, Pareto-optimality is identified according to the Pareto-dominance in Definition 1 as follows.

**Definition 2 (Pareto-optimal and Pareto Frontier)** *A feasible pair $(\pi^*, \hat{\Delta}^*)$ is claimed to be Pareto-optimal when it is not Pareto-dominated by any other feasible solution. Pareto Frontier $\mathcal{P}$ is denoted as the envelope of fronts of all Pareto-optimal solutions.*

For example, according to Definition 2, $\{\pi_i, \hat{\Delta}_i\}_{i \in [3]}$ is not dominated by each other in Figure 1. For more intuitive comprehension for practitioners, we provide the closed-form mathematical formulation in the following section.

## 4 Pareto-optimality

In the above section, we introduce the motivation and establishment of our `MAB-N` and then construct the mini-max trade-off problem along with the Pareto-optimality property. In this section, we explore in detail the lower bound of such trade-off and the geometric structure of Pareto optimality. According to the Definition 1-2, in the following text, our analysis upon optimality mainly focuses on the individual arm space $K$ and the time horizon $T$. Here $K$ is included in the exposure arm space $\mathcal{U}_\mathcal{E}$. Other parameters, such as $N$, are seen as a pre-fixed constant. We first introduce the following condition to restrict the fairly broad relationship between parameters.

**Condition 1** *Exposure mapping* **S** *and clusters* $\mathcal{C}$ *should satisfy* $2 \leq |\mathcal{U}_\mathcal{E}| \leq T$.

---

[6]Intuitively speaking, if we denote the region formed by $\mathcal{F}(\pi, \hat{\Delta}), \mathcal{F}(\pi', \hat{\Delta}')$, X-axis and Y-axis in the first quadrant as `Region`$(\pi, \hat{\Delta})$, `Region`$(\pi', \hat{\Delta}')$, respectively. Then $\{\pi, \hat{\Delta}\}$ Pareto-dominate $\{\pi', \hat{\Delta}'\}$ means `Region`$(\pi, \hat{\Delta}) \subseteq$ `Region`$(\pi', \hat{\Delta}')$.

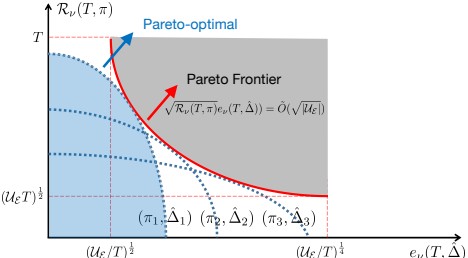
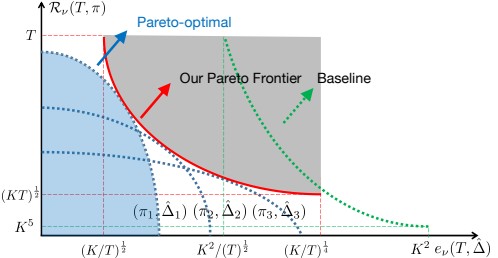

(a) Our general result under interference.

(b) The comparison with the baseline without interference.

Figure 1: Pareto-optimality. (a) We use three blue fronts (first quadrant) to show three different MAB algorithms $\{\pi_i, \hat{\Delta}_i\}_{i \in [3]}$, e.g., the blue regions represent the regrets and estimation errors that can be realistically achieved in all kinds of instances given $\{\pi_1, \hat{\Delta}_1\}$. MAB algorithm is Pareto-optimal if and only if its blue front is tangent to the Pareto Frontier (red) (otherwise, it is intersecting with the grey region). (b) The green line represents the baseline in Simchi-Levi and Wang [2024], which loses the Pareo-optimality concerning arm space.

Condition 1 restricts to the case where $T$ is relatively large with pre-specified non-empty $\mathcal{U}_\mathcal{E}$, which is inherently verifiable, adjustable and relevant. Regardless of any pre-fixed $\mathbb{H}$, we could manually design legitimate (2) and clusters to fit Condition 1. It is the weakest condition to date, without additional restriction upon network topology, compared to the previous literature mentioned in the above section. Additional justification on exposure mapping and feasibility of model conditions are in Appendix D and Appendix N. Under such conditions, we establish a general lower bound when simultaneously considering the regret and estimation error.

**Theorem 1** *Given any* **S** *and* $\mathcal{C}$ *that satisfies Condition 1. Given any online decision-making policy* $\pi$*, the trade-off between the regret and the estimation exhibits*

$$\inf_{\hat{\Delta}_T} \max_{\nu \in \mathcal{E}_0} \left( \sqrt{\mathcal{R}_\nu(T, \pi)} e_\nu(T, \hat{\Delta}) \right) = \Omega_{K,T} \left( \sqrt{|\mathcal{U}_\mathcal{E}|} \right). \tag{6}$$

We use the subscript $\{K, T\}$ to emphasize that the order just corresponds to these two parameters and omit the subscript in the following text.

**The challenge of the proof.** The core idea involves constructing two carefully designed multi-armed bandit instances, $\nu_1$ and $\nu_2$, such that any estimator $\hat{\Delta}_T$ faces challenges in simultaneously achieving low regret and high estimation accuracy across both instances. This difficulty is divided into three parts: (i) Regarding the goal, unlike the regret lower bound analysis in classic multi-armed bandit problems [Lattimore and Szepesvári, 2020a], we employ statistical hypothesis testing to bridge these two goals, rather than analyzing worst-case regret in isolation. (ii) Concerning instance construction, compared to Simchi-Levi and Wang [2024], constructing two distinct instances is challenging due to the interference affecting the entire system, making it difficult for an algorithm's regret or estimation behavior to differ significantly. (iii) From an information-theoretic perspective, the correlated structure complicates the issue. The networked nature of exposure rewards necessitates a refined divergence measure that accounts for shifts in probability mass across dependent actions, such as when applying the Kullback-Leibler inequality.

**The sketch of the proof.** We defer the detailed proof in Appendix H. To tackle these challenges, we carefully construct a pair of instances via slighting perturbing the reward of $Y(A_t)$ compatible with specific exposure arms: we let $Y_i(A) := f_i(A) \in (\varepsilon_0, 1 - \varepsilon_0), \varepsilon_0 \in (0, 1), r_{i,t}(A) \in \{-1, 1\}$. It means $r_{i,t}(A) = \text{Rad}(\frac{1-f_i(A)}{2}, \frac{1+f_i(A)}{2})$. Moreover, We establish :

$$r'_{i,t}(A) := \begin{cases} r_{i,t}(A) & \forall A \text{ satisfying } \mathbb{P}(A_t = A \mid S) = 0. \\ \text{Rad}(\frac{1-f_i(A)+\alpha}{2}, \frac{1+f_i(A)-\alpha}{2}) & \forall A \text{ satisfying } \mathbb{P}(A_t = A \mid S) > 0. \end{cases} \tag{7}$$

with $\alpha > 0$ sufficiently small, and $S$ is specifically selected. Conducting the information-theoretic argument, we prove

$$\inf_{\hat{\Delta}_T} \max_{\nu \in \mathcal{E}_0} \mathbb{P}_\nu \left( \max_{i,j \in \mathcal{U}_\mathcal{E}} |\hat{\Delta}_T^{(i,j)} - \Delta_\nu^{(i,j)}| \geq \frac{\alpha}{2} \right) \geq \frac{1}{2} \left[ 1 - \sqrt{\frac{1}{2} q' N \alpha^2 \frac{\mathcal{R}_{\nu_1}(T,\pi)}{|\mathcal{U}_\mathcal{E}|}} \right].$$

Here $q'$ is a constant. Such inequality bridges the relationship between the statistical power and regret efficiency under these two instances and thus induces the final lower bound in Theorem 1.

Theorem 1 states that for any given policy $\pi$, there always exists at least one hard MAB instance $\nu$, in which no matter what legitimate $\mathbf{S}$, $\mathcal{C}$, and estimator $\hat{\Delta}_T$ we choose, the lower bound $\Omega(\sqrt{|\mathcal{U}_\mathcal{E}|})$ always holds. In other words, there are always challenging instance $\nu$ such that $e_\nu(T, \hat{\Delta}) = \Omega_{K,T}(\sqrt{|\mathcal{U}_\mathcal{E}|}/\sqrt{\mathcal{R}_\nu(T,\pi)})$. We take examples considering *the worst case of $\nu$*: according to the fact $\mathcal{R}_\nu(T,\pi) = O(T)$, Theorem 1 states that the worst estimation error is at least $\Omega((|\mathcal{U}_\mathcal{E}|/T)^{\frac{1}{2}})$ and could not be further decreased; stepping forwards, as we will show in Section 5 that our proposed MAB-N algorithm's regret is upper bounded by $O(\sqrt{|\mathcal{U}_\mathcal{E}|T})$, then Theorem 1 additionally states that the worst estimation error of our algorithm will be ideally at least $(|\mathcal{U}_\mathcal{E}|/T)^{\frac{1}{4}}$ without need of further implementation. In sum, Theorem 1 serves as a *free lunch*, enabling practitioners to perform interactive inference and prediction regarding the trade-off between the algorithm's regret efficiency and statistical power. A natural question is what is the relationship between the lower bound and the Pareto-optimality? We provide the following closed-form for Pareto Frontier following the lower bound in Theorem 1.

**Theorem 2** *Following the condition in Theorem 1, a feasible pair $\{\pi, \hat{\Delta}\}$ is Pareto-optimal if the pair satisfies* $\max_{\nu \in \mathcal{E}_0} \left( \sqrt{\mathcal{R}_\nu(T,\pi)} e_\nu(T, \hat{\Delta}) \right) = \tilde{O}\left( \sqrt{|\mathcal{U}_\mathcal{E}|} \right)$. *The Pareto Frontier is represented as* $\mathcal{P} = \left\{ (\mathcal{R}_\nu(T,\pi), e_\nu(T, \hat{\Delta})) : \sqrt{\mathcal{R}_\nu(T,\pi)} e_\nu(T, \hat{\Delta}) = \tilde{O}(\sqrt{|\mathcal{U}_\mathcal{E}|}) \right\}$.

Theorem 2 establishes the sufficiency condition for the Pareto-optimal property. We also analyze the necessity conditions in Appendix I. For a visual representation, readers are referred to Figure 1, which illustrates the Pareto-optimal pairs $\pi, \hat{\Delta}$ (blue region) and the Pareto Frontier (red line). Theorems 1 and 2 are applicable to any complex network topology $\mathbb{H}$ under mild conditions on exposure mapping (Condition 1). These results not only generalize non-trivial trade-offs under network interference but also enhance the degenerated results without interference. Specifically, when compared to the setting of Simchi-Levi and Wang [2024], (i) we advance the Pareto-optimality trade-off concerning arm space, and (ii) we eliminate their additional assumption on ATE, specifically that $\hat{\Delta}^{i,j} = \Theta(1)$. Furthermore, our reward $r_t$ is not constrained to the interval $[-1, 1]$, allowing for unbounded values.

## 5   Algorithm

In Section 5, we introduce the **U**pper **C**onfidence **B**ound algorithm with **T**wo **S**tages under **N**etwork interference (UCB-TSN). Our UCB-TSN operates in two phases: (i) uniformly explore the exposure super arm space using a round-robin approach to estimate the ATE within $T_1$ rounds. and (ii) applying the UCB exploration strategy to minimize regret. The pseudo code is provided in the appendix due to the space limitation. Initially, we demonstrate that phase (i) effectively reduces the estimation error, as detailed below.

**Theorem 3 (ATE estimation upper bound)** *Following the condition in Theorem 1. If $T_1 \geq |\mathcal{U}_\mathcal{E}|$, for any $S_i \neq S_j \in \mathcal{U}_\mathcal{E}$, the ATE estimation error of UCB-TSN can be upper bounded as* $\mathbb{E}\left[ |\hat{\Delta}_T^{(i,j)} - \Delta^{(i,j)}| \right] = \tilde{O}\left( \sqrt{|\mathcal{U}_\mathcal{E}|/T_1} \right)$.

Theorem 3 asserts that uniform exploration in phase (i) aids in estimating the ATE. This is intuitive, as UCB-TSN explores the exposure action space using a round-robin approach. Provided that the practitioner selects $T_1 = \Omega(T^\alpha)$ for $\alpha \in (0,1)$, the ATE estimation is consistent. Following the uniform exploration in phase (i), phase (ii) focuses on identifying the optimal arm, leading to the convergence of the overall regret.

**Theorem 4 (Regret upper bound)** *Following the condition in Theorem 1. With $\delta = 1/T^2$ and $T_1 \geq |\mathcal{U}_\mathcal{E}|$, the regret of UCB-TSN can be upper bounded as* $\mathcal{R}(T,\pi) = \tilde{O}\left( \sqrt{|\mathcal{U}_\mathcal{E}|T} + T_1 \right)$.

Theorem 4 claims the regret could converge as $o(T)$, accommodating with well-selected $T_1$, such as $T_1 = \sqrt{|\mathcal{U}_\mathcal{E}|T}$. Theorem 4 is consistent with Proposition 1 when we omit phase (i), i.e., $T_1 = 0$ and reserve phase (ii). By the combination of Theorem 3-4, we claim the Pareto-optimality as stated in Section 4 in our UCB-TSN as follows.

**Corollary 1 (Trade-off result)** *Following the condition in Theorem 1. Set $T_1 \geq \sqrt{|\mathcal{U}_\mathcal{E}|T}$, for all $\nu \in \mathcal{E}_0$, UCB-TSN can guarantee $e_\nu(T, \hat{\Delta})\sqrt{\mathcal{R}_\nu(T, \pi)} = \tilde{O}(\sqrt{|\mathcal{U}_\mathcal{E}|})$.*

Corollary 1 states that under a stricter but still mild condition upon the uniform exploration process $T_1$ (since $\sqrt{|\mathcal{U}_\mathcal{E}|T} \geq |\mathcal{U}_\mathcal{E}|$ under Condition 1), UCB-TSN could achieve the Pareto-optimal property according to Theorem 1.

**Comparison with the baseline algorithm.** To facilitate the fair comparison, we consider the degenerated case as in Simchi-Levi and Wang [2024], where we choose $N = 1, |\mathcal{U}_\mathcal{E}| = K \geq 2$ in our UCB-TSN. Here $\mathcal{K}$ corresponds to $\mathcal{U}_\mathcal{E}$. We compare the regret in (i) and estimation in (ii). (i) For the regret, they proposed their EXP3EG which guarantees the regret upper bound as $\mathcal{R}_\nu(T, \pi) = \tilde{O}(K^5 + T^{1-\alpha})$, where $\alpha \in [0, 1]$[7]. Such result is build upon their assumption $\frac{1}{N}\sum_{i' \in \mathcal{U}}(\tilde{Y}_{i'}(S^*) - \tilde{Y}_{i'}(S_i)) = \Theta(1)$ for all $S_i \neq S^*$. In this single-agent setting with such assumption, it should be pointed out that our regret upper bound in Theorem 4 could be naturally strengthened to $\tilde{O}(K + T_1)$ (refer to our instance dependent regret upper bound in Lemma 1 in the Appendix). Thus our regret upper bound is strictly stronger than theirs if we force $T_1 = O(T^{1-\alpha})$. (ii) For the estimation error, they state that ATE could be upper bounded by $e_\nu(T, \pi) = \tilde{O}(K^2 T^{-\frac{1-\alpha}{2}})$. Therefore our estimation error in Theorem 3, i.e., $\tilde{O}(\sqrt{|\mathcal{U}_\mathcal{E}|/T_1}) = \tilde{O}(\sqrt{K/T_1})$ is strictly stronger than theirs since it is legitimate to force $T_1 = T^{1-\alpha} \vee |\mathcal{U}_\mathcal{E}|$. Such strict improvement (i)-(ii) is illustrated in Figure 1. It validates the statements under Theorem 2 that we achieve the Pareto optimality with respect to time period $T$ and additionally, the exposure super arm space $|\mathcal{U}_\mathcal{E}|$.

## 6 Extension to adversarial setting

**The adversarial setting.** We cover Simchi-Levi and Wang [2024]'s adversarial setting when considering trade-offs. We consider $r_{i,t}(A_t) = Y_i(A_t) + f_t + \eta_{i,t}$, where $\eta_{i,t}$ is i.i.d. zero means noise. In addition to the standard setting in the preliminaries, there is an $f_t$, a pre-specified function w.r.t. period $t$, which is an adversarial noise. We suppose $r_{i,t}(A) \in [0, 1]$ for all $i \in \mathcal{U}, A \in \mathcal{K}^\mathcal{U}$ and $t \in [T]$. It is also easy to verify that $\tilde{r}_{i,t}(S) \in [0, 1]$ for all $t \in [T]$, $S_i \in \mathcal{U}_\mathcal{E}$, $i \in \mathcal{U}$ and $\mathbb{E}[\tilde{r}_{i,t}(S)] = \tilde{Y}_i(S) + f_t$. Motivated by the fact that the UCB algorithm discussed in the previous section cannot be applied directly in this context, we provide the advanced EXP3-TSN algorithm for substitution. The pseudo-code and details of the EXP3-TSN are provided in the Appendix.

**Theorem 5 (Pareto-optimality trade-off in the adversarial setting)** *Following the condition in Theorem 1, let $\mathcal{T}(t) \equiv (2|\mathcal{U}_\mathcal{E}| + 1)^2 \log(t|\mathcal{U}_\mathcal{E}|^2)/2(e-2)|\mathcal{U}_\mathcal{E}|$, then (i)[ATE estimation] Suppose $T \geq \mathcal{T}(T)$ and $T_1 \geq \mathcal{T}(T_1)$. For any $S_i \neq S_j$, the ATE estimation error of the EXP3-TSN can be upper bounded as in Theorem 3. (ii)[Regret] Stepping back, if we only suppose $T \geq \mathcal{T}(T)$, then the regret of EXP3-TSN could be upper bounded as in Theorem 4. (iii)[Pareto-optimality] Stepping forward, additionally set $T_1 \geq \mathcal{T}(T_1) \vee \sqrt{|\mathcal{U}_\mathcal{E}|T}$. then EXP3-TSN can also guarantee the Pareto-optimality trade-off, i.e., $e_\nu(T, \hat{\Delta})\sqrt{\mathcal{R}(T, \pi)} = \tilde{O}(\sqrt{|\mathcal{U}_\mathcal{E}|})$.*

Theorem 5 states that under additional mild conditions, i.e., $T \geq \mathcal{T}(T)$ and $T_1 \geq \mathcal{T}(T_1)$[8], the regret, ATE estimation error and the Pareto-Optimality trade-off could still keep their original form in Theorem 3-4. In such an adversarial setting, our result can also outperforms Simchi-Levi and Wang [2024] with the same argument as in Section 5, and the discussion concerning the order of the node number $N$ aligns analogously.

---

[7]In their paper, $\mathcal{R}_\nu(T, \pi) = O(\sum_{A \in \mathcal{K}/\{A^*\}} K^4 log(T) + T^{1-\alpha} log(T)) = \tilde{O}(K^5 + T^{1-\alpha})$. Here $A^*$ denotes the best arm.

[8]Since $\mathcal{T}(t) = O(|\mathcal{U}_\mathcal{E}| log(|\mathcal{U}_\mathcal{E}|t))$, such conditions are natural to satisfy given that $T$ is sufficiently large.

# 7 Conclusion and future work

We establish a unified online learning framework under network interference via statistical exposure mapping, balancing learning efficiency and statistical power through a Pareto-optimal trade-off between regret and estimation error. We also introduce UCB-TSN, an algorithm achieving this balance with provable guarantees. In the future, we aim to investigate the estimation-regret trade-off in fully adversarial networked bandit, and extend it to more general and complex topics such as multi-agent reinforcement learning, online learning in causal inference and bandit in large language models.

## Acknowledgements

Zhiheng Zhang is supported by "the Fundamental Research Funds for the Central Universities" (number: 2025110602) of Shanghai University of Finance and Economics.

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

# A    Technical Appendices and Supplementary Material

Appendix B summarizes key symbols in the main text for reference.

Appendix C provides a detailed literature review for better comprehension of the background.

Appendix D and N provide the justification for exposure mapping and model conditions.

Appendix E illustrates the experiments.

Appendix F further analyzes the structure of the exposure mapping and the re-scaled noise.

Appendix G provides the proof the Proposition 1.

Appendix H-I contain the proof of Theorem 1 and Theorem 2, respectively.

Appendix K presents the proofs of Theorem 3-1 in Section 5.

Appendix L provides an algorithm for Non-stochastic Settings.

Appendix M delivers the proof of Theorem 5. Finally, Appendix O includes the auxiliary lemmas.

# B    Notations

| | |
|---|---|
| $\mathcal{K}$ | Real arm set |
| $K$ | Number of arms |
| $\mathcal{U}$ | Unit set |
| $N$ | Number of units |
| $\mathcal{C}$ | Cluster set |
| $C$ | Number of clusters |
| $\nu$ | Instance |
| $\mathcal{E}_0$ | Set of the legitimate instance |
| $\pi$ | Learning policy |
| $\mathcal{R}(T, \pi)$ | Cumulative regret of policy $\pi$ |
| $T$ | Time horizon |
| $T_1$ | Length of the first exploration phase |
| $Y_i(\cdot)$ | Potential outcome of unit $i$ |
| $\tilde{Y}_i(\cdot)$ | Exposure potential outcome of unit $i$ |
| $\mathbf{S}(\cdot)$ | Exposure mapping |
| $\mathbb{H}$ | Adjacency matrix |
| $a_{i,t}$ | Action of unit $i$ |
| $s_{i,t}$ | Exposure action of unit $i$ |
| $A_t$ | Supper arm played $t$ |
| $S_t$ | Exposure super arm played $t$ |
| $S^*$ | Optimal exposure super arm |
| $d_s$ | Number of the exposure arm |
| $\mathcal{U}_s$ | Exposure arm set |
| $\mathcal{U}_{\mathcal{C}}$ | Cluster-wise switchback exposure super arm set |
| $\mathcal{U}_{\mathcal{O}}$ | Set of exposure supper arm that can be triggered by real supper arm |
| $\mathcal{U}_{\mathcal{E}}$ | Legitimate exposure super arm set |
| $\tilde{r}_{i,t}(S)$ | Reward feedback of unit $i$ in round $t$ if exposure super arm $S$ is pulled |
| $\Delta^{(i,j)}$ | ATE between $S_i$ and $S_j$ |
| $\Delta^i$ | ATE between $S^*$ and $S_i$ |
| $\hat{\Delta}_T^{(i,j)}$ | Estimated ATE between $S_i$ and $S_j$ |
| $\hat{R}_t(S)$ | Reward estimator of exposure super arm $S$ |
| $e_\nu(T, \hat{\Delta})$ | Largest ATE estimation error |
| $\mathcal{N}_S^t$ | Observation number of exposure super arm $S$ until round $t$ |

# C    Literature Review

In this section, we present a literature review on network interference within the causality and bandit communities. Additionally, we discuss relevant variants of bandit problems. Finally, we provide a brief summary of recent advancements in the estimation-regret trade-off within the context of MAB.

**Offline causality estimation under network interference.**    In the current causality literature, interference is a well-known concept. It is a violation of the conventional "SUTVA" setting, repre-

| Interference-based MAB | Exposure mapping ($S(i, A, \mathbb{H})$) | Action space ($|\mathcal{U}_{\mathcal{E}}|$) | Clusters ($\mathcal{C}$) | Estimation goal ($\Delta^{(i,j)}$) |
|---|---|---|---|---|
| Simchi-Levi and Wang [2024] | $A$ | $K$ | 1 | $Y(A_i) - Y(A_j)$ |
| Jia et al. [2024] | $Ae_i$ | $K$ | 1 | $\frac{1}{N}\sum_{i'\in\mathcal{U}}(Y_{i'}(i*\mathbf{1}_N) - Y_{i'}(j*\mathbf{1}_N))$ |
| Agarwal et al. [2024] | $Ae_i$ | $K^N$ | $N$ | $\frac{1}{N}\sum_{i'\in\mathcal{U}}(Y_{i'}(A_i) - Y_{i'}(A_j))$ |
| MAB-N (**Ours**) | General $\mathbf{S}(i, A, \mathbb{H})$ | $O(|d_s|^{\mathcal{C}})$ | $\mathcal{C}$ | $\frac{1}{N}\sum_{i'\in\mathcal{U}}(\bar{Y}_{i'}(S_i) - \bar{Y}_{i'}(S_j))$ |

Table 1: MAB-N surrogates the previous bandit under interference as special cases. Here $A_i, A_j \in \mathcal{K}^{\mathcal{U}}$, and $S_i, S_j \in \mathcal{U}_{\mathcal{E}}$. We omit the subscript in Simchi-Levi and Wang [2024] since it only considers sole individual.

senting that one individual's treatment would potentially affect another individual's outcome, which is relevant in practice. Current literature resort to clustering Zhang and Imai [2023], Viviano et al. [2023] or exposure mapping Leung [2022a,b, 2023].

**Bandit under network interference.** Previous attempts are being made to consider the multi-armed bandit problem upon network interference. Agarwal et al. [2024] conduct the Fourier analysis to transform the traditional stochastic multi-armed bandit into a sparse linear bandit. However, in order to reduce the exponential action space, they made a strong assumption of sparsity for network structures, i.e., the number of neighbors of each node is manually upper limited. On the other hand, Jia et al. [2024] analyzes the action space at the other extreme that considers an adversarial bandit setting and thus forces each node to a simultaneous equal arm. It does not consider that the optimal arm could differ for each node or subgroup. Moreover, Xu et al. [2024] further considers the contextual setting under the specific linear structure between the potential outcome and the interference intensity.

**Trade-off between inference (estimation) and regret.** A significant body of research has been dedicated to developing statistical methods for inference in MABs. Numerous studies focus on deriving statistical tests or central limit theorems for MABs while ensuring that the bandit algorithm remains largely unaltered [Hadad et al., 2021, Luedtke and Van Der Laan, 2016, Deshpande et al., 2023, Zhang et al., 2020a, 2021, Han et al., 2022, Dimakopoulou et al., 2017, 2019, 2021], thereby facilitating aggressive regret minimization. However, these works all rely on the SUTVA assumption and fail to account for potential interference between units.

Previous literature upon adaptive inference in multi-armed bandits include Dimakopoulou et al. [2021], Liang and Bojinov [2023] whereas without strict trade-off analysis. To our best knowledge, the only state-of-the-art trade-off result is primarily constructed by Simchi-Levi and Wang [2024] whereas also be cursed by the SUTVA assumption without a network connection. Moreover, Duan et al. [2024] argue that such Pareto-optimality could be further improved, i.e., the regret and estimation error could simultaneously achieve their optimality, if additionally assuming the "covariate diversity" of each node without network interference. Stepping forward, when we shift our attention to the network setting, Jia et al. [2024] is also intuitively aware of the potential "incompatibility" of decision-making and statistical inference: specifically, Jia et al. [2024] emphasizes that the truncated HT estimator directly into the policy learning system is no longer robust because policy learning gives different propensity probabilities to different arms, making the propensity score more extreme.

**Relevant bandit variants: multiple-play bandits, multi-agent bandits, combinatorial bandits, and multi-tasking bandits.** In bandit literature, the problem where a bandit algorithm plays multiple arms in each time period has been a subject of study for a long time. Our work is related to the *multi-play bandit* problem, where the algorithm selects multiple arms in each round and observes their corresponding reward feedback [Anantharam et al., 1987, Uchiya et al., 2010, Komiyama et al., 2015, 2017, Louëdec et al., 2015, Lagrée et al., 2016, Zhou and Tomlin, 2018, Besson and Kaufmann, 2018, Jia et al., 2023, Wang et al., 2023b]. Additionally, this is related to the *multi-agent bandit* problem (including distributed and federated bandits), where multiple agents each pull an arm in every time period. By exchanging observation histories through communication, these agents can collaboratively accelerate the learning process. [Hillel et al., 2013, Szörényi et al., 2013, Wu et al., 2016, Wang et al., 2019, Li and Wang, 2022, He et al., 2022, Wang et al., 2023b]. Furthermore, our work is also connected to the *combinatorial bandit* problem, where the action set consists of a subset of the vertices of a binary hypercube [Cesa-Bianchi and Lugosi, 2012, Chen et al., 2013, 2014, Combes et al., 2015, Qin et al., 2014, Kveton et al., 2015, Li et al., 2016, Saha and Gopalan, 2019, Wang et al., 2023a]. Some of these works account for interference between units, but they typically

| | Estimation (offline) | Regret (online) | Trade-off between Estimation&Regret |
|---|---|---|---|
| Without interference | SUTVA causality | Auer et al. [2002] Burtini et al. [2015] | Simchi-Levi and Wang [2024] Duan et al. [2024] |
| With interference | Leung [2022a,b, 2023] Hudgens and Halloran [2008] Sävje [2024] | Agarwal et al. [2024] Jia et al. [2024] Xu et al. [2024] | **Our paper** |

Table 2: Most related and representative works in causality estimation and regret analysis with (without) network interference.

assume that the interference is either explicitly known to the learning algorithm, or the interference follows a specific pattern. In contrast, our setting makes no such assumptions about the nature or structure of interference between units. Our paper also closely related to the field of multitasking bandits, where the learning algorithm is designed to achieve multiple objectives simultaneously during the learning process. Yang et al. [2017] explore the regulation of the false discovery rate while identifying the best arm. Yao et al. [2021] focus on ensuring the ability to detect whether an intervention has an effect, while also leveraging contextual bandits to tailor consumer actions. Jamieson et al. [2013], Cho et al. [2024] aim to minimize cumulative regret while identifying the best arm with minimal sample complexity.

## D Justification, discussion and future work

**Justification on exposure mapping.** It is a well-known concept in causality. From a statistical perspective, it serves as a functional tool for mapping a high-dimensional action space to a low-dimensional manifold; from a machine learning standpoint, it can be interpreted as a specialized input representation layer. However, its utility has not been fully explored in interference-based online learning settings like Bandits. Interference-based bandit referred to as exposure mapping has recently been explored in Jia et al. [2024] to our knowledge. This additionally assumes the intensity of interference decays with distance. Still, the low-dimensional vectors from their exposure mapping are not involved in the computation of the target regret. In contrast, their regret, directly uses the adversarial setting that "the original super arm must be a vector of the form $a * \mathbf{1}^N, a \in \mathcal{K}$", which is limited in realistic compared to our settings, e.g. when the optimal arm takes place when the individuals in the network are assigned to different treatments; to tackle this problem, although Agarwal et al. [2024] can identify the best arm beyond $a * \mathbf{1}^N, a \in \mathcal{K}$, their approach relies on a stronger assumption: the rewards of each node are influenced solely by its limited first-order neighbors, and the number of these neighbors is significantly smaller than $N$. In sum, our paper first presents an integration of exposure mapping with bandit regret frameworks and demonstrates its generality and applicability.

**Justification on Condition 1.** Condition 1 states that $\mathcal{U}_{\mathcal{E}} \geq 2$ is not empty. It is already weaker than the previous interference-based bandit setting [Jia et al., 2024, Agarwal et al., 2024] whereas it could be further relaxed. We consider the generalized metric to describe the distance between $\mathcal{U}_{\mathcal{O}}$ and $S_t \in \mathcal{U}_{\mathcal{C}}$: $\mathcal{D}(\mathcal{U}_{\mathcal{O}}, S_t) := \min_{S' \in \mathcal{U}_{\mathcal{O}}} ||S' - S_t||_1$ via Manhattan distance. When the number of clusters grows, the action space $|\mathcal{U}_{\mathcal{C}}|$ exponentially expands and their compatibility $\mathcal{D}(\mathcal{U}_{\mathcal{C}}, \mathcal{U}_{\mathcal{O}})$ also decreases. These previous literature and Condition 1 all satisfy $\mathcal{D}(\mathcal{U}_{\mathcal{C}}, \mathcal{U}_{\mathcal{O}}) = 0$, and the former literature together with additional network structure [Agrawal and Goyal, 2012] or interference intensity [Jia et al., 2024] assumption as above. In Appendix N we claim that under the weakened assumption $\mathcal{D}(\mathcal{U}_{\mathcal{C}}, \mathcal{U}_{\mathcal{O}}) \leq \epsilon$, where $\epsilon > 0$ is a prior constant, our model remains capable of reasonable modeling by appropriately adjusting the definition of exposure-based rewards accordingly. The interplay between this assumption and other well-known assumptions, such as the neighbor sparsity assumption Agarwal et al. [2024], the decaying interference assumption Jia et al. [2024], and the approximate interference assumption Leung [2022a], is left as an avenue for future work.

Moreover, a natural next step is to deepen the theoretical and methodological connections between networked bandit design [Wang et al., 2025] and causal inference. In particular, our analysis of Pareto-optimal trade-offs provides a foundation for extending the notion of *learnability* in adaptive designs toward formal *identifiability* in causal models. Future research may incorporate the theory of partial identification [Zhang and Su, 2024, Zhang, 2024] to characterize the minimal information required for interference-aware effect estimation, thereby turning online exploration into a process of

progressively tightening causal bounds. Another promising direction is to embed robust and proxy-based causal adjustments [Zhang et al., 2023] into the online decision loop, allowing adaptive designs to remain valid in the presence of latent or noisy confounders. Extending our results to dynamic and time-varying network structures would further connect to recent advances in dynamic Granger-type causal analysis [Zhang et al., 2020b], while bridging to active treatment-effect estimation with limited sampling budgets [Zhang et al., 2025]. Finally, structural and geometric constraints arising from network topology or combinatorial design feasibility [Su et al., 2023, Zhang, 2022] suggest that future online experiments could jointly optimize over both exploration–regret trade-offs and the attainable identification sets of causal effects, forming a unified theory of inference-aware sequential design under complex interference.

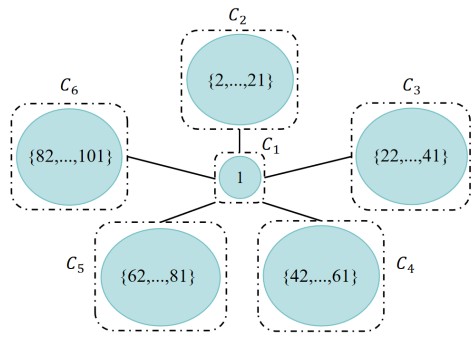

Figure 2: Network structure.

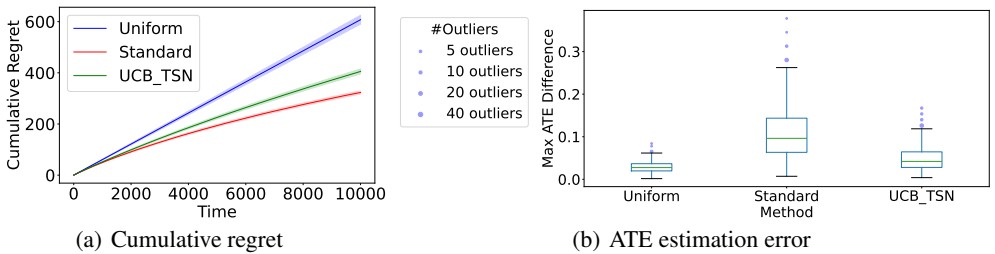

(a) Cumulative regret

(b) ATE estimation error

Figure 3: Experimental results.

## E Experiments

**Setup.** We consider a network consisting of 101 units. Specifically, there is a central cluster $C_1 = \{1\}$ that contains a single unit, which is connected to every unit in the five peripheral clusters $C_2, \ldots, C_6$ (namely, $C_2 = \{2, \ldots, 21\}$, $C_3 = \{22, \ldots, 41\}$, $C_4 = \{42, \ldots, 61\}$, $C_5 = \{62, \ldots, 81\}$, and $C_6 = \{82, \ldots, 101\}$, with each outer cluster containing 20 units, as shown in Fig. 2). We set the action set as $\mathcal{K} = \{0, 1\}$. Inspired by [Leung, 2022a, Gao and Ding, 2023], we define the exposure mapping as $\mathbf{S}(i, A, \mathcal{H}) = \mathbf{1}\left\{ \frac{\sum_j h_{ij} a_j}{\sum_j h_{ij}} \in \left[0, \frac{1}{2}\right) \right\}$, which explores the influence of the proportion of neighbors taking action 1 on each unit; this exposure mapping implies that $d_s = 2$. For every $S \in \mathcal{U}_{\mathcal{E}}$, we define $\mathbb{P}(A_t = A \mid S)$ as uniform sampling. Moreover, for each selected super arm corresponding to an exposure $S$, the reward is sampled from a Bernoulli distribution.

We evaluate the performance of `UCB-TSN` ($T_1 = \sqrt{|\mathcal{U}_{\mathcal{E}}| T}$) against two baseline methods: Standard (i.e., `UCB-TSN` with $T_1 = 0$) and Uniform (i.e., `UCB-TSN` with $T_1 = T$). Each algorithm is executed 1000 times, and we report the averaged results.

**Results.** The simulation results are shown in Fig. 3(a) and Fig. 3(b). As seen in Fig. 3(a), both the Standard method and `UCB-TSN` achieve the lowest cumulative regret, while Uniform exhibits the highest cumulative regret. Fig. 3(b) presents a box plot of the maximum ATE estimation error, $e_\nu(T, \hat{\Delta})$, where the green line represents the median. The results indicate that `UCB-TSN` and Uniform yield lower ATE estimation errors with compact interquartile ranges and few outliers, whereas the Standard method shows a wider spread of errors and multiple outliers. This relatively poorer performance of the Standard method in statistical estimation is due to its lower frequency of exploring sub-optimal arms compared to Uniform and `UCB-TSN`. Our code is available at: `https://github.com/ZHzhang01/NeurIPS2025-Online-ABtest`.

## F The Discussion of Exposure Mapping and Noise Rescaling

We denote the policy and exposure reward inheriting from Leung [2022a] as $\mathbb{P}_{\text{Leung}}$ and $\tilde{Y}_{i,\text{Leung}}(\cdot)$, respectively. Considering Eq (3), we take the exposure mapping function's output as $d_s$ cardinality without loss of generality. We choose $\mathbb{P}(A_t = A \mid S_t) := \mathbb{P}_{\text{Leung}}(A_t = A \mid S_t \boldsymbol{e}_i)$ then $\forall S_t \boldsymbol{e}_i =$

$s, \tilde{Y}_{i,\text{Leung}}(s) = \sum_{A \in \mathcal{K}^{\mathcal{U}}} \mathbb{P}_{\text{Leung}}(A_t = A \mid s)Y_i(A) = \sum_{A \in \mathcal{K}^{\mathcal{U}}} \mathbb{P}(A_t = A \mid S_t)Y_i(A) = \tilde{Y}_i(S_t)$.
Hence our exposure-based reward notation is generalized from Leung [2022a].

Moreover, we discuss the re-scaling of noise. When $\forall S \in \mathcal{U}_{\mathcal{E}}, |\{A : \{\mathbf{S}(i, A, \mathbb{H})\}_{i \in \mathcal{U}} = S\}| = 1$, it naturally leads to the variance proxy $\sigma^2 = \frac{1}{N}$ of the Sub-Gaussian variables $\sum_{i \in \mathcal{U}} \tilde{r}_{i,t}(S)/N$. Hence, we mainly consider other cases. Notice that Eq (3) defines

$$[\tilde{Y}_i(S_t), \tilde{r}_{i,t}(S_t)]^\top := \sum_{A \in \mathcal{K}^{\mathcal{U}}} [Y_i(A), r_{i,t}(A)]^\top \mathbb{P}(A_t = A \mid S_t),$$

namely, for each $S_t$, practitioners select random legitimate $r_{i,t}(A_t)$ to approximate $\tilde{r}_{i,t}(S_t)$, each with probability $\mathbb{P}(A_t = A \mid S_t)$. The randomness includes the sub-Gaussian noise and sampling noise. It follows that for all $m \in \mathbb{R}$,

$$
\begin{aligned}
&\mathbb{E}\left[\exp\left(\frac{m}{N}\sum_{i \in \mathcal{U}}\left(\tilde{r}_{i,t}(S_t) - \tilde{Y}_i(S_t)\right)\right) \mid A_t = A\right] \\
&= \mathbb{E}\left[\exp\left(\frac{m}{N}\sum_{i \in \mathcal{U}}\left(r_{i,t}(A) - Y_i(A) + Y_i(A) - \tilde{Y}_i(S_t)\right)\right)\right] \\
&= \exp\left(\frac{m}{N}\sum_{i \in \mathcal{U}}\left(Y_i(A) - \tilde{Y}_i(S_t)\right)\right)\mathbb{E}\left[\exp\left(\frac{m}{N}\sum_{i \in \mathcal{U}}\left(r_{i,t}(A) - Y_i(A)\right)\right)\right] \\
&\leq \exp\left(\frac{m}{N}\sum_{i \in \mathcal{U}}\left(Y_i(A) - \tilde{Y}_i(S_t)\right)\right)\exp\left(\frac{m^2}{2N}\right).
\end{aligned}
\tag{8}
$$

Taking expectation upon both sides of Eq (8), it leads to

$$\mathbb{E}\left[\mathbb{E}\left[\exp\left(\frac{m}{N}\sum_{i \in \mathcal{U}}\left(\tilde{r}_{i,t}(S_t) - \tilde{Y}_i(S_t)\right)\right) \mid A_t = A\right]\right] \leq \exp\left(\frac{m^2}{2N}\right)\mathbb{E}\left[\exp\left(\frac{m}{N}\sum_{i \in \mathcal{U}}\left(Y_i(A) - \tilde{Y}_i(S_t)\right)\right)\right].$$
$$\tag{9}$$

According to the boundary $\frac{1}{N}\sum_{i \in \mathcal{U}}(Y_i(A) - \tilde{Y}_i(S_t)) \in [-1, 1]$, it is natural to derive

$$\mathbb{E}\left[\exp\left(\frac{m}{N}\sum_{i \in \mathcal{U}}\left(Y_i(A) - \tilde{Y}_i(S_t)\right)\right)\right] \leq \cosh(m/2) \leq \exp(m^2/8).$$

Then Eq (9) achieves that

$$(9) \leq \exp\left(\frac{m^2}{2N}\right)\exp(m^2/8) = \exp\left(\frac{m^2}{2}\left(\frac{1}{N} + \frac{1}{4}\right)\right). \tag{10}$$

Therefore the Sub-Gaussian variables $\sum_{i \in \mathcal{U}} \tilde{r}_{i,t}(S)/N$ could achieve the variance proxy at most $1/N + 1/4$. In the following part, we set the variance proxy as $\sigma^2 = 2$ without loss of generality.

**Comment on the order of node number $N$.** For a supplement, in Theorem/Corollary 3-1, we additionally consider the order of node number $N$. (i) In Theorem 3, we emphasize that if $\forall S' \in \mathcal{U}_{\mathcal{E}}, |\{A : \{S(i, A, \mathbb{H})\}_{i \in \mathcal{U}} = S'\}| = 1$, namely, there is only one legitimate $A$ which is compatible with each exposure arm $S'$, then Theorem 3 could be strengthened as $\mathbb{E}[|\hat{\Delta}_T^{(i,j)} - \Delta^{(i,j)}|] = \tilde{O}(\sqrt{|\mathcal{U}_{\mathcal{E}}|/T_1 N})$. Take the cluster-wise switchback experiment $(S(i, A, \mathbb{H}) = a_{i,t})$ for instance, which is the generalized case of Jia et al. [2024]. In this case, since $|\mathcal{U}_{\mathcal{E}}| = K^C \ll N$ via manually selecting $d_s, C$, then we can claim the estimation is consistent when $N \to +\infty$[9]. Moreover, in the setting of Agarwal et al. [2024], it is equivalent to the case $C = N$ and thus the result in Theorem 3 is transformed as $\tilde{O}(\sqrt{K^N/T_1 N})$. It serves as a supplement of Proposition 1, claiming that not only

---

[9]Essentially, it is due to the re-scaling of noise. Under the one-to-one mapping in this paragraph, the result is intuitive since $\sum_{i \in \mathcal{U}} \tilde{r}_{i,t}(S)/N$ exhibits a re-scaled Sub-Gaussian noise with variance proxy $1/N$. It degenerates to the offline setting when $N \to +\infty$. Otherwise, we could only ensure $\sum_{i \in \mathcal{U}} \tilde{r}_{i,t}(S)/N$ is a Sub-Gaussian noise with variance proxy $(1/N + 1/4)$. We defer the details to Appendix F.

the regret but also the estimation error is hard to control without exposure mapping. (ii) Analogously, in Theorem 4, the result is transformed to $\mathcal{R}_\nu(T, \pi) = \tilde{O}\big(\sqrt{|\mathcal{U}_\mathcal{E}|T/N} + T_1\big)$ under the above one-to-one mapping. (iii) Finally, in Corollary 1, the trade-off is transferred to be $\tilde{O}(\sqrt{|\mathcal{U}_\mathcal{E}|/N})$ when we slightly modify the condition of $T_1$ as $T_1 \geq \sqrt{|\mathcal{U}_\mathcal{E}|T/N} \vee |\mathcal{U}_\mathcal{E}|$. This result is also aligned with the proof of Theorem 1.

## G   Proof of Proposition 1

**Proof 1 (Proof of Proposition 1)** *We here define $\mathcal{K}^\mathcal{U} := \{A_k\}_{k=1}^{K^N}$ as the set of the super arm. Define a MAB instance $\nu_1 \in \mathcal{E}_0$ that $Y_i(A) = \Delta \mathbf{1}\{A = A_1\}$ for all $i \in \mathcal{U}$ and $A \in \mathcal{K}^\mathcal{U}$, where $\Delta \in [0, 1/2]$ will be defined later. We suppose that the noise of all unit $\eta_{i,t}$ follows a $\mathcal{N}(0, 1)$ Gaussian distribution, and therefore the normalized noise of the super arm $(1/N)\sum_{i \in \mathcal{U}} \eta_{i,t}$ follows a $\mathcal{N}(0, 1/N)$ Gaussian distribution. Hence, we have $1/N \sum_{i \in \mathcal{U}} Y_i(A_1) = \Delta$ and $1/N \sum_{i \in \mathcal{U}} Y_i(A_k) = 0$ for all $k \in [K^N]/\{1\}$. This implies in $\nu_1$, $A_1 = A^*$ is the best arm with potential outcome $\Delta$ and $A \neq A_1$ is the sub-optimal arm with potential outcome $0$. Due to*

$$\mathcal{R}_{\nu_1}(T, \pi) = \sum_{k=2}^{K^N} \Delta_k \mathbb{E}_{\nu_1, \pi}[\mathcal{N}_{A_k}^T], \tag{11}$$

*where $\mathcal{N}_{A_k}^T := \sum_{t \in [T]} \mathbf{1}\{A_t = A_k\}$ denotes the number that supper arm $A_k$ is trigger till $T$ and $\Delta_k$ denotes the reward gap between super arm $A_1$ and $A_k$ (i.e., $\Delta_k = (1/N)(\sum_{i \in \mathcal{U}} Y_i(A_1) - Y_i(A_k))$). Suppose the super arm $A_j$, $j = \arg\min_{j \in [K^N]/\{1\}} \mathbb{E}_{\nu_1, \pi}[\mathcal{N}_{A_j}^T]$, then*

$$\mathbb{E}_{\nu_1, \pi}[\mathcal{N}_{A_j}^T] \leq \frac{T}{K^N - 1}. \tag{12}$$

*Besides, we define another $\mathcal{N}(0, 1)$ Gaussian MAB instance $\nu_2 \in \mathcal{E}_0$, where $Y_i'(A) = Y_i(A) + 2\Delta \mathbf{1}\{A = A_j\}$ for all $i \in \mathcal{U}$ and $A \in \mathcal{K}^\mathcal{U}$. In $\nu_2$, $A_j$ is the best arm with potential outcome $2\Delta$. Based on the decomposition of the regret Eq (11), we have*

$$\mathcal{R}_{\nu_1}(T, \pi) \geq \mathbb{P}_{\nu_1, \pi}\big(\mathcal{N}_{A_1}^T \leq T/2\big)\frac{\Delta T}{2}, \quad and \quad \mathcal{R}_{\nu_2}(T, \pi) \geq \mathbb{P}_{\nu_2, \pi}\big(\mathcal{N}_{A_1}^T \geq T/2\big)\frac{\Delta T}{2}. \tag{13}$$

*Let $\mathbb{P}_{\nu_1, \pi}$ and $\mathbb{P}_{\nu_2, \pi}$ denote the probability measures on the canonical bandit model induced by the $T$-round interaction between $\pi$ and $\nu_1$, and $\pi$ and $\nu_2$, respectively. Finally, we have*

$$\begin{aligned}
&\mathcal{R}_{\nu_1}(T, \pi) + \mathcal{R}_{\nu_2}(T, \pi) \\
&\geq \Big(\mathbb{P}_{\nu_1, \pi}\big(\mathcal{N}_{A_1}^T \geq T/2\big) + \mathbb{P}_{\nu_2, \pi}\big(\mathcal{N}_{A_1}^T < T/2\big)\Big)\frac{\Delta T}{2} \\
&\geq exp\Big(-KL(\mathbb{P}_{\nu_1, \pi}, \mathbb{P}_{\nu_2, \pi})\Big)\frac{\Delta T}{4} \\
&\geq exp\Big(-\mathbb{E}_{\nu_1, \pi}[\mathcal{N}_{A_j}^T]KL\Big(\mathcal{N}(0, 1/N), \mathcal{N}(2\Delta, 1/N)\Big)\Big)\frac{\Delta T}{4} \\
&\geq exp\Big(-\mathbb{E}_{\nu_1, \pi}[\mathcal{N}_{A_j}^T]2N\Delta^2\Big)\frac{\Delta T}{4} \\
&\geq exp\Big(-\frac{2TN\Delta^2}{K^N - 1}\Big)\frac{\Delta T}{4},
\end{aligned} \tag{14}$$

*where KL denotes the KL divergence, the second inequality is owing to the Bretagnolle–Huber inequality, the third inequality is due to the Lemma 15.1 in Lattimore and Szepesvári [2020b], the fourth inequality is due to the definition of the noise distribution (i.e., $\mathcal{N}(0, 1/N)$) of the super arm. Finally, select $\Delta = \sqrt{\frac{K^N - 1}{4TN}} \wedge \frac{1}{2}$, based on the above result, we have ($i = 1$ or $2$)*

$$\mathcal{R}_{\nu_i}(T, \pi) \geq \begin{cases} e^{-1/2}\frac{T}{8\sqrt{N}}, & when \ T \leq K^N \\ \frac{e^{-1/2}}{4}\sqrt{\frac{(K^N - 1)T}{N}}, & when \ T \geq K^N. \end{cases} \tag{15}$$

# H Proof of Theorem 1

**Proof 2 (Proof of Theorem 1)** *In this section, to simplify the notations in Section G, we abbreviate $\mathbb{P}_{\nu,\pi}$ as $\mathbb{P}_\nu$ and $\mathbb{E}_{\nu,\pi}$ as $\mathbb{E}_\nu$. We consider two kinds of instances for a fixed policy $\pi$ and a fixed strategy of constructing an ATE estimator $\hat{\Delta}_T$. For the first one (i.e., $\nu_1$), we denote it as $r_{i,t}(A) = f_i(A) + \eta_{i,t}$. Here we let $Y_i(A) := f_i(A) \in [0,1], r_{i,t}(A) \in \{-1,1\}$. It means $r_{i,t}(A) = Rad(\frac{1-f_i(A)}{2}, \frac{1+f_i(A)}{2})$. For each feasible cluster-wise exposure super arm $S \in \mathcal{U}_\mathcal{E}$, recall that*

$$\tilde{Y}_i(S) = \sum_{A \in \mathcal{K}^\mathcal{U}} f_i(A)\mathbb{P}(A_t = A \mid S). \tag{16}$$

*The difference of expected reward of $S, S'$ could be represented by $\Delta_1(S, S') := \frac{1}{N}\sum_{i \in \mathcal{U}}(\tilde{Y}_i(S) - \tilde{Y}_i(S'))$, which is*

$$\Delta_1(S, S') = \frac{1}{N}\sum_{i \in \mathcal{U}}\sum_{A \in \mathcal{K}^\mathcal{U}} f_i(A)\big(\mathbb{P}(A_t = A \mid S) - \mathbb{P}(A_t = A \mid S')\big). \tag{17}$$

*Without loss of generality, we select the feasible super arm to set $\Delta_1(S, S') < 0$. For brevity, we omit the expression of the parentheses in the following text. Namely, we choose $S'$ as the best arm, and $S$ as a sub-optimal arm in $\mathcal{U}_\mathcal{E}$. We choose $S = \arg\min_{S_i \in \mathcal{U}_\mathcal{E}, S_i \neq S'} \Delta_1(S_i, S')\mathbb{E}_{\nu_1}[\mathcal{N}_{S_i}^T]$. In this process, we use $\hat{\Delta}^{(i,j)} := \{\hat{\Delta}_t^{(i,j)}\}_{t \geq 1}, \hat{\Delta} := \{\hat{\Delta}^{(i,j)}\}_{S_i,S_j \in \mathcal{U}_\mathcal{E}}$. We then construct a new MAB instance $\nu_2$ and hope to get a different ATE value. We define it as $r'_{i,t}(A)$. We establish :*

$$r'_{i,t}(A) := \begin{cases} r_{i,t}(A) & \forall A \text{ satisfying } \mathbb{P}(A_t = A \mid S) = 0. \\ Rad(\frac{1-f_i(A)+\alpha}{2}, \frac{1+f_i(A)-\alpha}{2}) & \forall A \text{ satisfying } \mathbb{P}(A_t = A \mid S) > 0. \end{cases} \tag{18}$$

*Here $\alpha > 0$ should be chosen sufficiently small. Remind that following Eq (17), the ATE between super arm $S, S'$ is*

$$\Delta_2 := \Delta_{2,1} + \Delta_{2,2}, \text{ where}$$

$$\Delta_{2,1} := \frac{1}{N}\sum_{i \in \mathcal{U}}\sum_{A \in \mathcal{K}^\mathcal{U}} (f_i(A) - \alpha)\big(\mathbb{P}(A_t = A \mid S) - \mathbb{P}(A_t = A \mid S')\big)\mathbf{1}\{\mathbb{P}(A_t = A \mid S) > 0\},$$

$$\Delta_{2,2} := \frac{1}{N}\sum_{i \in \mathcal{U}}\sum_{A \in \mathcal{K}^\mathcal{U}} f_i(A)\big(\mathbb{P}(A_t = A \mid S) - \mathbb{P}(A_t = A \mid S')\big)\mathbf{1}\{\mathbb{P}(A_t = A \mid S) = 0\}.$$

*Hence, it implies that the ATEs in these two MAB instances, respectively, contain a difference*

$$\begin{aligned} &\Delta_2 - \Delta_1 \\ =&\frac{1}{N}\sum_{i \in \mathcal{U}}\sum_{A \in \mathcal{K}^\mathcal{U}} -\alpha\big(\mathbb{P}(A_t = A \mid S) - \mathbb{P}(A_t = A \mid S')\big)\mathbf{1}\{\mathbb{P}(A_t = A \mid S) > 0\} \\ =&\frac{1}{N}\sum_{i \in \mathcal{U}}\sum_{A \in \mathcal{K}^\mathcal{U}} -\alpha\mathbb{P}(A_t = A \mid S)\mathbf{1}\{\mathbb{P}(A_t = A \mid S) > 0\} \\ =&\frac{1}{N}\sum_{i \in \mathcal{U}}\sum_{A \in \mathcal{K}^\mathcal{U}} -\alpha\mathbb{P}(A_t = A \mid S) = -\alpha < 0. \end{aligned} \tag{19}$$

*Naturally, our setting leads to $0 > \Delta_1 > \Delta_2$. The second equality is because $\mathbb{P}(A_t = A \mid S)\mathbb{P}(A_t = A \mid S') = 0$ when $S \neq S'$. In this sense, we consider a given estimate strategy, which is summarized by $\{\hat{\Delta}_{t'}\}_{t' \in [t]}$. We define a minimum test $\psi(\hat{\Delta}_t) = \arg\min_{i=\{1,2\}} |\hat{\Delta}_t - \Delta_i|$. Naturally, it implies that $\psi(\hat{\Delta}_t) \neq i, i \in \{1,2\}$ is a sufficient condition of $|\hat{\Delta}_t - \Delta_i| \geq \frac{\alpha}{2}$. As a consequence,*

$$\begin{aligned} \inf_{\hat{\Delta}_t}\max_{\nu \in \mathcal{E}_0}\mathbb{P}_\nu\left(|\hat{\Delta}_t - \Delta_\nu| \geq \frac{\alpha}{2}\right) &\geq \inf_{\hat{\Delta}_t}\max_{i \in \{1,2\}}\mathbb{P}_{\nu_i}\left(|\hat{\Delta}_t - \Delta_i| \geq \frac{\alpha}{2}\right) \\ &\geq \inf_{\hat{\Delta}_t}\max_{i \in \{1,2\}}\mathbb{P}_{\nu_i}\left(\psi(\hat{\Delta}_t) \neq i\right) \\ &\geq \inf_\psi\max_{i \in \{1,2\}}\mathbb{P}_{\nu_i}(\psi \neq i). \end{aligned} \tag{20}$$

Here, the probability space is constructed on the exposure arm $\{S(i, A, \mathbb{H})\}_{i \in \mathcal{U}}$ in each time period $t$, and the observed exposure reward. We use the technique in min-max bound. Notice that the original feasible region of MAB instances as $\mathcal{E}_0$; we get

$$
\begin{aligned}
RHS \text{ of } (20) \geq & \frac{1}{2} \inf_{\psi} (\mathbb{P}_{\nu_1}(\psi = 2) + \mathbb{P}_{\nu_2}(\psi = 1)) \\
= & \frac{1}{2}(1 - TV(\mathbb{P}_{\nu_1}, \mathbb{P}_{\nu_2})) \\
\geq & \frac{1}{2}\left[1 - \sqrt{\frac{1}{2}KL(\mathbb{P}_{\nu_1}, \mathbb{P}_{\nu_2})}\right].
\end{aligned}
\tag{21}
$$

We aim to provide an upper bound of KL divergence $KL(\mathbb{P}_{\nu_1}, \mathbb{P}_{\nu_2})$, inspired by the divergence decomposition:

$$
KL(\mathbb{P}_{\nu_1}, \mathbb{P}_{\nu_2}) = \mathbb{E}_{\nu_2}\left[\log\left(\frac{d\mathbb{P}_{\nu_1}}{d\mathbb{P}_{\nu_2}}\right)\right].
\tag{22}
$$

For any instance $\nu \in \{\nu_1, \nu_2\}$, the density function of the series is denoted as (we denote $X_t$ as the observed exposure reward $\{\tilde{r}_{i,t}(S)\}_{i \in \mathcal{U}}$)

$$
\mathbb{P}_\nu(S_1, X_1, \ldots, S_t, X_t) = \prod_{t'=1}^{t} \pi_t(S_t \mid S_1, X_1, \ldots, S_{t'-1}, X_{t'-1}) \mathbb{P}_{\nu,S_t}(X_t).
\tag{23}
$$

Here $\mathbb{P}_{\nu,S}(\cdot)$ denotes the reward density distribution conditioning on arm $S$ in $\nu$. Hence Eq (22) can be transformed as

$$
\begin{aligned}
KL(\mathbb{P}_{\nu_1}, \mathbb{P}_{\nu_2}) = & \sum_{t' \in [t]} \mathbb{E}_{\nu_1} \log\left(\frac{\mathbb{P}_{\nu_1,S_{t'}}(X_{t'})}{\mathbb{P}_{\nu_2,S_{t'}}(X_{t'})}\right) \\
= & \sum_{t' \in [t]} \mathbb{E}_{\nu_1}\left[\mathbb{E}_{\nu_1} \log\left(\frac{\mathbb{P}_{\nu_1,S_{t'}}(X_{t'})}{\mathbb{P}_{\nu_2,S_{t'}}(X_{t'})}\right) \mid S_{t'}\right] \\
= & \sum_{t' \in [t]} \mathbb{E}_{\nu_1}\left[KL(\mathbb{P}_{\nu_1,S_{t'}}(\cdot), \mathbb{P}_{\nu_2,S_{t'}}(\cdot))\right] \\
= & \mathbb{E}_{\nu_1}[\mathcal{N}_S^t] KL(\mathbb{P}_{\nu_1,S}(\cdot), \mathbb{P}_{\nu_2,S}(\cdot)).
\end{aligned}
\tag{24}
$$

The last equation is derived from the construction in Eq (18). We aim to compute $KL(\mathbb{P}_{\nu_1,S}(\cdot), \mathbb{P}_{\nu_2,S}(\cdot))$:

$$
KL(\mathbb{P}_{\nu_1,S}(\cdot), \mathbb{P}_{\nu_2,S}(\cdot)) = \int_X \mathbb{P}_{\nu_1,S}(X) \log\left(\frac{\mathbb{P}_{\nu_1,S}(X)}{\mathbb{P}_{\nu_2,S}(X)}\right) dX \leq qN\alpha^2.
\tag{25}
$$

Here $q$ is a constant via second-order Taylor expansion.

As a consequence, it implies that

$$
KL(\mathbb{P}_{\nu_1}, \mathbb{P}_{\nu_2}) \leq qN\alpha^2 \mathbb{E}_{\nu_1}[\mathcal{N}_S^t] \leq qN\alpha^2 \frac{\mathcal{R}_{\nu_1}(t, \pi)}{|\mathcal{U}_\mathcal{E}||\Delta_1|}.
\tag{26}
$$

The last inequality is due to $S := \arg\min_{S_i \in \mathcal{U}_\mathcal{E}, S_i \neq S'} \Delta_1(S_i, S') \mathbb{E}_{\nu_1}[\mathcal{N}_{S_i}^t]$. Combined with Eq (20), (21), (26):

$$
\inf_{\hat{\Delta}_t} \max_{\nu \in \mathcal{E}_0} \mathbb{P}_\nu\left(\max_{i,j \in \mathcal{U}_\mathcal{E}} |\hat{\Delta}_t^{(i,j)} - \Delta_\nu^{(i,j)}| \geq \frac{\alpha}{2}\right) \geq \frac{1}{2}\left[1 - \sqrt{\frac{1}{2}qN\alpha^2 \frac{\mathcal{R}_{\nu_1}(t, \pi)}{|\mathcal{U}_\mathcal{E}||\Delta_1|}}\right].
\tag{27}
$$

On this basis, we derive the final trade-off as follows:

$$
\begin{aligned}
& \inf_{\hat{\Delta}_t} \max_{\nu \in \mathcal{E}_0} \mathbb{E}_\nu\left(\max_{i,j \in \mathcal{U}_\mathcal{E}} |\hat{\Delta}_t^{(i,j)} - \Delta_\nu^{(i,j)}|\right) \\
\geq & \frac{\alpha}{2} \inf_{\hat{\Delta}_t} \max_{\nu \in \mathcal{E}_0} \mathbb{P}_\nu\left(\max_{i,j \in \mathcal{U}_\mathcal{E}} |\hat{\Delta}_t^{(i,j)} - \Delta_\nu^{(i,j)}| \geq \frac{\alpha}{2}\right) \\
\geq & \frac{\alpha}{4}\left[1 - \alpha\sqrt{\frac{1}{2}qN \frac{\mathcal{R}_{\nu_1}(t, \pi)}{|\mathcal{U}_\mathcal{E}||\Delta_1|}}\right].
\end{aligned}
\tag{28}
$$

*As a consequence, when $t = T$,*

$$
\inf_{\hat{\Delta}_T} \max_{\nu \in \mathcal{E}_0} \mathbb{E}_\nu \left( \max_{i,j \in \mathcal{U}_\mathcal{E}} |\hat{\Delta}_T^{(i,j)} - \Delta_\nu^{(i,j)}| \right) \sqrt{\mathcal{R}_\nu(T, \pi)}
$$
$$
\geq \frac{\alpha}{4} \left[ 1 - \sqrt{\frac{1}{2} q \alpha^2 N \frac{\mathcal{R}_{\nu_1}(T, \pi)}{|\mathcal{U}_\mathcal{E}||\Delta_1|}} \right] \sqrt{\mathcal{R}_{\nu_1}(T, \pi)}.
\tag{29}
$$

*Due to the sqrt-term spans $[0, +\infty]$ with $\alpha \in [0, 1]$, hence we could set $q\alpha^2 N \frac{\mathcal{R}_{\nu_1}(T, \pi)}{|\mathcal{U}_\mathcal{E}||\Delta_1|} = \frac{1}{2}$, then, when $T \geq |\mathcal{U}_\mathcal{E}|$, it leads to*

$$
(29) \geq \frac{\alpha}{8} \sqrt{\frac{|\mathcal{U}_\mathcal{E}||\Delta_1|}{2Nq\alpha^2}} = \Omega_{T,N,K}(\sqrt{\frac{|\mathcal{U}_\mathcal{E}|}{N}}) = \Omega_{T,K}(\sqrt{|\mathcal{U}_\mathcal{E}|}).
\tag{30}
$$

*Theorem 2 also follows. Q.E.D.*

# I   Proof of Theorem 2

**Proof 3 (Proof of Theorem 2)** *We prove such sufficiency via contradiction. On the one hand, suppose that the MAB pair $\{\pi, \hat{\Delta}\}$ satisfies $\max_{\nu \in \mathcal{E}_0} \left( \sqrt{\mathcal{R}_\nu(T, \pi)} e_\nu(T, \hat{\Delta}) \right) = \tilde{O}(\sqrt{|\mathcal{U}_\mathcal{E}|})$. If it is not Pareto-optimal, it is equivalent to claim that there is another pair $\{\pi', \hat{\Delta}'\}$ to dominate $\{\pi, \hat{\Delta}\}$. In this sense, according to Theorem 1, there exists an instance $\nu'$ such that $\sqrt{\mathcal{R}_{\nu'}(T, \pi')} e_{\nu'}(T, \hat{\Delta}') = \Omega(\sqrt{|\mathcal{U}_\mathcal{E}|})$. Moreover, according to the definition of Pareto-dominance, there further exists another instance $\nu''$, such that $\forall \otimes \in \{K, T\}, \sqrt{|\mathcal{U}_\mathcal{E}|} \prec_\otimes \sqrt{\mathcal{R}_{\nu''}(T, \pi)} e_{\nu''}(T, \hat{\Delta})$. It is a contradiction.*

**Remark 1** *On the other hand, we additionally consider the proof of necessity part, also by contradiction. It is a rigorous refinement of Theorem.5 in Simchi-Levi and Wang [2024] with the extension to the network interference case. We additionally condition that $\mathcal{R}_\nu(T, \pi)$ and $e_\nu(T, \hat{\Delta})$ could both be lower bounded by a polynomial form of $T$, i.e., the Pareto-dominance is only considered in the region of $\mathcal{V}_{lower} := \{\nu : \mathcal{R}_\nu(T, \pi) = \Omega(T^\alpha), e_\nu(T, \hat{\Delta}) = \Omega(\sqrt{|\mathcal{U}_\mathcal{E}|}T^\beta)\}$, where $\alpha > 0, \beta < 0$ are constants. Recalling our goal is to prove any Pareto-optimal pair $\{\pi, \hat{\Delta}\}$ satisfies*

$$
\max_{\nu \in \mathcal{V}_{lower}} \left( \sqrt{\mathcal{R}_\nu(T, \pi)} e_\nu(T, \hat{\Delta}) \right) = \tilde{O}\left( \sqrt{|\mathcal{U}_\mathcal{E}|} \right).
$$

*Suppose that for a Pareto-optimal pair, there exist hard instances $\nu^* \in \mathcal{V}_{hard} \subseteq \mathcal{V}_{front} \cap \mathcal{V}_{lower} \subseteq \mathcal{E}_0$ such that (here $\mathcal{V}_{front} := \{\nu : (\sqrt{\mathcal{R}_\nu(T, \pi)}, e_\nu(T, \hat{\Delta})) \in \mathcal{F}(\pi, \hat{\Delta})\}$):*

$$
\forall \nu^* \in \mathcal{V}_{hard}, \sqrt{\mathcal{R}_{\nu^*}(T, \pi)} e_{\nu^*}(T, \hat{\Delta}) > C\sqrt{|\mathcal{U}_\mathcal{E}|}, \text{when } T \text{ is sufficiently large.}
$$

*Here, $C$ is a constant. According to our condition, it induces that $\mathcal{R}_\nu(T, \pi) \succ_T C_1 T^{2\alpha_1}$, $e_\nu(T, \hat{\Delta}) \succ_T C_2|\mathcal{U}_\mathcal{E}|^{1/2}T^{\alpha_2}$, where $C_1, C_2 \geq 0, C_1 C_2 = C, \alpha_1 + \alpha_2 > 0, \alpha_2 \leq 0, \alpha_1 \in [0, 1/2]$ since the regret is bounded as $O(T)$. It indicates that $\alpha_2 \geq -1/2$. On this basis, we could construct feasible pair $\{\pi_{alg}, \hat{\Delta}_{alg}\}$ via selecting suitable $T_1 := T^{-2\alpha_2}$ in Algorithm 1 to satisfy $e_\nu(T, \hat{\Delta}) \simeq_T e_\nu(T, \hat{\Delta})^{10}$. According to Theorem 1, it follows that the pair $\{\pi_{alg}, \hat{\Delta}_{alg}\}$ would Pareto-dominate the original $\{\pi, \hat{\Delta}\}$. Contradiction.*

# J   Algorithm UCB-Two Stage-Network

The `UCB-TSN` algorithm operates in two phases: an initial round-robin exploration phase over all exposure super arms to obtain empirical estimates of their rewards, followed by a UCB-based selection phase where, at each time step, the super arm with the highest upper confidence bound is chosen for sampling.

---

[10]Here $\simeq$ is the combination of $\succ$ and $\prec$.

**Algorithm 1** UCB-Two Stage-Network (UCB-TSN)

---

**Input:** arm set $\mathcal{A}$, time $\{T_1, T\}$, unit number $N$, exposure super arm set $\mathcal{U}_{\mathcal{E}}$, estimator set $\{\hat{R}_0(S) = 0\}_{S \in \mathcal{U}_{\mathcal{E}}}$, $\{\mathcal{N}_0^S = 0\}_{S \in \mathcal{U}_{\mathcal{E}}}$, $\{\text{UCB}_{0,S} = 0\}_{S \in \mathcal{U}_{\mathcal{E}}}$, counter $k = 1$
**for** $t = 1 : T_1$ **do**
  Select exposure super arm $S_t = S_k$ and implement $\texttt{Sampling}(S_t)$
  Set $k = k + 1$ if $k + 1 \leq |\mathcal{U}_{\mathcal{E}}|$, else set $k = 1$
**end for**
For all $S_i, S_j \in \mathcal{U}_{\mathcal{E}}$, $S_i \neq S_j$, output $\hat{\Delta}_T^{(i,j)} = \hat{R}_{T_1}(S_i) - \hat{R}_{T_1}(S_j)$
**for** $t = T_1 + 1 : T$ **do**
  Select $S_t = \arg \max_{S \in \mathcal{U}_{\mathcal{E}}} \text{UCB}_{t-1,S}$ and implement $\texttt{Sampling}(S_t)$
**end for**
# Parameter 1: $\mathcal{N}_S^t = \sum_{t'=1}^t \mathbf{1}\{S_{t'} = S\}$
# Parameter 2: $\hat{R}_t(S) = \left(\hat{R}_{t-1}(S)\mathcal{N}_S^{t-1} + \mathbf{1}\{S_t = S\}\frac{1}{N}\sum_{i \in \mathcal{U}} \tilde{r}_{i,t}(S)\right)/\mathcal{N}_S^t$
# Parameter 3: $\text{UCB}_{t,S} = \hat{R}_t(S) + \sqrt{18 \log(1/\delta)/\mathcal{N}_S^t}$

---

**Algorithm 2** $\texttt{Sampling}$

---

**Input:** $S_t$
Derive the set $\{Z_{l'}\}_{l' \in [l]}$ such that $\{\mathbf{S}(i, Z_{l'}, \mathbb{H})\}_{i \in \mathcal{U}} = S_t, \forall l' \in [l]$; sample $A_t$ from set $\{Z_{l'}\}_{l \in [l']}$ based on $\mathbb{P}(A_t = A \mid S_t)$, pull $A_t$, and observe reward $\{\tilde{r}_{i,t}(S_t) = r_{i,t}(A_t)\}_{i \in \mathcal{U}}$

---

# K   Proof of Theorems in Section 5

## K.1   Proof of Theorem 3

**Proof 4 (Proof of Theorem 3)** *Based on the design of the Algorithm 1, in the first phase, we have*
$\mathcal{N}_S^{T_1} \geq \lfloor \frac{T_1}{|\mathcal{U}_{\mathcal{E}}|} \rfloor \geq 1$ *for all* $S \in \mathcal{U}_{\mathcal{E}}$. *Define the good event as* $\mathcal{E}_{T_1} := \left\{ \hat{R}_{T_1}(S) - \frac{1}{N}\sum_{i \in \mathcal{U}} \tilde{Y}_i(S) \leq \sqrt{4\log(T_1|\mathcal{U}_{\mathcal{E}}|)/\mathcal{N}_S^{T_1}}, \forall S \in \mathcal{U}_{\mathcal{E}} \right\}$ *and its complement as* $\mathcal{E}_{T_1}^c$. *Based on the previous discussion, the sub-Gaussian proxy of any exposure super arm's reward distribution is at most* 2, *then based on the Hoffeding inequality (Lemma 4), we have for a exposure super arm* $S \in \mathcal{U}_{\mathcal{E}}$:

$$\mathbb{P}\left( \hat{R}_t(S) - \frac{1}{N}\sum_{i \in \mathcal{U}} \tilde{Y}_i(S) > a \right) \leq e^{-\frac{\mathcal{N}_S^t a^2}{4}}, \tag{31}$$

*substituting* $t = T_1$ *and* $a = \sqrt{\frac{4\log(T_1|\mathcal{U}_{\mathcal{E}}|)}{\mathcal{N}_S^{T_1}}}$ *into Eq (31) and we can derive*

$$\mathbb{P}\left( \hat{R}_{T_1}(S) - \frac{1}{N}\sum_{i \in \mathcal{U}} \tilde{Y}_i(S) > \sqrt{\frac{4\log\left(T_1|\mathcal{U}_{\mathcal{E}}|\right)}{\mathcal{N}_S^{T_1}}} \right) \leq \frac{1}{T_1|\mathcal{U}_{\mathcal{E}}|}. \tag{32}$$

*Utilize the union bound, there is*

$$
\begin{aligned}
\mathbb{P}\left(\mathcal{E}_{T_1}^c\right) &\leq \sum_{S \in \mathcal{U}_{\mathcal{E}}} \mathbb{P}\left( \left\{ \hat{R}_{T_1}(S) - \frac{1}{N}\sum_{i \in \mathcal{U}} \tilde{Y}_i(S) > \sqrt{\frac{4\log\left(T_1|\mathcal{U}_{\mathcal{E}}|\right)}{\mathcal{N}_S^t}} \right\} \right) \\
&\leq \sum_{S \in \mathcal{U}_{\mathcal{E}}} \frac{1}{T_1|\mathcal{U}_{\mathcal{E}}|} \\
&\leq \frac{1}{T_1},
\end{aligned} \tag{33}
$$

*and $\mathbb{P}(\mathcal{E}_{T_1}) \geq 1 - \frac{1}{T_1}$. Therefore, for all $S_i, S_j \in \mathcal{U}_{\mathcal{E}}$, we have:*

$$\mathbb{E}\left[\left|\Delta^{(i,j)} - \hat{\Delta}_T^{(i,j)}\right|\right]$$

$$\leq \mathbb{P}(\mathcal{E}_{T_1})\mathbb{E}\left[\left|\Delta^{(i,j)} - \hat{\Delta}_T^{(i,j)}\right| \mid \mathcal{E}_{T_1}\right] + \mathbb{P}(\mathcal{E}_{T_1}^c)\mathbb{E}\left[\left|\Delta^{(i,j)} - \hat{\Delta}_T^{(i,j)}\right| \mid \mathcal{E}_{T_1}^c\right]$$

$$\leq \mathbb{P}(\mathcal{E}_{T_1})\mathbb{E}\left[\left|\hat{R}_t(S) - \frac{1}{N}\sum_{i'\in\mathcal{U}}\tilde{Y}_{i'}(S_i)\right| + \left|\hat{R}_t(S) - \frac{1}{N}\sum_{i'\in\mathcal{U}}\tilde{Y}_{i'}(S_j)\right| \mid \mathcal{E}_{T_1}\right] + \frac{1}{T_1} \qquad (34)$$

$$\leq 2\sqrt{\frac{4\log\left(T_1|\mathcal{U}_{\mathcal{E}}|\right)}{\lfloor\frac{T_1}{|\mathcal{U}_{\mathcal{E}}|}\rfloor}} + \frac{1}{T_1}$$

$$= \tilde{O}\left(\sqrt{\frac{|\mathcal{U}_{\mathcal{E}}|}{T_1}}\right),$$

*where the second inequality is owing to the triangle inequality and $\Delta^{(i,j)}$ and $\hat{\Delta}_T^{(i,j)} \in [0,1]$, and the last inequality is owing to $\mathcal{N}_S^{T_1} \geq \lfloor\frac{T_1}{|\mathcal{U}_{\mathcal{E}}|}\rfloor$. Here we finish the proof of Theorem 3.*

### K.2   Proof of Theorem 4

In this section, we will first provide an instance-dependent regret upper bound (in the following Lemma 1), and then, we will provide an instance-independent regret upper bound based on the instance-dependent one.

**Lemma 1 (Instance-dependent regret)** *Given any instance that satisfies Condition 1. The regret of the `UCB-TSN` can be upper bounded as follows*

$$\mathcal{R}(T,\pi) = O\left(\sum_{S_i\neq S^*,\Delta^i>0}\frac{\log(T)}{\Delta^i} + T_1\right). \qquad (35)$$

**Proof 5 (Proof of Lemma 1)** *Define $\mathcal{N}_S^{(t,T)} = \sum_{t'=t}^T \mathbf{1}\{S_{t'} = S\}$. Besides, define the good event for $S_i$ as:*

$$\mathcal{E}_i = \left\{\frac{1}{N}\sum_{i'\in\mathcal{U}}\tilde{Y}_{i'}(S^*) \leq UCB_{t,S^*}, \; \forall t \in [T_1+1, T]\right\} \cap \left\{\hat{R}_{\mathcal{T}_i,S_i} + \sqrt{\frac{18\log\left(\frac{1}{\delta}\right)}{\mathcal{T}_i}} \leq \frac{1}{N}\sum_{i'\in\mathcal{U}}\tilde{Y}_{i'}(S^*)\right\},$$

*where $\mathcal{T}_i = \frac{72\log(1/\delta)}{(\Delta^i)^2}$ and we utilize $\hat{R}_{\mathcal{T}_i,S_i}$ to represent $\hat{R}_t(S_i)$ when $\mathcal{N}_{S_i}^t = \mathcal{T}_i$. Based on Lemma 2, we have $\mathbb{P}(\mathcal{E}_i) \geq 1 - (T - T_1 + 1)\delta$ and its complement has $\mathbb{P}(\mathcal{E}_i^c) \leq (T - T_1 + 1)\delta$.*

*We can decompose and bound the regret as*

$$\mathcal{R}(T,\pi) = \frac{T}{N}\sum_{i\in\mathcal{U}}\tilde{Y}_i(S^*) - \mathbb{E}_\pi\left[\sum_{t\in[T]}\sum_{i\in\mathcal{U}}\tilde{r}_{i,t}(S_t)\right],$$

$$\leq \underbrace{\sum_{S_i\neq S^*,\Delta^i>0}\Delta^i\mathbb{E}_\pi\left[\mathcal{N}_{S_i}^{(T_1+1,T)}\right]}_{\text{regret in second phase}} + \underbrace{\lceil\frac{T_1}{\mathcal{U}_{\mathcal{E}}}\rceil\sum_{S_i\neq S^*}\Delta^i}_{\text{regret in first phase}}$$

$$= \sum_{S_i\neq S^*,\Delta^i>0}\left(\Delta^i\mathbb{E}_\pi\left[\mathcal{N}_{S_i}^{(T_1+1,T)} \mid \mathcal{E}_i\right] + \Delta^i\mathbb{E}_\pi\left[\mathcal{N}_{S_i}^{(T_1+1,T)} \mid \mathcal{E}_i^c\right]\right) + \lceil\frac{T_1}{\mathcal{U}_{\mathcal{E}}}\rceil\sum_{S_i\neq S^*}\Delta^i$$

$$\leq \sum_{S_i\neq S^*,\Delta^i>0}\Delta^i\mathbb{E}_\pi\left[\mathcal{N}_{S_i}^{(T_1+1,T)} \mid \mathcal{E}_i\right] + T^2\delta + \lceil\frac{T_1}{\mathcal{U}_{\mathcal{E}}}\rceil\sum_{S_i\neq S^*}\Delta^i.$$

$$(36)$$

Besides, we want to show that under the event $\mathcal{E}_i$, we have $\mathcal{N}_{S_i}^{(T_1+1,T)} \leq \mathcal{T}_i$. If $T_1 = T$, then this inequality trivially holds. If $T_1 < T$, suppose $\mathcal{N}_{S_i}^{(T_1+1,T)} > \mathcal{T}_i$, then, there exists a time $t_i \in [T_1 + 1, T]$, such that $S_{t_i} = S_i$ ($S_i$ is pulled in round $t_i$), and $\mathcal{N}_{S_i}^{(t_i,T)} = \mathcal{T}_i + 1$. Based on the exploration strategy in Algorithm 1, we have $UCB_{t_i-1,S_i} \geq UCB_{t_i-1,S^*}$. However, based on the definition of the event $\mathcal{E}_i$, we have

$$UCB_{t_i-1,S^*} \geq \frac{1}{N} \sum_{i' \in \mathcal{U}} \tilde{Y}_{i'}(S^*)$$

$$> \hat{R}_{\mathcal{T}_i,S_i} + \sqrt{\frac{18 \log(1/\delta)}{\mathcal{T}_i}}$$

$$= \hat{R}_{t_i-1}(S_i) + \sqrt{\frac{18 \log(1/\delta)}{\mathcal{N}_{S_i}^{t_i-1}}}$$

$$= UCB_{t_i-1,S_i},$$

which contradicts the previous assumption. Therefore, under the event $\mathcal{E}_i$, we have $\mathcal{N}_{S_k}^T \leq \mathcal{T}_k$. Substituting this result and $\delta = 1/T^2$ into Eq (36), we have

$$
\begin{aligned}
\mathcal{R}(T,\pi) &\leq \sum_{S_i \neq S^*, \Delta^i > 0} \Delta^i \mathbb{E}_\pi \left[ \mathcal{N}_{S_i}^{(T_1+1,T)} \mid \mathcal{E}_i \right] + T^2 \delta + \lceil \frac{T_1}{\mathcal{U}_\mathcal{E}} \rceil \sum_{S_i \neq S^*} \Delta^i \\
&\leq \sum_{S_i \neq S^*, \Delta^i > 0} \Delta^i \mathcal{T}_i + 1 + \lceil \frac{T_1}{\mathcal{U}_\mathcal{E}} \rceil \sum_{S_i \neq S^*} \Delta^i \\
&\leq \sum_{S_i \neq S^*, \Delta^i > 0} \frac{144 \log(T)}{\Delta^i} + 1 + \lceil \frac{T_1}{\mathcal{U}_\mathcal{E}} \rceil \sum_{S_i \neq S^*} \Delta^i \\
&= O\left( \sum_{S_i \neq S^*, \Delta^i > 0} \frac{\log(T)}{\Delta^i} + \lceil \frac{T_1}{\mathcal{U}_\mathcal{E}} \rceil \sum_{S_i \neq S^*} \Delta^i \right).
\end{aligned}
\tag{37}
$$

Here we finish the proof of Lemma 1.

The proof of Lemma 1 relies on the following Lemma 2.

**Lemma 2** We have $\mathbb{P}(\mathcal{E}_i) \geq 1 - (T - T_1 + 1)\delta$ for all $S_i$ satisfies $S_i \neq S^*$ and $\Delta^i > 0$.

**Proof 6 (Proof of Lemma 2)** Define the complement of $\mathcal{E}_i$ as

$$\mathcal{E}_i^c = \left\{ \frac{1}{N} \sum_{i' \in \mathcal{U}} \tilde{Y}_{i'}(S^*) > UCB_t^*, \ \exists t \in [T_1 + 1, T] \right\} \cup \left\{ \hat{R}_{\mathcal{T}_i,S_i} + \sqrt{\frac{18 \log\left(\frac{1}{\delta}\right)}{\mathcal{T}_i}} > \frac{1}{N} \sum_{i' \in \mathcal{U}} \tilde{Y}_{i'}(S^*) \right\}.$$

Based on the union bound, we have

$$
\begin{aligned}
\mathbb{P}(\mathcal{E}_i^c) \leq &\mathbb{P}\left( \left\{ \frac{1}{N} \sum_{i' \in \mathcal{U}} \tilde{Y}_{i'}(S^*) \geq UCB_{t,S^*}, \ \exists t \in [T_1 + 1, T] \right\} \right) \\
&+ \mathbb{P}\left( \left\{ \hat{R}_{\mathcal{T}_i,S_i} + \sqrt{\frac{18 \log\left(\frac{1}{\delta}\right)}{\mathcal{T}_i}} \geq \frac{1}{N} \sum_{i' \in \mathcal{U}} \tilde{Y}_{i'}(S^*) \right\} \right) \\
\leq &\sum_{t=T_1+1}^{T} \mathbb{P}\left( \left\{ \frac{1}{N} \sum_{i' \in \mathcal{U}} \tilde{Y}_{i'}(S^*) \geq UCB_{t,S^*} \right\} \right) \\
&+ \mathbb{P}\left( \left\{ \hat{R}_{\mathcal{T}_i,S_i} + \sqrt{\frac{18 \log\left(\frac{1}{\delta}\right)}{\mathcal{T}_i}} \geq \frac{1}{N} \sum_{i' \in \mathcal{U}} \tilde{Y}_{i'}(S^*) \right\} \right).
\end{aligned}
\tag{38}
$$

*Based on Hoeffding's inequality, we can bound the first term in Eq (38) by:*

$$\sum_{t=T_1+1}^{T} \mathbb{P}\left(\left\{\frac{1}{N}\sum_{i'\in\mathcal{U}}\tilde{Y}_{i'}(S^*) \geq UCB_{t,S^*}\right\}\right) \leq (T-T_1)\delta. \tag{39}$$

*Besides, we can bound the second term in Eq (38) by:*

$$\begin{aligned}
&\mathbb{P}\left(\left\{\hat{R}_{\mathcal{T}_i,S_i} + \sqrt{\frac{18\log\left(\frac{1}{\delta}\right)}{\mathcal{T}_i}} \geq \frac{1}{N}\sum_{i'\in\mathcal{U}}\tilde{Y}_{i'}(S^*)\right\}\right) \\
=&\mathbb{P}\left(\left\{\hat{R}_{\mathcal{T}_i,S_i} - \frac{1}{N}\sum_{i'\in\mathcal{U}}\tilde{Y}_{i'}(S_i) \geq \Delta^i - \sqrt{\frac{18\log\left(\frac{1}{\delta}\right)}{\mathcal{T}_i}}\right\}\right) \\
\leq&\mathbb{P}\left(\left\{\hat{R}_{\mathcal{T}_i,S_i} - \frac{1}{N}\sum_{i'\in\mathcal{U}}\tilde{Y}_{i'}(S_i) \geq \frac{1}{2}\Delta^i\right\}\right) \\
\leq& exp\left(-\frac{\mathcal{T}_i(\Delta^i)^2}{16}\right) \\
\leq&\delta,
\end{aligned} \tag{40}$$

*where the first and last inequality is owing to the definition of $\mathcal{T}_i$, and the second inequality is owing to Hoeffding's inequality. Based on Eq (39) and Eq (40), we have $\mathbb{P}(\mathcal{E}_i) \geq 1 - (T - T_1 + 1)\delta$ for all $S_i$ satisfies $S_i \neq S^*$ and $\Delta^i > 0$. Here we finish the proof of Lemma 2.*

Now we can prove Theorem 4.

**Proof 7 (Proof of Theorem 4)** *In the Proof of Lemma 1, we shows that for all $S_i \neq S^*$, $\Delta^i > 0$, we have*

$$\mathbb{E}_\pi\left[\mathcal{N}_{S_i}^{(T_1+1,T)}\right] \leq \frac{144\log(T)}{(\Delta^i)^2} + 1. \tag{41}$$

*Define $\Lambda = 6\sqrt{\frac{|\mathcal{U}_\mathcal{E}|\log(T)}{T}}$, we can decompose the regret as*

$$\begin{aligned}
\mathcal{R}(T,\pi) \leq& \sum_{S_i\neq S^*,\Delta^i<\Lambda} \Delta^i \mathbb{E}_\pi\left[\mathcal{N}_{S_i}^{(T_1+1,T)}\right] + \sum_{S_i\neq S^*,\Delta^i\geq\Lambda} \Delta^i \mathbb{E}_\pi\left[\mathcal{N}_{S_i}^{(T_1+1,T)}\right] + \lceil\frac{T_1}{\mathcal{U}_\mathcal{E}}\rceil \sum_{S_i\neq S^*} \Delta^i \\
\leq& T\Lambda + \sum_{S_i\neq S^*,\Delta^i\geq\Lambda} \left(\frac{144\log(T)}{\Delta^i} + \Delta^i\right) + \lceil\frac{T_1}{\mathcal{U}_\mathcal{E}}\rceil \sum_{S_i\neq S^*} \Delta^i, \\
\leq& T\Lambda + \frac{144|\mathcal{U}_\mathcal{E}|\log(T)}{\Lambda} + \left(1 + \lceil\frac{T_1}{\mathcal{U}_\mathcal{E}}\rceil\right) \sum_{S_i\neq S^*} \Delta^i \\
\leq& 30\sqrt{|\mathcal{U}_\mathcal{E}|T\log(T)} + \left(1 + \lceil\frac{T_1}{\mathcal{U}_\mathcal{E}}\rceil\right) \sum_{S_i\neq S^*} \Delta^i \\
=& \tilde{O}\left(\sqrt{|\mathcal{U}_\mathcal{E}|T} + \frac{T_1}{|\mathcal{U}_\mathcal{E}|}\sum_{S_i\neq S^*} \Delta^i\right).
\end{aligned} \tag{42}$$

*Here we finish the proof of Theorem 4.*

## L    Algorithm for Adversarial Setting in Simchi-Levi and Wang [2024]

This section introduces our algorithm, EXP3-TSN, which operates in two distinct phases. In the first phase, the algorithm uniformly samples exposure super arms from the set $\mathcal{U}_\mathcal{E}$. Upon receiving reward feedback, it leverages this data to build unbiased inverse probability weighting (IPW) estimators to estimate the potential outcomes for the super arms. In the second phase, the algorithm applies the EXP3 strategy to minimize regret effectively.

**Algorithm 3** EXP3-Two Stage Network (EXP3-TSN)

---

**Input:** arm set $\mathcal{A}$, unit number $N$, exposure super arm set $\mathcal{U}_\mathcal{E}$, estimator set $\{\hat{R}_0(S) = 0\}_{S \in \mathcal{U}_\mathcal{C}}$, active super exposure arm set $\mathcal{A}_0 = \mathcal{U}_\mathcal{E}$, $T_1$, $\alpha = (e-2)(1 + 2|\mathcal{U}_\mathcal{E}|)e^2 \log(2/\delta)$, $\epsilon = \sqrt{\frac{\log(|\mathcal{U}_\mathcal{E}|)}{|\mathcal{U}_\mathcal{E}|T}}$

**for** $t = 1 : T_1$ **do**
   $\forall S \in \mathcal{U}_\mathcal{E} : \pi_t(S) = \frac{1}{|\mathcal{U}_\mathcal{E}|}$ and sample $S_t$ based on $\pi_t$
   Sample $S_t$ based on $\pi_t$, implement $\texttt{Sampling}(S_t)$
**end for**
Output $\hat{\Delta}^{(i,j)} = \frac{1}{T_1}\hat{R}_{T_1}(S_i) - \frac{1}{T_1}\hat{R}_{T_1}(S_j)$ for any $S_i, S_j \in \mathcal{U}_\mathcal{E}$, $S_i \neq S_j$
$\forall S \in \mathcal{U}_\mathcal{E}$ : set $\hat{R}_{T_1}(S) = 0$
**for** $t = T_1 + 1 : T$ **do**
   $\forall S \in \mathcal{U}_\mathcal{E}$: $\pi_t(S) = \frac{\exp(\epsilon \hat{R}_{t-1}(S))}{\sum_{S \in \mathcal{S}_t} \exp(\epsilon \hat{R}_{t-1}(S))}$
   Sample $S_t$ based on $\pi_t$, implement $\texttt{Sampling}(S_t)$
   $\forall S \in \mathcal{U}_\mathcal{E}$: set $\hat{R}_t(S) = \hat{R}_{t-1}(S) + 1 - \frac{\mathbf{1}\{S_t = S\}\left(1 - \frac{1}{N}\sum_{i \in \mathcal{U}} \tilde{r}_{i,t}(S)\right)}{\pi_t(S)}$
**end for**

---

**Unbiased estimators for exposure mapping**    We construct unbiased inverse probability weighting (IPW) estimators to estimate the potential outcome of each exposure super arm, i.e.,

$$\hat{R}_t(S) = \hat{R}_{t-1}(S) + 1 - \frac{\mathbf{1}\{S_t = S\}\left(1 - \frac{1}{N}\sum_{i \in \mathcal{U}} \tilde{r}_{i,t}(S)\right)}{\pi_t(S)}. \tag{43}$$

It is easy to verify that for all $S \in \mathcal{U}_\mathcal{E}$, for all $t \in [1, T]$:

$$\mathbb{E}\left[1 - \frac{\mathbf{1}\{S_t = S\}\left(1 - \frac{1}{N}\sum_{i \in \mathcal{U}} \tilde{r}_{i,t}(S)\right)}{\pi_t(S)} \mid \mathcal{H}_{t-1}\right] = \frac{1}{N}\sum_{i \in \mathcal{U}} \tilde{Y}_i(S) + f_t. \tag{44}$$

Using our unbiased estimator $\hat{R}_t(S)$, we can accurately estimate the ATE (which is demonstrated in Theorem 6). We define the martingale sequence as $\left(\{M_{t'}^{(i,j)}\}_{S_i \neq S_j}\right)_{t'=1}^{t}$, where $M_t^{(i,j)} = \hat{R}_t(S_i) - \hat{R}_t(S_j) - \Delta^{(i,j)}$, and it is easy to verify that $\mathbb{E}\left[M_t^{(i,j)} \mid \mathcal{H}_{t-1}\right] = 0$.

## M    Proof of Theorem 5

Theorem 5 could be equivalently separated as the following Theorem 6 and Theorem 7.

### M.1    Proof of Theorem 6

**Theorem 6 (Bounding the ATE estimation)** *Given any instance that satisfy $T \geq \mathcal{T}(T)$ and $|\mathcal{U}_\mathcal{E}| \geq 2$. Set $T \geq T_1 \geq \mathcal{T}(T_1)$. For any $S_i \neq S_j$, the ATE estimation error of the EXP3-TS can be upper bounded as follows: $\mathbb{E}\left[|\hat{\Delta}_T^{(i,j)} - \Delta^{(i,j)}|\right] = \tilde{O}\left(\sqrt{\frac{|\mathcal{U}_\mathcal{E}|}{T_1}}\right)$.*

**Proof 8 (Proof of Theorem 6)** *The proof of this lemma is based on the Bernstein Inequality. To utilize it, we first need to upper bound $|M_t^{(i,j)} - M_{t-1}^{(i,j)}|$, $\forall t \in [T_1]$. It can be expressed as:*

$$\begin{aligned}
&\left|M_t^{(i,j)} - M_{t-1}^{(i,j)}\right| \\
=&\left|\frac{\mathbf{1}\{S_t = S_i\}\left(1 - \frac{1}{N}\sum_{i' \in \mathcal{U}} \tilde{r}_{i',t}(S_i)\right)}{\pi_t(S_i)} - \frac{\mathbf{1}\{S_t = S_j\}\left(1 - \frac{1}{N}\sum_{i' \in \mathcal{U}} \tilde{r}_{i',t}(S_j)\right)}{\pi_t(S_j)} - \Delta^{(j,i)}\right| \\
\leq&\frac{1}{\pi_t(S_i)} + \frac{1}{\pi_t(S_j)} + 1 \\
=&2|\mathcal{U}_\mathcal{E}| + 1,
\end{aligned}$$

*where the first inequality is owing to the $\tilde{r}_{i,t}(\cdot) \in [0,1]$ and $\Delta^{(j,i)} \in [-1,1]$, and the second equality is due to the definition of $\pi_t(S)$ in the first phase. We also need to upper bound the variance of the martingale in the first phase, denoted as $V_t^{(i,j)}$, i.e.,*

$$V_t^{(i,j)}$$

$$= \sum_{t \in [T_1]} \mathbb{E}\left[ \left( \frac{\mathbf{1}\{S_t = S_i\}\left(1 - \frac{1}{N}\sum_{i' \in \mathcal{U}} \tilde{r}_{i',t}(S_i)\right)}{\pi_t(S_i)} - \frac{\mathbf{1}\{S_t = S_j\}\left(1 - \frac{1}{N}\sum_{i' \in \mathcal{U}} \tilde{r}_{i',t}(S_j)\right)}{\pi_t(S_j)} - \Delta^{(i,j)} \right)^2 \mid \mathcal{H}_{t-1} \right]$$

$$\leq \sum_{t \in [T_1]} \left( \frac{1}{\pi_t(S_i)} + \frac{1}{\pi_t(S_j)} \right)$$

$$\leq 2T_1 |\mathcal{U}_\mathcal{E}|.$$

*Based on this fact that $T_1 \geq \frac{(2|\mathcal{U}_\mathcal{E}|+1)^2 \log(2T_1|\mathcal{U}_\mathcal{E}|^2)}{2(e-2)|\mathcal{U}_\mathcal{E}|}$, we have*

$$\sqrt{\frac{\log(2T_1|\mathcal{U}_\mathcal{E}|^2)}{2(e-2)|\mathcal{U}_\mathcal{E}|T_1}} \leq \frac{1}{2|\mathcal{U}_\mathcal{E}|+1},$$

*which implies we can utilize the Bernstein Inequality (Lemma 5). By the Bernstein inequality, we have: $\forall t \in [T_1]$, with probability at least $1 - \frac{1}{T_1|\mathcal{U}_\mathcal{E}|^2}$, there is*

$$\left| M_t^{(i,j)} \right| \leq 2\sqrt{2(e-2)|\mathcal{U}_\mathcal{E}|T_1 \log(2T_1|\mathcal{U}_\mathcal{E}|^2)}.$$

*Dividing both sides by $T_1$, based on the definition of the martingale $M_t^{(i,j)}$ and the ATE estimator $\hat{\Delta}^{(i,j)}$, we have:*

$$\left| \Delta^{(i,j)} - \hat{\Delta}_T^{(i,j)} \right| \leq 2\sqrt{\frac{4(e-2)|\mathcal{U}_\mathcal{E}|\log\left(2T_1|\mathcal{U}_\mathcal{E}|\right)}{T_1}}. \tag{45}$$

*Define the good event as $\mathcal{E}_{T_1} := \left\{ \left| \Delta^{(i,j)} - \hat{\Delta}_T^{(i,j)} \right| \leq 2\sqrt{\frac{4(e-2)|\mathcal{U}_\mathcal{E}|\log(2T_1|\mathcal{U}_\mathcal{E}|)}{T_1}}, \forall S_i \neq S_j \right\}$. By applying the union bound, it is easy to know that*

$$\mathbb{P}\left( \mathcal{E}_{T_1} \right) \geq 1 - \frac{1}{T_1}. \tag{46}$$

*Based on the above result, for any $S_i \neq S_j$, we have*

$$\mathbb{E}\left[ \left| \hat{\Delta}_T^{(i,j)} - \Delta^{(i,j)} \right| \right] \leq \mathbb{P}(\mathcal{E}_{T_1})\mathbb{E}\left[ \left| \Delta^{(i,j)} - \hat{\Delta}_T^{(i,j)} \right| \mid \mathcal{E}_{T_1} \right] + \mathbb{P}(\mathcal{E}_{T_1}^c)\mathbb{E}\left[ \left| \Delta^{(i,j)} - \hat{\Delta}_T^{(i,j)} \right| \mid \mathcal{E}_{T_1}^c \right]$$

$$\leq 2\sqrt{\frac{4(e-2)|\mathcal{U}_\mathcal{E}|\log\left(2T_1|\mathcal{U}_\mathcal{E}|\right)}{T_1}} + \frac{1}{T_1}$$

$$= \tilde{O}\left( \sqrt{\frac{|\mathcal{U}_\mathcal{E}|}{T_1}} \right). \tag{47}$$

*Here we finish the proof of Theorem 6.*

**Theorem 7 (Regret upper bound)** *Given any instance that satisfy $T \geq \mathcal{T}(T)$ and $|\mathcal{U}_\mathcal{E}| \geq 2$. The regret of* EXP3-TS *can be upper bounded by $\mathcal{R}(T, \pi) = \tilde{O}\left( \sqrt{|\mathcal{U}_\mathcal{E}|T} + T_1 \right)$.*

## M.2 Proof of Theorem 7

**Proof 9 (Proof of Theorem 7)** *Define $R(t,j) = \frac{1}{N}\sum_{i' \in \mathcal{U}} \left( \tilde{Y}_{i'}(S_j) \right) + f_t$ as the potential outcome of exposure super arm $S_j \in \mathcal{U}_\mathcal{E}$ in round $t$. For all $S_i \in \mathcal{U}_\mathcal{E}$, we define*

$$\mathcal{R}(T, \pi, i) = \sum_{t \in [T]} R(t,i) - \mathbb{E}_\pi\left[ \frac{1}{N} \sum_{t \in [T]} \sum_{i' \in \mathcal{U}} \tilde{r}_{i',t}(S_t) \right] \tag{48}$$

*as the expected "regret" if the exposure super arm $S_i$ is the best arm. If we can upper bound $\mathcal{R}(T, \pi, i)$ for all $S_i \in \mathcal{U}_\mathcal{E}$, then we can upper bound $\mathcal{R}(T, \pi)$. Based on the unbiased property of the IPW estimator, for all $t \in \{T_1 + 1, \ldots, T\}$, we have*

$$\mathbb{E}_\pi\left[\hat{R}_T(S_i')\right] = \sum_{t=T_1+1}^{T} R(t, i') \quad \text{and}$$

$$\mathbb{E}_\pi\left[\frac{1}{N} \sum_{t \in [T]} \sum_{i' \in \mathcal{U}} \tilde{r}_{i',t}(S_t) \mid \mathcal{H}_{t-1}\right] = \sum_{t \in [T]} \sum_{S_{i'} \in \mathcal{U}_\mathcal{E}} \pi_t(S_{i'}) R(t, i') = \sum_{t \in [T]} \sum_{S_{i'} \in \mathcal{U}_\mathcal{E}} \pi_t(S_{i'}) \mathbb{E}_\pi\left[\hat{R}_t(S_{i'}) - \hat{R}_{t-1}(S_{i'}) \mid \mathcal{H}_{t-1}\right].$$

$$(49)$$

*Based on Eq (49), Eq (48) can be rewritten as*

$$
\begin{aligned}
\mathcal{R}(T, \pi, i) &\leq \mathbb{E}_\pi[\hat{R}_T(S_i)] - \mathbb{E}_\pi\left[\frac{1}{N} \sum_{t=T_1+1}^{T} \sum_{i' \in \mathcal{U}} \tilde{r}_{i',t}(S_t)\right] + T_1 \\
&= \mathbb{E}_\pi[\hat{R}_T(S_i)] - \mathbb{E}_\pi\left[\mathbb{E}_\pi\left[\frac{1}{N} \sum_{t=T_1+1}^{T} \sum_{i' \in \mathcal{U}} \tilde{r}_{i',t}(S_t) \mid \mathcal{H}_{t-1}\right]\right] + T_1 \\
&= \mathbb{E}_\pi[\hat{R}_T(S_i)] - \mathbb{E}_\pi\left[\sum_{t=T_1+1}^{T} \sum_{S_{i'} \in \mathcal{U}_\mathcal{E}} \pi_t(S_{i'}) \mathbb{E}_\pi\left[\left(\hat{R}_t(S_{i'}) - \hat{R}_{t-1}(S_{i'})\right) \mid \mathcal{H}_{t-1}\right]\right] + T_1 \\
&= \mathbb{E}_\pi\left[\hat{R}_T(S_i) - \sum_{t=T_1+1}^{T} \sum_{S_{i'} \in \mathcal{U}_\mathcal{E}} \pi_t(S_{i'})\left(\hat{R}_t(S_{i'}) - \hat{R}_{t-1}(S_{i'})\right)\right] + T_1 \\
&= \mathbb{E}_\pi\left[\hat{R}_T(S_i) - \hat{R}_T\right] + T_1,
\end{aligned}
$$

$$(50)$$

*where the first and third equality is owing to the tower rule, and the last equality is owing to we define*
$\hat{R}_T = \sum_{t=T_1+1}^{T} \sum_{S_{i'} \in \mathcal{U}_\mathcal{E}} \pi_t(S_{i'})\left(\hat{R}_t(S_{i'}) - \hat{R}_{t-1}(S_{i'})\right).$

*Define $W_T = \sum_{S_{i'} \in \mathcal{U}_\mathcal{E}} exp\big(\epsilon \hat{R}_T(S_{i'})\big)$, we have*

$$W_T = W_{T_1} \frac{W_{T_1+1}}{W_{T_1}} \cdots \frac{W_T}{W_{T-1}}$$

$$= |\mathcal{U}_\mathcal{E}| \prod_{t=T_1+1}^{T} \frac{W_t}{W_{t-1}}$$

$$= |\mathcal{U}_\mathcal{E}| \prod_{t=T_1+1}^{T} \left( \sum_{S_{i'} \in \mathcal{U}_\mathcal{E}} \frac{exp\big(\epsilon \hat{R}_{t-1}(S_{i'})\big)}{W_{t-1}} exp\Big( \epsilon\big(\hat{R}_t(S_{i'}) - \hat{R}_{t-1}(S_{i'})\big)\Big) \right)$$

$$= |\mathcal{U}_\mathcal{E}| \prod_{t=T_1+1}^{T} \left( \sum_{S_{i'} \in \mathcal{U}_\mathcal{E}} \pi_t(S_{i'}) exp\Big( \epsilon\big(\hat{R}_t(S_{i'}) - \hat{R}_{t-1}(S_{i'})\big)\Big) \right)$$

$$\le |\mathcal{U}_\mathcal{E}| \prod_{t=T_1+1}^{T} \left( 1 + \epsilon \sum_{S_{i'} \in \mathcal{U}_\mathcal{E}} \pi_t(S_{i'})\big(\hat{R}_t(S_{i'}) - \hat{R}_{t-1}(S_{i'})\big) \right.$$

$$\left. + \epsilon^2 \sum_{S_{i'} \in \mathcal{U}_\mathcal{E}} \pi_t(S_{i'})\big(\hat{R}_t(S_{i'}) - \hat{R}_{t-1}(S_{i'})\big)^2 \right) \tag{51}$$

$$\le |\mathcal{U}_\mathcal{E}| \prod_{t=T_1+1}^{T} exp\left( \epsilon \sum_{S_{i'} \in \mathcal{U}_\mathcal{E}} \pi_t(S_{i'})\big(\hat{R}_t(S_{i'}) - \hat{R}_{t-1}(S_{i'})\big) \right.$$

$$\left. + \epsilon^2 \sum_{S_{i'} \in \mathcal{U}_\mathcal{E}} \pi_t(S_{i'})\big(\hat{R}_t(S_{i'}) - \hat{R}_{t-1}(S_{i'})\big)^2 \right)$$

$$= |\mathcal{U}_\mathcal{E}| exp\left( \epsilon \hat{R}_T + \epsilon^2 \sum_{t'=T_1+1}^{T} \sum_{S_{i'} \in \mathcal{U}_\mathcal{E}} \pi_t(S_{i'})\big(\hat{R}_t(S_{i'}) - \hat{R}_{t-1}(S_{i'})\big)^2 \right),$$

*where the fourth equality is owing to the definition of $\pi_t(S)$, the first inequality is owing to $exp(x) \le 1 + x + x^2$ for all $x \le 1$ and $\hat{R}_t(S) - \hat{R}_{t-1}(S) \le 1$ for all exposure super arm S, the last inequality is owing to $1 + x \le exp(x)$ for all $x$, and the last equality is owing to the definition of $\hat{R}_T$. Based on the last term of Eq (51), we can derive*

$$\hat{R}_T(S_i) - \hat{R}_T \le \frac{\log(|\mathcal{U}_\mathcal{E}|)}{\epsilon} + \epsilon \sum_{t=T_1+1}^{T} \sum_{S_{i'} \in \mathcal{U}_\mathcal{E}} \pi_t(S_{i'})\big(\hat{R}_t(S_{i'}) - \hat{R}_{t-1}(S_{i'})\big)^2, \tag{52}$$

*and $\mathcal{R}(T, \pi, i)$ can be bounded by*

$$\mathcal{R}(T, \pi, i) \le \mathbb{E}_\pi\Big[\hat{R}_T(S_i) - \hat{R}_T\Big] + T_1$$

$$\le \frac{\log(|\mathcal{U}_\mathcal{E}|)}{\epsilon} + \mathbb{E}_\pi\left[ \epsilon \sum_{t=T_1+1}^{T} \sum_{S_{i'} \in \mathcal{U}_\mathcal{E}} \pi_t(S_{i'})\big(\hat{R}_t(S_{i'}) - \hat{R}_{t-1}(S_{i'})\big)^2 \right] + T_1. \tag{53}$$

We then try to bound $\mathbb{E}_\pi\left[\epsilon\sum_{t=T_1+1}^{T}\sum_{S_{i'}\in\mathcal{U}_\mathcal{E}}\pi_t(S_{i'})\left(\hat{R}_t(S_{i'})-\hat{R}_{t-1}(S_{i'})\right)^2\right]$, define $\tilde{R}(t,j)=1-\frac{1}{N}\sum_{i'\in\mathcal{U}}\tilde{r}_{i',t}(S_j)$, there is

$$\mathbb{E}_\pi\left[\epsilon\sum_{t=T_1+1}^{T}\sum_{S_{i'}\in\mathcal{U}_\mathcal{E}}\pi_t(S_{i'})\big(\hat{R}_t(S_{i'})-\hat{R}_{t-1}(S_{i'})\big)^2\right]$$

$$=\mathbb{E}_\pi\left[\epsilon\sum_{t=T_1+1}^{T}\sum_{S_{i'}\in\mathcal{U}_\mathcal{E}}\pi_t(S_{i'})\left(1-\frac{\mathbf{1}\{S_t=S_{i'}\}\tilde{R}(t,i')}{\pi_t(S_{i'})}\right)^2\right]$$

$$=\mathbb{E}_\pi\left[\epsilon\sum_{t=T_1+1}^{T}\sum_{S_{i'}\in\mathcal{U}_\mathcal{E}}\pi_t(S_{i'})\left(1-\frac{2\times\mathbf{1}\{S_t=S_{i'}\}\tilde{R}(t,i')}{\pi_t(S_{i'})}+\frac{\mathbf{1}\{S_t=S_{i'}\}\big(\tilde{R}(t,i')\big)^2}{\pi_t(S_{i'})^2}\right)\right]$$

$$=\mathbb{E}_\pi\left[\epsilon\sum_{t=T_1+1}^{T}\left(\frac{2}{N}\sum_{i'\in\mathcal{U}}\tilde{r}_{i',t}(S_t)-1\right)+\mathbb{E}_\pi\left[\epsilon\sum_{t=T_1+1}^{T}\sum_{S_{i'}\in\mathcal{U}_\mathcal{E}}\pi_t(S_{i'})\left(\frac{\mathbf{1}\{S_t=S_{i'}\}\big(\tilde{R}_{t,i'}\big)^2}{\pi_t(S_{i'})^2}\right)\mid\mathcal{H}_{t-1}\right]\right]$$

$$=\mathbb{E}_\pi\left[\epsilon\sum_{t=T_1+1}^{T}\left(\frac{2}{N}\sum_{i'\in\mathcal{U}}\tilde{r}_{i',t}(S_t)-1\right)+\epsilon\sum_{t=T_1+1}^{T}\sum_{S_{i'}\in\mathcal{U}_\mathcal{E}}\big(\tilde{R}_{t,i'}\big)^2\right]$$

$$\leq|\mathcal{U}_\mathcal{E}|T\epsilon.$$

Based on the definition of $\epsilon$, we can finally bound $\mathcal{R}(T,\pi,i)$ by $\sqrt{|\mathcal{U}_\mathcal{E}|T\log(|\mathcal{U}_\mathcal{E}|)}+T_1$. Here we finish the proof of Theorem 7.

# N Optimization perspective

We provide more justification upon Condition 1. Notice that we search the best arm within $\mathcal{U}_\mathcal{E}=\mathcal{U}_\mathcal{C}\cap\mathcal{U}_\mathcal{O}$, then a natural question arises that how to search elements of the intersection of these two sets? What if it is an empty set? The optimization problem is formalized as follows:

$$\sum_{i=1}^{C}c_i\mathbf{e}_i$$
$$s.t.\ \forall i\in\mathcal{U},c_i\in\mathcal{U}_s, \tag{54}$$
$$\exists A\in K^\mathcal{U},d_M\left(\big(\mathbf{S}(i,A,\mathbb{H})\big)_{i\in\mathcal{U}},\sum_{i=1}^{C}c_i\mathbf{e}_i\right)=0.$$

Here $\mathbf{e}_i$ is a binary indicator $(\mathbf{e}_i)_j=\begin{cases}1,&\text{if }j\in\mathcal{C}_i\\0,&\text{if }j\notin\mathcal{C}_i\end{cases}$. Moreover, $d_M$ denotes the Manhattan Distance.

**Searching efficiency** It would be an NP-hard problem with a high computation load without additional assumptions. However, we argue that when we select many common exposure mapping structures, the optimization problem may degenerate into a simpler case, such as an integer programming problem. Consider the mapping $\mathbf{S}(i,A,\mathbb{H}):=\boldsymbol{S}(i,A,\mathbb{H}):=\mathbf{1}\{\sum_{j\in\mathcal{U}}h_{ij}a_j>0\}$. Then Eq (54) could be transformed to

$$\sum_{i=1}^{C}\mathbf{1}\{\sum_{j\in\mathcal{U}}h_{ij}a_j>0\}\mathbf{e}_i$$
$$s.t.\ \exists A\in K^\mathcal{U},\forall p,q\text{ satisfying }\mathcal{C}^{-1}(p)=\mathcal{C}^{-1}(q), \tag{55}$$
$$\mathbf{1}\big(\sum_{j\in\mathcal{U}}h_{pj}a_j>0\big)=\mathbf{1}\big(\sum_{j\in\mathcal{U}}h_{qj}a_j>0\big).$$

To solve it, we recommend practitioners adopt the off-the-shelf optimization techniques in Mixed-Integer Nonlinear Programming Belotti et al. [2013].

**Practical issue** Another question arises: what if Condition 1 fails, even if it is easy to satisfy via adjusting legitimate exposure mapping function and clustering strategy? We formalize it as a relaxed optimization problem and claim its impact on previous modeling is negligible under mild assumptions upon interference effect:

$$\forall \{c_i\}_{i \in [C]}, \min_{A \in \mathcal{K}^{\mathcal{U}}} d_M \Big( \big( \mathbf{S}(i, A, \mathbb{H}) \big)_{i \in \mathcal{U}}, \sum_{i=1}^{C} c_i \mathbf{e}_i \Big). \tag{56}$$

Apparently, when Condition 1 is violated, then $\max_{\{c_i\}_{i \in [C]}} \min_{A \in \mathcal{K}^{\mathcal{U}}} d_M \Big( \big( \mathbf{S}(i, A, \mathbb{H}) \big)_{i \in \mathcal{U}}, \sum_{i=1}^{C} c_i \mathbf{e}_i \Big) > 0$. We recommend practitioners to collect the most *similar* exposure arm compared to the form $\sum_{i=1}^{C} c_i \mathbf{e}_i$ as above to substitute the original intersection set $\mathcal{U}_{\mathcal{E}}$. Specifically, $\forall \{c_i\}_{i \in [C]}$, we collect $\{\mathbf{S}(i, \mathbf{A}', \mathbb{H})\}_{i \in \mathcal{U}}$, where $\mathbf{A}' := \arg\min_{A \in \mathcal{K}^{\mathcal{U}}} d_M \Big( \big( \mathbf{S}(i, A, \mathbb{H}) \big)_{i \in \mathcal{U}}, \sum_{i=1}^{C} c_i \mathbf{e}_i \Big)$ as a substitute of the original corresponding cluster-wise super exposure arm. We call the substituted exposure arm set as $\mathcal{U}_{\mathcal{E}}'$.

In this sense, we recommend practitioners to re-define the arm as (modified from (3))

$$[\tilde{Y}_i^{\text{ideal}}(S_t), \tilde{r}_{i,t}^{\text{ideal}}(S_t)]^{\top} := \sum_{A \in \arg\min_{A' \in \mathcal{K}^{\mathcal{U}}} d_M \big( (\mathbf{S}(i, A', \mathbb{H}))_{i \in \mathcal{U}}, S_t \big)} [Y_i(A), r_{i,t}(A)]^{\top} \mathbb{P}(A_t = A \mid S_t).$$
$$\tag{57}$$

We denote the newly collected *similar* arm of the ideally best arm $S^*$ as $S^*_{\text{real}}$, where the former is constructed via cluster-wise exposure arm (might not be compatible with the original arm), and the latter is defined as

$$S^*_{\text{real}} := \mathbf{S}(i, A^*_{\text{real}}, \mathbb{H}), \text{ where } A^*_{\text{real}} \in \arg\min_{A' \in \mathcal{K}^{\mathcal{U}}} d_M \big( (\mathbf{S}(i, A', \mathbb{H}))_{i \in \mathcal{U}}, S^* \big). \tag{58}$$

It could be verified that under legitimate policy $\pi$ (such as uniform sampling), it leads to $\tilde{Y}_i^{\text{ideal}}(S^*) = \tilde{Y}_i(S^*_{\text{real}})$. Furthermore, the remaining part of the regret analysis could be replicated from the main text, paying attention to the new selection set $\mathcal{U}_{\mathcal{E}}'$.

## O   Auxiliary Lemmas

**Lemma 3 (Sub-Gaussian)** *A random variable $X$ is said to be **sub-Gaussian** if there exists a constant $\sigma > 0$ such that for all $m \in \mathbb{R}$, the moment generating function of $X$ satisfies:*

$$\mathbb{E}\left[ e^{mX} \right] \leq e^{\frac{\sigma^2 m^2}{2}}.$$

*The smallest such $\sigma^2$ is known as the sub-Gaussian proxy of $X$.*

**Lemma 4 (Hoeffding's Inequality)** *Let $X_1, X_2, \cdots, X_n$ i.i.d. drawn from a $\sigma$-sub-Gaussian distribution, $\overline{X} = \frac{1}{n} \sum_{i=1}^{n} X_i$ and $\mathbb{E}[X]$ be the mean, then we have*

$$\mathbb{P}\left( \overline{X} - \mathbb{E}[X] \geq a \right) \leq e^{-na^2/2\sigma^2} \quad \text{and} \quad \mathbb{P}\left( \overline{X} - \mathbb{E}[X] \leq -a \right) \leq e^{-na^2/2\sigma^2}.$$

**Lemma 5 (Bernstein's Inequality)** *Let $X_1, X_2, \ldots, X_n$ be a martingale difference sequence, where each $X_t$ satisfies $|X_t| \leq \alpha$ almost surely for a non-decreasing deterministic sequence $\alpha_1, \alpha_2, \ldots, \alpha_n$. Define $M_t := \sum_{t'=1}^{t} X_\tau$ as the cumulative sum up to time $t$, forming a martingale. Let $\overline{V}_1, \overline{V}_2, \ldots, \overline{V}_n$ be deterministic upper bounds on the variance $V_t := \sum_{t'=1}^{t} \mathbb{E}[X_\tau^2 | X_1, \ldots, X_{t'-1}]$ of the martingale $M_t$, and suppose $\overline{V}_t$ satisfies the condition*

$$\sqrt{\frac{\ln\left(\frac{2}{\delta}\right)}{(e-2)\overline{V}_t}} \leq \frac{1}{\alpha}.$$

*Then, with probability at least $1 - \delta$ for all $t$:*

$$|M_t| \leq 2\sqrt{(e-2)\overline{V}_t \ln \frac{2}{\delta}}.$$

