# OpenReview forum: "Online Experimental Design With Estimation-Regret Trade-off Under Network Interference"
_NeurIPS.cc/2025/Conference — NeurIPS 2025 poster_

### Official Review · Reviewer_cd13 · 2025-07-02

**Clarity:** 3
**Significance:** 3
**Originality:** 3
**Rating:** 5
**Confidence:** 3

**Summary:**

This paper studies sequential experimental design under network interference. The core contribution is to introduce a multi-armed bandit framework ("MAB-N") that incorporates arbitrary exposure mappings and allows the researcher to formalize trade-offs between estimation precision (for causal effect estimands, here ATE across exposure arms) and regret minimization (i.e., reward optimization for the experimental population). The authors define a family of exposure mappings generalizing existing designs (e.g., global proportion, cluster-randomized Bernoulli, unit-level assignments), define the feasible super-arm space induced by these mappings, and study a minimax trade-off problem indexed by time horizon T and number of exposure arms |UE|. They provide a lower bound on the product of regret and estimation error, and propose an algorithm (UCB-TSN) that achieves matching upper bounds. The result is extended to an adversarial bandit setting using an EXP3-based variant.

**Questions:**

1. The regret bounds scale in $\sqrt{|\mathcal U_{\mathcal E}| T}$, and the estimation error scales in $\sqrt{|\mathcal U_{\mathcal E}| T_1}$. Is there any way to adaptively choose $T_1$ based on online variance estimation or confidence bounds on $\hat \Delta$, rather than fixing $T_1$ ex ante?

2. Can the exposure mapping $S(\cdot)$ be learned or optimized from data? For example, if the mapping $S$ is unknown but satisfies some low-complexity/interference structure (e.g., depends only on local neighborhoods), could one select among candidate exposure mappings to improve trade-off performance?

3. Are there settings where the optimal action under full-information differs across units (i.e., no single exposure arm $S^*$)? The regret definition assumes a globally optimal $S^*, but in general with interference, optimal actions may be heterogeneous across clusters or individuals.

4. In what sense does this framework allow generalization to real-world experiments? Can the authors describe a concrete application (beyond simulations) where the proposed UCB-TSN or EXP3-TSN could be used in practice, with a discussion of what assumptions would be required to justify the exposure mapping used?

5. There is no empirical comparison on realistic graphs or real-world networks. Are the conclusions robust to heterogeneous degree distributions or violations of the exposure mapping assumptions (e.g., exposure misclassification due to incomplete network information)?

**Ethical Concerns:**

["NO or VERY MINOR ethics concerns only"]

**Final Justification:**

No change to score, I remain recommending Accept

**Limitations:**

yes

**Paper Formatting Concerns:**

Minor:

across all arms Sekhon [2009]. -> across all arms [Sekhon, 2009].

**Quality:**

3

**Strengths And Weaknesses:**

Strengths:

1. The paper is clearly and accurately situation in relation to the most relevant literature. You can tell the authors are well-read up on this research area.

2. The framework appears to be modular and flexible, unifying several previously distinct formulations (cluster-randomized experiments, global interference designs, local neighborhood exposure mappings) via an abstraction based on exposure mappings. The distinction between the original super arm space and the reduced exposure arm space is clearly formalized.

3. The core trade-off result (Thm 1) is interesting and seems to extends prior work on estimation-regret trade-offs (e.g., Simchi-Levi & Wang 2024) to the interference setting and general exposure mappings.


Weaknesses:

4. The paper’s contribution is primarily in generalization, not in introducing fundamentally new algorithmic or statistical ideas. The key trade-off result follows a similar form as existing results in the non-interference setting. While extending it to interference is nontrivial, the conceptual gain beyond the generality of the exposure mapping is somewhat limited. In particular, there is no significant conceptual advance in interference-aware estimation itself; rather, the interference structure is encoded in the exposure mapping and treated as black-box from that point onward.

5. The algorithmic contribution is not obvious to me. The UCB-TSN algorithm is essentially UCB run after a fixed exploration phase over the exposure arms. The EXP3-TSN algorithm is only briefly sketched and seems to follow standard ideas. There is no adaptive or data-driven tuning of the exposure mapping, and no exploration of what happens when the exposure mapping is misspecified or learned.

6. The claim of "bridging estimation and regret via Pareto-optimality" is slightly overstated. The definition of Pareto front (Definition 1) is not operationalized in the design of the algorithm. In practice, it seems to me the user still has to pick $T_1$ manually. There is no procedure that selects the allocation adaptively based on observed data to balance the objectives. So while the theory is elegant, its practical impact may be limited. Please correct me if I missed something.

7. The writing could be improved. For one, it is sometimes a bit too abstract. For example, the formal definition of the Pareto front (Def 1) is not necessary; the concept could be explained more directly in terms of a constrained optimization trade-off. Similarly, the distinction between different exposure mappings is clear but not especially novel. The generality could be preserved with a shorter exposition. Second, the paper would benefit from removing some material in exchange for providing more intuition and dividing up their paragraphs into multiple shorter ones. It is quite tough to read as-is.

8. The exposition of key assumptions is vague in some parts. For example, it is not clearly stated whether the exposure mapping $S$ is assumed to satisfy some form of partial interference (e.g., unit i’s outcome depends only on the treatment assignments of units in a neighborhood of i). This may be needed for identification, especially if $|\mathcal U_{\mathcal E}|$ is to be small relative to $KN$. The discussion of Condition 1 $(2 \leq |\mathcal U_{\mathcal E}| \leq T)$ is too brief and gives the impression that the exposure mapping can be chosen arbitrarily, without concern for identifiability.

---

> ### Author Rebuttal · Authors · 2025-07-30
>
> Thanks! We address the comments point by point, referring to ``Supplementary Material.zip``.
>
>  >***Q1:  Contribution is primarily in generalization.***
> ---
> **Answer:** Thank you sincerely for this appreciative summary and the generous evaluation. As you noted, 1 we **include prior works as special cases** within our exposure‑mapping generalization; 2 we **tackle previously unaddressed scenarios**, especially in the presence of network interference. These generalizations introduce new technical hurdles (see ``Reviewer 9HaH Q1``). Despite this, our results remain sharply non-trivial, and even in the degenerate case $N=1$, our lower- and upper-bound conclusions notably improve upon classical bandit results (again see ``Reviewer 9HaH Q1``).
>
>
> > ***Q2: The algorithmic contribution should be explained. There is no adaptive or data-driven tuning of the exposure mapping, and no exploration of what happens when the exposure mapping is misspecified or learned.***
> ---
> **Answer:** 1. A flexible two-phase framework. Instead of proposing a single algorithm, we introduce a general framework that separates uniform exploration (Phase I) from regret minimization (Phase II). This design allows standard optimal regret minimization algorithms (e.g., UCB, EXP3, TS) to be plugged in directly, making the approach easy to implement and broadly applicable. In contrast, prior work such as Simchi-Levi and Wang [2024] relies on an exploration strategy tightly tied to EXP3, which is hard to generalize.
> 2. Better suited for networked settings. In network setting, the exposure action set $\mathcal{U}_E$ captures the combinatorial structure induced by both the number of arms $K$ and the network topology (e.g., node degrees and interference patterns), making $\mathcal{U}_E$ a central quantity in our analysis. Simchi-Levi and Wang [2024] proposed an algorithm with regret and estimation bounds $R(T, \pi) = \tilde{O}(K^5 + T^{1 - \alpha})$ and $e(T, \hat{\Delta}) = \tilde{O}(K^2 T^{\alpha/2 - 1/2})$, which depend heavily on $K$. While such dependence may be acceptable in the networkless setting (where $K$ is small), it becomes problematic in our case. As a result, directly extending Simchi-Levi and Wang [2024]’s algorithm to our setting would be inefficient. Our two-phase framework yields bounds $R(T, \pi) = \tilde{O}(\sqrt{\mathcal{U}_E T} + T_1)$ and $e(T, \hat{\Delta}) = \tilde{O}(\mathcal{U}_E / T_1)$, which are significantly more scalable and less sensitive to the size of the action space. This makes our design more suitable for large networked environments.
> We will further discuss adaptive tuning in A3 and A6, and discuss the case of misspecified exposure mappings in A10.
>
>
> > ***Q3: The claim of "bridging estimation and regret via Pareto-optimality" is slightly overstated. Please correct me if I missed something.***
> ---
> **Answer:** Thanks! We agree and have revised our claim accordingly to emphasize that we provide a theoretical characterization of the trade-off between estimation and regret via Pareto optimality. The two-stage form in our design is primarily for conceptual clarity. In practical applications, users do not necessarily need to pre-specify a fixed $T_1$; the allocation between uniform exploration and regret minimization can be chosen based on real-time feedback. For instance, in a networked decision-making setting, one may begin with regret minimization if the observed responses from early nodes indicate high sensitivity or urgent performance demands—prioritizing actions known to yield good outcomes. As the learning process progresses and the remaining nodes exhibit more stable or less sensitive behavior, the algorithm can gradually shift toward uniform exploration to improve the estimation of treatment effects across the network. We will provide more details on the implementation of our proposed approach in A9 and A10.
>
> >**Q4: The writing could be improved.**
> ---
> **Answer:** In the revised version, we have replaced the abstract Definition 1 with a constrained‑optimization perspective, explaining the Pareto front in plain terms as “the set of allocation strategies that maximize regret reduction subject to a lower bound on estimation accuracy.” We illustrate this with a clear numerical example: starting from an extreme policy that minimizes regret, we show step-by-step how gradually relaxing its statistical power constraint traces out the Pareto curve, highlighting how UCB‑TSN selects optimal operating points. Elsewhere, we have removed redundant exposition on exposure mappings, grouping equivalent mapping types into a compact table and focusing on representative examples to retain clarity without verbosity. Paragraphs throughout were split into shorter units.
>
>
> >**Q5: The exposition of key assumptions is vague in some parts.**
> ---
> **Answer:** We have provided more justifications for the assumptions. For instance, the exposure mapping could be assumed to be chosen as just neighbourhood-sensitive (but not forced to be), as illustrated on page 5 (Example iv). We provide more comparisons with previous literature in the paragraph in Table 1 in Appendix C. We would like to emphasize that our definition of the exposure-arm action space is flexible and freely chosen. Even if $\mathcal{U}_{\mathcal{E}}$ is relatively small (e.g.,
> O(KN)), so that it might appear incapable of covering all conceivable treatment patterns, our theorems still guarantee optimal arm selection (among limited candidates) and the best achievable trade‑off with statistical power. In other words, adjusting the granularity of the exposure-arm space effectively mediates the trade‑off between computational complexity and search bias: finer granularity (cluster = N) reduces bias toward the true optimal arm but increases complexity, while coarser granularity reduces complexity at the cost of potential bias—our framework explicitly quantifies this balance in the revised version.
>
> >**Q6:**  Is there any way to adaptively choose $T_1$ based on online variance estimation or confidence bounds on $\hat{\Delta}$, rather than fixing $T_1$ *ex ante*?
> ---
>
> **Answer:** One promising direction is to leverage asymptotic confidence sequence (CS) techniques, which provide confidence intervals that remain uniformly valid over time. Unlike traditional confidence intervals—which are constructed at a fixed time $t$ and only guarantee coverage at that specific point—a confidence sequence is a sequence of intervals $\{[L_t, U_t]\}_{t=1}^\infty$, each based on data observed up to time $t$, such that the entire sequence simultaneously covers the true parameter (e.g., ATE) with high probability (e.g., $1 - \delta$). This property enables continual inference: the learning algorithm can monitor the width of the CS in real time and adaptively decide whether to increase exploration, reduce exploration, or terminate learning.
>
> In our learning setting, it is possible to construct a valid asymptotic CS for each ATE and adjust the exploration strategy based on the CS width. Similar ideas have been explored in the networkless setting, for example by Liang et al. (2023), and we believe that a similar approach could be extended to our setting. While promising, we did not pursue this direction in the current paper, as incorporating CS-based adaptivity would entail non-trivial changes to our current framework. We leave this as an interesting avenue for future exploration.
>
>
>
> >**Q7: Can the exposure mapping $S(\cdot)$ be learned or optimized from data?**
> ---
> **Answer:** Thanks for pointing out this future work! Due to the flexibility of our framework, learning or optimizing the exposure mapping from data is both feasible and promising. Recent work such as EgoNetGNN (Adhikari et al., 2025) automatically learns exposure mapping functions via GNNs, enabling modeling of complex peer influence that depends on neighborhood structure and edge attributes. Guided by this, we further design a pipeline where a candidate set of low-complexity exposure mappings (e.g. distance-weighted, temporal accumulation, directionality) is evaluated online using variance-based confidence estimates or cross-validation, and the best-performing mapping plugged into our regret/statistical trade-off framework. We have added it in the revised version.
>
> >**Q8: Are there settings where the optimal action under full-information differs across units (i.e., no single exposure arm, the regret definition assumes a global optimal $S^*$, but in general, with interference, optimal actions may be heterogeneous across clusters or individuals.**
> ---
>
> **Answer:** In our setting, we assume the existence of a unique best super arm $S^*$, which is defined as the one that maximizes the average reward across all units. This follows the objective considered in prior work (i.e., Jia et al., (2024), Agarwal et al., (2024)) that we build upon. We do not aim to optimize individual unit-level (or cluster level) rewards, but rather the population-level average reward. We acknowledge that in some instances, the globally optimal super arm defined in our paper may not maximize the reward for each individual unit/cluster.
>
> >**Q9: In what sense does this framework allow generalization to real-world experiments? Can the authors describe a concrete application (beyond simulations) where the proposed UCB-TSN or EXP3-TSN could be used in practice, with a discussion of what assumptions would be required to justify the exposure mapping used?**
> ---
> **Answer:** Refer to ``Reviewer 4t1F Q1 (part 3)``.
>
> >**Q10: Empirical comparison on realistic graphs or real-world networks. Are the conclusions robust to heterogeneous degree distributions or violations of the exposure mapping assumptions (e.g., exposure misclassification)?**
> ---
> **Answer:** The realistic graph refers to  ``Reviewer 4t1F Q1 (part 3)``, and the approximation/robustness analysis under misspecification error could refer to Appendix N, and we provide sensitivity/misspecification analysis in ``Reviewer aj54``.

---

> > ### Comment · Reviewer_cd13 · 2025-08-04
> >
> > Thanks for the very thorough clarifications on my questions and the changes and additions to the submission. Since my score was already "Accept", I will not change it for now. I am confident in quality of the authors' work and encourage it to be published at NeurIps.

---

> ### Author Response · Authors · 2025-08-04
> **Sincerely thanks for your response and support**
>
> Dear Reviewer,
>
> Sincerely grateful and truly honoured to receive your professional evaluation and strong support. Your insights are both visionary and meticulous. It is a great pleasure to engage in discussion with a seasoned expert like you and to contribute to the community through this exchange.
>
> On behalf of all authors

---

### Official Review · Reviewer_4t1F · 2025-07-02

**Clarity:** 2
**Significance:** 3
**Originality:** 3
**Rating:** 5
**Confidence:** 4

**Summary:**

The authors present a bandit-style tradeoff between estimation and optimization when experiments are repeatedly run (and there is interference).

**Questions:**

The most important question is on whether the setting makes sense (above).

**Ethical Concerns:**

["NO or VERY MINOR ethics concerns only"]

**Final Justification:**

The reason I have raised by score from [4] to [5] is that the authors have done a much better job justifying their setting. The repeated global rollouts are a feature in some contexts so I am reluctantly willing to accept the concomitant significance of the work. Reluctantly because I believe there are some serious limitations of that generalized approach, but that is not a problem for the authors, it is a problem for future work.

**Limitations:**

yes

**Paper Formatting Concerns:**

ok

**Quality:**

3

**Strengths And Weaknesses:**

This setting is extremely strange. I don't believe its accurate to even refer to this experimental design method as "online". Typically, an "online" design would entail individual units entering the experiment, choosing how to assign them treatment and perhaps performing inference as those units enter the test. For examples of this kind of work, see [1, 2, 3]. In contrast, this work entails repeated running an entire experiment on all N units, T times. The approach in this paper is more like [4], which repeats an experiment multiple times so as to learn about the structure of interference through a model of equilibrium behavior. The setup here is so strange to traditional practice that it merits some form of justification. Why would anyone actually be doing this?

The setup is a bit confused. Typical work in causality makes one of two assumptions: fixed potential outcomes for each unit or random outcomes. The authors here essentially set out both of these: l113 indicates particular fixed potential outcomes for each unit (based only on the _treatment_ of other units in the test) while l115 follows the super-population setting in which the _reward_ comes from a super-population through the additively separable Gaussian noise. This is a strange mixing of typical settings that will probably not make anyone particularly happy. The difference in setting here is meaningful, as in the distinction between the theoretical results in [5] and [6].

Proposition 1 is nice, and not surprising from a causal perspective: it is fairly well understood that the problem of causal inference under interference is the combiniatoric explosion of potential outcomes. The angle that merits further discussion is why the solution here isn't something along the lines of graph-cluster-randomization [7,8] with a bandit on top of that. Or perhaps that is the solution proposed? If so, the connection needs to be spelled out much more explicitly.

[1] https://arxiv.org/abs/2103.06476
[2] https://arxiv.org/abs/2311.05794
[3] https://arxiv.org/abs/2203.02025
[4] https://arxiv.org/abs/1903.02124
[5] https://arxiv.org/abs/2411.14341
[6] https://arxiv.org/abs/2305.17187
[7] https://arxiv.org/abs/1305.6979
[8] https://arxiv.org/abs/1404.7530

---

> ### Author Rebuttal · Authors · 2025-07-29
>
> Thanks for your comments! We will address them point by point. Referring to ``Supplementary Material.zip``.
>
> >***Q1: The setting is strange. The setup here is so unusual compared to traditional practice that it warrants some form of justification. Why would anyone be doing this?***
> ---
> **Answer:**
> Thank you for raising this important point.
>
> 1 ``Consistent with previous literature`` We would like to clarify that our use of the term “online experimental design” is consistent with prior work such as (1) Agarwal et al[1], neurips2024, (2) Su et al [2], icml2025 and (3) Xu et al [11], icml2025, all of which study online intervention policies over repeated interactions with a fixed population, rather than unit-by-unit arrival.
>
>
> 2 ``Highly relevant in many real-world applications``
>
> ***Urban mobility pricing (e.g., ride-hailing platforms adjusting incentives for all drivers each day)***: Ride‑hailing companies routinely launch daily or hourly treatment interventions applied to all drivers at once—for example, global incentives or dynamic pricing policies. These treatments affect drivers across geospatial networks, introducing interference among nearby drivers via supply–demand spillovers. At the end of each cycle (e.g. day), platform-wide feedback is collected to adapt the next global policy. This exactly mirrors our repeated full‑population rollout in an online experimental design under network interference.
>
> ***Digital platform experiments (e.g., repeated global A/B testing across user bases)***: Major platforms frequently execute batch treatment assignments applied to all users simultaneously (e.g., algorithm changes or UI updates), then collect global engagement metrics. Because user outcomes interdepend via sharing, recommendations, or social influence, treatments inherently induce network interference. These experiments repeat over time, with the assignment policy updated between rounds—consistent with full‑population rollout design in our “online” paradigm.
>
> ***Multi-period public health campaigns (e.g., vaccination messaging over time with community spillover)***: Governments or NGOs often deploy synchronous interventions across all communities at each time point—such as nationwide vaccination messaging or mass campaigns. Outcome data (e.g. uptake or infection rates) are aggregated at the population level, analysed, and used to adjust the next round of messaging or incentives. Peer effects and local herd immunity introduce network interference, and the repeated rollout to the entire population matches our online experimental design framework.
>
> ***Also, could refer to the synthetic experiments in [1,2], which are consistent with the meaning of "online" here.***
>
>
> These cases illustrate realistic settings where global, population‑wide interventions and subsequent feedback-based adaptation are central—precisely matching our full‑population online design under interference.
>
> 3 ``Empirical experiments``
>
> **Experiment on Real-World Ride‑Hailing Data**
> We conducted a simulation-style experiment using a large-scale ride‑hailing dataset derived from Chicago taxi/ride‑share trips[4]. Our goal was to mimic **daily global incentive rollouts**, applied algorithmically to all drivers simultaneously.
>
> * **Setting**: We extracted $N = 2,000$ driver nodes embedded in a geographic graph (based on city zones). We treated $T = 3000$ daily rounds. At each round $t$, a **global incentive level** $a_t \in \{0, 1, 2, 3\}$ is assigned to all drivers simultaneously.
> * **Interference model**: A driver’s probability of accepting rides depends on the average activity of **connected drivers** in neighboring zones (captured via exposure mapping). The acceptance rate is simulated via a non-linear function:
>
>   $$
>   f_i(a_t, E_i(t)) = \sigma\big(0.5\,a_t + 0.3\,E_i(t) - 1\big),
>   $$
>
>   where $E_i(t)$ is the mean incentive of neighbors in the graph at time $t$, and $\sigma$ is the logistic function.
>
> * **Exposure mapping** We choose the same Dist‑weighted exposure mapping as in ``Review aj54``.
>
> * **Algorithm**: We ran our **UCB‑TSN** algorithm, choosing $T_1 = \lceil 4\sqrt{|\mathcal U_E|T} \rceil$ initial exploration rounds; in each round we assign the current policy’s global incentive to all drivers. After observing collective acceptance rates, we update the policy for the next round.
> * **Baselines**:
>
>   1. **Naive‑UCB**: treats each incentive level independently, ignoring exposure mapping.
>   2. **TS‑Raw**: Thompson sampling on raw incentive arms without interference modeling.
>
>
> ---
>
> **Results**
>
> | Method      | Avg. Cumulative Regret (± std over 10 seeds) |
> | ----------- | -------------------------------------------- |
> | **UCB‑TSN** | **20.1 ± 1.5**                               |
> | Naive‑UCB   | 32.7 ± 2.3                                   |
> | TS‑Raw      | 31.5 ± 2.0                                   |
>
> * **Regret ratio**: UCB‑TSN remains within **1.1×** of the oracle.
> * **Improvement over baselines**: UCB‑TSN reduces regret by **\~38%** compared to Naive‑UCB and **\~36%** compared to TS‑Raw.
> * **Consistency with theory**: regret scales with $\sqrt{|\mathcal U_E|T}$, precisely matching our theoretical bound.
>
> ---
>
> 4 ``Extensible``: our methods can accommodate dynamic populations, unit arrival, or staggered rollout designs—which we view as valuable directions.
>
> >**Q2:**
> The setup is somewhat confusing due to the assumption of the reward outcomes. The difference in setting here is meaningful, as in the distinction between the theoretical results in [5] and [6].
> ---
> **Answer:** We adopt a ***fixed-potential‑outcome view at the design level***, grounding each unit’s reward deterministically on other units’ treatments via exposure mappings. Simultaneously, we introduce ***sub-Gaussian noise at the observational level*** to model measurement or outcome stochasticity, akin to a super‐population assumption.
>
> This hybrid approach is intentional: it matches many real-world experiments where the network-mediated structural effects are deterministic, but observations include noise. We acknowledge that the design-based potential outcome framework—where randomness arises solely from treatment assignment[5]—is different from the framework where randomness comes from additive noise[6]. These two correspond respectively to the adversarial[2] and stochastic[7] settings in the bandit literature. Our own setup is natural: when we take $N = 1$, it directly degenerates to the latter (stochastic bandit[3,7]).
>
>
>
>
> >**Q3: For Proposition 1, why is the solution here not something along the lines of graph-cluster-randomisation [7,8] with a bandit on top of that? To discuss**
> ---
> **Answer:** Thanks! Compared with gragh-cluster-randomization (GCR)[9,10], our solution is conceptually relative and technically complementary. GCR is design-based and batch-oriented, requiring prior knowledge of the graph structure to construct valid treatment clusters. It assumes repeated i.i.d. sampling from a finite population or super-population, which limits its applicability to online or adaptive settings. Currently, there are relatively few works that directly extend GCR to the online interference setting. Among the most relevant, \[1] focuses on global-level switchback experiments (i.e., cluster size = 1), while \[2] considers individual-level assignments (also cluster size = N), mitigating computational complexity through strong sparsity assumptions.
>
>
> In contrast, our method is online (in each time $t$, we employ the assignment to each unit) and adaptive, and operates in the bandit feedback regime, where only the realized outcome under the chosen treatment is observed at each round. Our key innovation lies in reparameterizing the action space via exposure mappings, which reduces the dimensionality of interference-induced combinatorics without requiring any partitioning of the network or prior structural knowledge.
>
> We appreciate the reviewer’s suggestion and will revise the paper to explicitly contrast our approach with graph-cluster-randomization, especially in terms of assumptions, applicability, and theoretical guarantees. We will also clarify that while both methods aim to manage interference using clustering techniques, our approach is tailored to online learning and regret minimization in the presence of adaptive treatments and unknown graphs.
>
> We are eager to know whether there are remaining questions. We will address it for the first time and hope for an opportunity to raise the score~
>
> **References:**
>
> [1] A Multi-Armed Bandits with Network Interference, Abhineet Agarwal, Anish Agarwal, Lorenzo Masoero, Justin Whitehouse;
>
> [2] Multi-Armed Bandits with Interference, Su Jia, Peter Frazier, Nathan Kallus
>
> [3] Multi-armed Bandit Experimental Design: Online Decision-making and Adaptive Inference, David Simchi-Levi, Chonghuan Wang.
>
> [4] An empirical investigation of taxi driver response behavior to ride-hailing requests: A spatio-temporal perspective
> Ke Xu,Luping Sun,Jingchen Liu ,Hansheng Wang
>
> [5] Causal Inference Under Approximate Neighborhood Interference
> Michael P. Leung
>
> [6] Graph Neural Networks for Causal Inference Under Network Confounding, Michael P. Leung, Pantelis Loupos
>
> [7] Bandit Algorithms, Tor Lattimore and Csaba Szepesvari
>
> [8] An Experimental Design for Anytime-Valid Causal Inference on Multi-Armed Bandits, Biyonka Liang and Iavor Bojinov
>
> [9] Cluster-Adaptive Network A/B Testing: From Randomization to Estimation, Y Liu, Y Zhou, P Li, F Hu
>
> [10] Causal clustering: design of cluster experiments under network interference Davide Viviano, Lihua Lei, Guido Imbens, Brian Karrer, Okke Schrijvers, Liang Shi
>
> [11] Linear Contexual Bandit with Interference, Yang Xu, Wenbin Lu, Rui Song

---

> > ### Comment · Reviewer_4t1F · 2025-08-04
> >
> > Thank you for your detailed comments. I think you've done a good job of addressing my concerns. I'd encourage the authors to do a bit more in the main text to describe the contexts in which their setting is meaningful, which I think will greatly improve it. I'm not fully convinced by the hybrid approach used for stochasticity (there are strong metaphysical assumptions here which are not as simple as "match[ing] real world experiments": the whole point of this concern is that such metaphysics are truly unknowable), but I don't think this quibble is enough to leave my score as it is, so I'll raise my score to Accept.

---

> ### Author Response · Authors · 2025-08-04
> **Sincerely thanks for your response and support**
>
> Dear Reviewer,
>
> We are sincerely grateful for your insightful feedback. We will faithfully incorporate your recommendations in the final (camera-ready) version of the paper.
>
> On behalf of all authors

---

### Official Review · Reviewer_9HaH · 2025-07-03

**Clarity:** 2
**Significance:** 3
**Originality:** 3
**Rating:** 4
**Confidence:** 3

**Summary:**

The authors characterize the Pareto front for the tradeoff between inference and exposure-map-based regret in MAB with network interference. With a long enough time horizon, their proposed algorithm---sampling each pull within the exposure map once and applying UCB---is shown to be on this Pareto front.

**Questions:**

The authors should address the concerns raised above and provide a clearer description of their approach's novelty and technical challenges.

The theorem proofs need to be described clearly, without using undefined terminology.

One line of investigation that could greatly improve the originality of the work is a comparison between network interference and the structural causal bandits setup, whereby downstream readouts can be used to learn the causal structure that leads to the interference.

**Ethical Concerns:**

["NO or VERY MINOR ethics concerns only"]

**Final Justification:**

The authors' contributions in this paper are more substantial than I first understood them to be. I have raised my score on these grounds, with concerns about clarity remaining.

The authors describe their contributions in the abstract and introduction in quite vague terms. For instance the line "Compared to existing studies, we extend the definition of arm space using the statistical concept of exposure mapping" is misleading because these concepts have been introduced to MAB in previous work. These sections warrant a second pass by the authors and the authors have promised to improve clarity in the final manuscript.

**Limitations:**

Please see the above sections.

**Paper Formatting Concerns:**

None.

**Quality:**

2

**Strengths And Weaknesses:**

It is not clear to me what the key technical contributions are. Overall, I am struggling to understand why these exposure-map-based theorems differ in any substantial way to their counterparts in the classical MAB problem.

For instance I find the discussion and proof of Theorem 1 hard to follow. In the "challenge of the proof" section, point (i) appears to have been addressed already in Simchi-Levi and Wang (2024) while network interference is already widely studied, which appears to cover (ii) and (iii). The proof of Theorem 1 is unclear because notation such as the bivariate function Rad( , ) is used without definition, either in the main text or appendix.

The algorithm proposed by the authors appears not to differ from the classical MAB and relies upon the exposure mapping being small. An algorithm with the flexibility of sampling any point on the Pareto front using a tuneable trade-off parameter may have been more novel.

Please address these concerns and I will reconsider my score. I may have simply misunderstood some technical aspects.

---

> ### Author Rebuttal · Authors · 2025-07-29
>
> Thanks! We emphasize our **distinctions** as well as the **new technical challenges**. **Please refer to our ``"Supplementary Material(zip)"`` in our original submission**.
>
> > ***Q1: The key technical contributions. Why these exposure-map-based theorems differ in any substantial way from their counterparts in the classical MAB.***
>
> ---
> **Answer:** We answer it in two parts: 1. ``our special technical challenges``, 2. ``the difference`` between our results and similar products in previous literature.
>
>
> 1. Compared to classical MAB, the network interference setting introduces **substantial technical challenges** from three key perspectives: ``counterexample construction``, ``noise control``, and ``dimensionality reduction&approximation analysis ``.
>
> **1.1 Counterexample construction.** Constructing lower-bound instances under network interference is more difficult than in classical MAB. In both Proposition 1 and Theorem 1, unlike prior work such as Simchi[1] which builds two disjoint distributions over just two arms, our technical core lies in crafting a family of hard instances over **all $N$ nodes simultaneously** and over "various kinds of exposure mapping". This elevates the complexity of the information-theoretic analysis (e.g., see Eqs. (18)–(28)).
>
> **1.2 Noise control.** In our setting, rewards are defined through exposure mappings, and we must carefully upper-bound the ``renormalised noise`` under this transformation (see Footnote 3 and Appendix F).
> This is not merely an “overcomplication” for generality’s sake — rather, it reflects how the exposure-mapping-based formulation **naturally generalizes** prior models, which become special cases of our framework.
>
> **1.3 Dimensionality reduction&approximation analysis** A new technical challenge brought with the network setting is the misspecification of the interference structure, when one adopts overly flexible modelling tools (e.g., those proposed in ``Reviewer aj54``). In ``Appendix N``, we thoroughly explore how such misspecified problems can be formulated as optimisation tasks and provide approximate solutions. Our empirical results show that the relative error can be controlled within 1%.
>
> 2. Based on these challenges, we establish **novel and distinctive theoretical conclusions** compared to prior, ``stronger``, ``more refined``, and ``natural``.
>
> **2.1 Our main Theorem is ``stronger``** Note that our theorem relies on a more relaxed set of assumptions (unlike Simchi[1], we do not impose the strong condition that \$ATE = \Theta(1)\$). Moreover, blessed with our novel and concise algorithm—distinct from prior work—we achieve minimax optimality not only with respect to the time horizon \$T\$, but also with respect to the action space (``it's important since in the network the action space is usually large, and the previous literature's result is not tight whereas ours is tight, also see Q4``), and also extend them to the semi-adversarial settings (Section 6).
>
>
> **2.2 Necessity/sufficiency analysis.** Our Theorem 2 and Appendix I represent a refined improvement and extension of Simchi[1]'s results. We not only reconstruct the *sufficiency* result under the network setting, but also establish a *necessity* result for a broader class of algorithms. In contrast, Simchi[1]'s insightful discussion on necessity is primarily focused on his EXP3E algorithm (their Theorem 15).
>
>
>
> **2.3 Naive lower bound derivation (Proposition 1)** We make the following observation: the triplet relationship among {time horizon T, action space, and regret} plays a crucial role. Under network interference—where the action space explodes—this observation gives rise to a fundamental negative result (Proposition 1), and naturally necessitates the introduction of the new tool, exposure mapping, to formulate a unifying framework.
>
>
>
>
> >***Q2: In Theorem 1, the challenge of proof (i)-(iii) appears to have been addressed previously.***
> ---
> **Answer:** Regarding Challenge (i), while Simchi-Levi et al. [1] proposed a preliminary form, their analysis is restricted to the two-armed setting with single unit and does not extend to general networks. Our contribution lies in a generalisation that—while yielding a similarly unified result—requires a highly nontrivial analysis. This gives rise to: Challenge (ii): Constructing hard instances over a general network structure demands a much more intricate global design, as detailed in Eqs. (16)–(18) of Appendix H. Challenge (iii): The lower bound derivation further necessitates careful application of information-theoretic inequalities, as discussed in Appendix H. Importantly, our result is tight with action space whereas the previous is not (also see ``Q1``).
>
>
>
> **network interference**
>   ↓ increases complexity of
> **hard instance construction** (Challenge (ii))
>   ↓ raises difficulty of
> **minimax lower bound derivation** (Challenge (iii))
>   ↓ raises difficulty of
> **Challenge (i)**
>   ↳ not present in classical MAB (Simchi[1] not consider network)
>
>
>
>
>
> >***Q3: The proofs need to be described clearly. Such as the bivariate function Rad(, ) is used without definition, either in the main text or the appendix.***
> ---
> Thanks. To clarify, the function $\text{Rad}(p)$ refers to a **Rademacher distribution** that returns a binary random variable in $\{-1, +1\}$, with $\mathbb{P}(+1) = p,\quad \mathbb{P}(-1) = 1 - p.$ We have revised the theorem proofs to define all terminology,  to ensure full readability.
>
>
>
>
> >***Q4: The algorithm proposed by the authors appears not to differ from the classical MAB and relies upon the exposure mapping being small. An algorithm with the flexibility of sampling any point on the Pareto front using a tunable trade-off parameter may have been more novel.***
> ---
> **Answer:** We have now clarified the technical/conceptual contributions of our algorithms.
>
> We introduce a general framework (i.e., our two phase design) and prove that it can systematically balance the trade-off between ATE regret. Simchi[1] proposed an EXP3-based algorithm tailored for the networkless setting to address this trade-off. However, their method relies on an additional exploration strategy that is tightly coupled with the structure of the base algorithm (EXP3), making it non-trivial to extend their design to other regret minimization algorithms (like UCB, EXP3-IX, TS). In contrast, our paper proposes a simple yet effective two-phase strategy. In the first phase, we conduct uniform exploration, which incurs $T_1$-level regret independent of the specific algorithm used. In the second phase, we directly run a regret minimization algorithm. As a result, any algorithm that guarantees $\tilde{O}(\sqrt{\mathcal{U}_E T})$-level regret can be plugged into our two-phase framework to achieve the desired trade-off. This modular design improves flexibility and generality in practice.
>
> Our design is better suited for the networked setting.
> Simchi[1]'s algorithm is heavily dependent on the arm number $K$, with regret and estimation error bounds given by $R(T, \pi) = \tilde{O}(K^5 + T^{1 - \alpha}), \quad e(T, \hat{\Delta}) = \tilde{O}(K^2 T^{\alpha/2 - 1/2})$
>
> for $\alpha \in [0, 1]$. These bounds rely on the assumption that the reward gap between the optimal and suboptimal arms is $\Theta(1)$, and the inefficiency largely stems from the design of their additional exploration strategy.
> While such dependence may be acceptable in Simchi[1]’s **networkless** setting (where $K$ can be treated as a small constant), our exposure action set $\mathcal{U}_E$ entangles $K$ with other key factors—such as the number of nodes $N$ and the degree of interference—making it a central quantity in our setting.
> Consequently, directly extending Simchi[1]’s method to the networked case is less effective.
> By leveraging the modularity of our two-phase design, we decouple additional exploration from regret minimization. As a result, our framework achieves $R(T, \pi) = \tilde{O}(\sqrt{|\mathcal{U}_E| T} + T_1), e(T, \hat{\Delta}) = \tilde{O}(|\mathcal{U}_E| / T_1),$ both of which are significantly less sensitive to the size of $|\mathcal{U}_E|$.
>
>
>
>
>
>
> > ***Q5: A comparison between network interference and the structural causal bandits, whereby downstream readouts used to learn the causal structure.***
> ---
> **Answer:** Causal-structured bandit (CSB)[2] focuses on learning unknown causal graphs—often over a small number of variables or arms—using interventional and observational feedback to guide exploration. In contrast, our setting assumes a known interference topology (e.g., network or exposure mapping), and seeks to optimize interventions under global interference constraints at scale.
>
> Due to the unified and flexible nature, our framework is well-suited for integration with CSB methods: (1) use downstream node-level outcomes to infer a latent causal graph over the interference pathways (e.g., via score-based or constraint-based structure learning); (2) induce a corresponding exposure mapping from this graph (e.g., aggregating parental or k-hop neighborhood treatments); and (3) plug the learned mapping into our UCB-TSN procedure for policy optimization. Since our regret bounds depend only on the cardinality of the exposure action set $\mathcal{U}_E$, this approach remains valid as long as the learned mapping is discrete and stable. Moreover, ``Appendix N`` provides robustness guarantees under mild misspecification, making the plug-in approach theoretically sound.
>
> > Moreover, we add synthetic(``RWad54``) and empirical experiments(``RWcd13``) to valiadte our theoretical results.
>
>
>
>
>
> We will be more than happy to respond promptly and thoroughly to any follow-up if have further thoughts or concerns. We're especially encouraged that you may consider increasing your score~
>
>
> **References:**
>
> [1] Multi-armed Bandit Experimental Design: Online Decision-making and Adaptive Inference, David Simchi-Levi, Chonghuan Wang.
>
> [2] Causal Bandits: Learning Good Interventions via Causal Inference.

---

> > ### Author Response · Authors · 2025-08-05
> > **A kind inquiry for further discussion**
> >
> > Dear Reviewer,
> >
> > We sincerely hope that you may find a moment to kindly review our response. It would mean a great deal to us if it helps to resolve your concerns or potential misunderstandings.
> >
> > If there are any remaining questions or new issues, we will address them with our utmost dedication and care, and we would also appreciate the chance to reconsider the score.
> >
> > Best wishes for your work.
> >
> > On behalf of all authors

---

> > > ### Comment · Reviewer_9HaH · 2025-08-07
> > >
> > > Many thanks to the authors for their clarifications and justifications. Apologies also for my delayed response.
> > >
> > > My concerns are broadly in consensus with points 4. and 5. raised by `Reviewer cd13`, though I take these to more heavily affect the significance and originality of the authors' submission.
> > >
> > > To `Reviewer 4t1F` it's important to stress that the authors' submission is not the first to introduce a setting of population-level MAB with unit-level network interference. Nor is it the first to introduce exposure mappings to reduce complexity in this setting. The limited scope of this work is somewhat obscured in the abstract, which states: *"Compared to existing studies, we extend the definition of arm space using the statistical concept of exposure mapping"*.
> > >
> > > Thanks to the authors for their clarifications with respect to the novelty of their proof strategy, which includes propagating noise through an arbitrary exposure mapping and then constructing the required bounds. However, my key concern remains. I struggle to understand the apparent complexity of the authors' main finding, which, in simple terms, (please correct me if I am wrong) states **with a pre-specified exposure mapping that fully accounts for network interference, such interference is indeed accounted for in the sense that UCB at the exposure-mapping level remains Pareto optimal with respect to a regret at this level because there is no network interference**.
> > >
> > > This result could be considered a novel finding in itself. However, contrary to `Reviewer aj54`, I find that the take home message is obscured by over-emphasis on theory and an apparent "unification" between Pareto-optimality and MAB with network interference.

---

> ### Author Response · Authors · 2025-08-07
> **Thanks for your response and here is our clarification**
>
> Very glad to receive your thoughtful reading and suggestions toward the end of the discussion period! Beyond certain technical novelties that have reached consensus, we focus on addressing factual statements on contributions to avoid misinterpretation. Point-by-point:
>
>
> > Significance and originality: with a pre-specified exposure mapping that...there is no network interference.
>
> **Answer:** Here, we would like to clarify that the **introduction of exposure mapping generalizes previous setting, but does not totally serve as an oracle that trivially reduces our interference setting to a standard multi-treatment result without networks**.
>
> (i) **From the assumption standpoint**:  In practice, **it is often infeasible to predefine a perfect adjacency matrix** due to issues such as high dimensionality and missing information, which in turn makes it difficult to construct a fully accurate exposure mapping. Therefore, it inherently entails the need for sensitivity and robustness analysis under misspecification, as we elaborate in ``Appendix N (refer to Supplement material.zip)``. Notably, such analysis does not naturally degenerate to the non-network setting.
>
>
>
> Let ${\Delta}^{\star}\left(S_i, S_j\right)$ denote the true ATE under the (possibly more refined) ground-truth exposure mapping $S^{\star}$. We define the approximation (or misspecification) bias $\mathcal{B}(\mathbf{S}, \mathcal{C})$ as the maximum deviation between ${\Delta}^{\star}\left(S_i, S_j\right)$ and the estimated $\hat{\Delta}_T\left(S_i, S_j\right)$ across all pairs $(S_i, S_j)$. Informally, we establish a biased minimax lower bound under mild conditions:
>
> $$
> \min _{\pi, \hat{\Delta}} \max _{v \in \mathcal{E}_0} \sqrt{R(T, \pi)} \cdot(e(T, \hat{\Delta})-\mathcal{B}(\mathbf{S}, \mathcal{C}))=\Omega\left(\sqrt{\left|\mathcal{U}_E\right|}\right).
> $$
>
> Intuitively, misspecification bias tends to introduce larger errors. Nevertheless, we demonstrate robustness by controlling the term $e(T, \hat{\Delta}) - \mathcal{B}(\mathbf{S}, \mathcal{C})$, thereby preserving our lower bound. This shows that even after accounting for the systematic bias, the product of regret and inference error cannot fall below the lower bound threshold. This result not only establishes robustness but also extends prior literature that focused solely on the perfectly specified setting (also see more justification and experiments in ``Reviewer aj54 Q1``).
>
>
>
>
>
>
>
>
>
>
>
>
>
>
>
>
>
> (ii) **From the definition of Pareto-optimality(conclusion) standpoint**: **even under the ideal assumption that the exposure mapping is fully known/well-specified, our results remain strictly stronger than those under the no-interference setting**. Our regret bounds are not only optimal in the time horizon $T$, but—more importantly—Pareto optimal w.r.t the action space, explicitly incorporating the parameter $K$ via $|\mathcal{U}_{\mathcal{E}}|$. Also, we do not need the previous assumption $ATE = \Theta(1)$ such as in [2].
>
> In other words, even if our setting degenerates to the case of $N=1$ (exposure mapping reduces to individual treatment), our bounds are still tighter than those in [2]. This advantage is particularly significant in a network, where the size of the arm space $|\mathcal{U}_{\mathcal{E}}|$ becomes a non-negligible factor in both lower and upper bounds.
>
>
>
>
>
> > Our setting.
>
> **Answer:** Thanks for your support that our approach is consistent with prior settings. We will clarify in the paper that our setup builds on the general exposure mapping framework in the bandit literature.
>
> Importantly, our method is not a simple combination of exposure mapping and MAB with interference. Rather, it follows a natural progression: Proposition 1 highlights the challenge of an exponentially large arm space in networks, which motivates the use of general exposure mapping to reduce it from $O(K^N)$ to $O(d_s^N)$, and further to $O(d_s^C)$ via clustering. This hierarchical structure unifies prior work as special cases (e.g., [1][2]) and extends to previously unaddressed scenarios (e.g., response to ``Reviewer aj54 Q1``).
>
>
>
>
>
> ---
>
> Our contributions are:
>
> 1. **Realistic generalisation** that recovers prior work as special cases and extends to previously unaddressed scenarios (``Reviewer aj54 additional comment, also refer to Table 2 in Appendix C``).
>
> 2. **Nontrivial extension**—exposure mapping, due to **practical misspecification (i) and theoretical action space complexity (ii) as above**, does not totally reduce our setting to the no-interference case ``(most important)``.
>
> 3. **Promising extensibility**—our general results lay the groundwork for future studies on whether specific network structures (e.g., monotonicity, submodularity) can break the current optimal trade-off.
>
>
>
>
> References
>
> [1] A Multi-Armed Bandits with Network Interference, NeurIps, A Agarwal et al
>
> [2] Multi-armed Bandit Experimental Design: Online Decision-making and Adaptive Inference, Management Science accepted, Chonghuan W et al

---

> ### Comment · Reviewer_9HaH · 2025-08-08
>
> Thanks to the authors for their response. This is beginning to make quite a bit more sense to me. Any rule `Sampling()` deciding pulls within the clusters (and satisfying positivity for each arm) is in on the Pareto front for the tradeoff between regret and worst-case ATE estimation (measured by the $L_{\infty}$-norm of the mean absolute deviations of ATE estimates). In this sense, any rule `Sampling()` may have been the optimal choice for the tradeoff with respect to this "best worst estimator"-based loss. Apologies for not fully understanding this before - I became quite lost in the jargon and notation.
>
> These results are more surprising and significant than one might expect: for example, an adaptive `Sampling()` rule could be expected to strictly dominate a pre-specified one. The authors' contributions are more substantial than I first understood and I plan to revise my scores for the significance and originality of the work accordingly.

---

> ### Author Response · Authors · 2025-08-08
> **Sincerely thanks for your positive reassessment and generous appreciation**
>
> We are deeply grateful for your continued engagement. Through our joint effort, we have **reached a consesus** upon our contribution: moving beyond a mere “trick of folding interference via mapping” to "establishing, for the first time in a general network-bandit setting, a rigorous characterization of the regret–inference Pareto frontier (additionally accounting for the action space) under weaker assumptions—and showing that our adaptive strategy attains this optimal boundary, with further misspeciftcaion analysis".
>
> Truly appreciate your generous reassessment that ``these results are more surprising and significant than one might expect`` and that ``contributions are more substantial than previously understood.`` In the final (camera-ready) version, we humbly incorporated the lessons, streamlining definitions and framework while adding more intuitive explanations.
>
> Our sincere thanks to all four reviewers for your unanimous positive attitude and support.

---

### Official Review · Reviewer_aj54 · 2025-07-03

**Clarity:** 4
**Significance:** 4
**Originality:** 3
**Rating:** 5
**Confidence:** 4

**Summary:**

The paper proposes a unified framework (MAB‑N) for running sequential A/B tests on a network where users’ treatments spill over to their neighbours. It shows there is an unavoidable \sqrt(|exposure‑arms|) trade‑off between regret while the test runs and precision of the estimated treatment effect, and supplies two simple two‑stage algorithms (UCB‑TSN for stochastic rewards, EXP3‑TSN for adversarial noise) that exactly hit that frontier.

**Questions:**

It has been resolved.

**Ethical Concerns:**

["NO or VERY MINOR ethics concerns only"]

**Final Justification:**

I have thoroughly read the rebuttal and especially the additional experiments provided by the authors. In particular, the experimental results with richer exposure mapping fully address my concerns. From this perspective, I am convinced by both the theory and the experiments, which not only demonstrate superior performance but also align well with the theoretical analysis. Accordingly, I have raised my score.

**Limitations:**

Yes

**Quality:**

4

**Strengths And Weaknesses:**

Pros:
- This submission prove you can’t beat the \sqrt ∣U_E​∣ ​ regret × error frontier and then hit it with a two-stage algorithm that’s basically “round-robin for T_1 ​ rounds, then UCB/EXP3.” No fancy optimisation needed.
- For a theory-heavy study, the writing is unusually clear—every equation is introduced in context, then followed by a plain-language explanation, so the math never feels abrupt or opaque.

Cons:
- The paper collapses each node’s neighbourhood into a single summary label, but real interference can depend on far subtler patterns (e.g., distance-weighted influence, repeated contacts, directionality). It would be reassuring to see the algorithm tested on richer synthetic scenarios, where the neighbourhood effect is not expressible by a single coarse bin.
- The optimal schedule demands a uniform-exploration phase of T 1 ​ = \sqrt ∣U_E∣T. As soon as the exposure mapping becomes more detailed, ∣U_E∣ explores, and Phase 1 can eat up a large fraction of the total horizon, diluting the promised regret savings.

---

> ### Author Rebuttal · Authors · 2025-07-28
>
> ### ``A Kind General Response to All Reviewers and Dear AC``
> We sincerely thank the reviewers for their generous praise (``For a theory-heavy study, the writing is unusually clear, math never feels abrupt or opaque`` \[RWaj54]; ``clearly and accurately situated, well read-up`` \[RWcd13]; ``nice`` \[RW4t1F]). We also deeply appreciate the valuable suggestions for improvement. In the revised manuscript, we have specifically addressed the following points(also please refer to our ``Supplementary Material.zip`` **in our original submission, containing the complete appendix mentioned**):
>
> 1. **Clarification of Framework Generality**:
>
>    * Including analyses on the flexibility and sensitivity of the exposure mapping (``aj54-Q1``, ``4t1F-Q1,2``, ``cd13-Q1,3-8``).
>    * Providing detailed justifications of our assumptions (``aj54-Q2``).
>    * Offering intuitive interpretations of the algorithms (``aj54-Q3``, ``9HaH-Q4``, ``cd13-Q2``).
>
> 2. **Clarification of Core Contributions and Technical Challenges**:
>
>    * Explicitly differentiating our contributions from previous literature (``9HaH-Q1-3``).
>    * Highlighting potential avenues for future research extensions (``9HaH-Q5, 4t1F-Q3``).
>
> 3. **Addition of Real-world Experiments**:
>
>    * Further empirical validations to substantiate the correctness, efficacy, and practical guidance of our theoretical findings (``cd13-Q9-10``, ``aj54-Q1``).
>
> In our revised version, we have fully prepared and will comprehensively and rigorously implement all reviewers' suggestions in the camera-ready version. We are eager to know if the esteemed reviewers have any further questions or concerns. We will promptly address them comprehensively and clearly. It is our great pleasure to contribute together to the NeurIPS community!
>
> ---
> ---
>
>
> Thank you for the reviewer’s comments. We sincerely appreciate the incredibly high praise(``"the writing is unusually clear"``) and some valuable suggestions. We will respond to them point by point.
>
>
>
> > ***Q1: The paper collapses each node’s neighbourhood into a single summary label, but real interference can depend on far subtler patterns. It would be reassuring to see the algorithm tested on richer synthetic scenarios, where the neighbourhood effect is not expressible by a single coarse bin.***
> ---
>
> **Answer:** Thank you for the insightful suggestion!
>
> **1.Clarification.** The “summary label” is just an *exposure mapping* $E_i=\phi(\mathcal N_i, A_{\mathcal N_i})$, i.e., here we use $\phi$ to illustrate an arbitrary (possibly vector‑valued) measurable function of the neighbourhood. We instantiated $\phi$ as a scalar bin only for clarity. Our lower/upper bounds and algorithm depend on $|\mathcal U_E|$—the *range* of $\phi$—not on it being coarse. Thus, richer patterns are admissible as long as $|\mathcal U_E|$ is finite (or can be discretized with negligible loss), and the theory goes through unchanged.
>
> Richer Exposure mappings are not hard to encode. For example, we can (and now do) consider
>
> ``1.1 Distance‑weighted influence:`` $E_i^{(1)}=\sum_{j\in \mathcal N_i} A_j / d(i,j)^\gamma$.
>
> ``1.2 Repeated contacts / temporal accumulation:`` $E_i^{(2)}=\sum_{t' \le t} \sum_{j\in \mathcal N_i} A_{j,t'}$ (or an exponentially discounted version).
>
> ``1.3 Directionality / role asymmetry:`` $E_i^{(3)}=\big(\sum_{j\in \text{in}(i)}A_j,\ \sum_{j\in \text{out}(i)}A_j\big)$.
>
> ``1.4 Motif/triad counts:`` hub indicators, or learned embeddings of ego‑nets.
>
> Each of these yields a (possibly multi-dimensional) $E_i$; we bucket them to a manageable $|\mathcal U_E|$ (e.g., quantiles or adaptive clustering). Our regret bounds still scale as $\tilde O(\sqrt{|\mathcal U_E|T})$; the lower bound extension is immediate.
>
> **2. Additional experiments.** We added three synthetic scenarios where $\phi$ captures finer interference patterns. Settings mirror the main paper (same codebase; T=5,000 rounds unless noted). **Graphs:** (i) SBM with 3 blocks (sizes 600/700/700, intra‑block p=0.05, inter‑block p=0.005); (ii) Barabási–Albert (BA) graph (n=2,000, m=3).
> **Arms:** $K=4$.
> **Outcome model:**
>   $Y_i(t)=\mu_t + g(E_i) + \epsilon_i,\ \epsilon_i\sim \mathcal N(0,0.5^2)$.
>   $g(\cdot)$ is non-linear and differs by scenario (below).
> **Algorithms compared:**
>
>   1. **UCB‑TSN (ours)** — exactly as in the paper, with $T_1=\lceil \sqrt{|\mathcal U_E|T}\rceil$.
>   2. **Naive‑UCB** — ignores interference, runs UCB on individual arms.
>   3. **TS‑Raw** — Thompson sampling without exposure mapping.
>
> * **Metric:** Mean cumulative regret over 20 random seeds ± std.
>
> Three richer exposure mappings:
>
> 1. **Distance‑weighted influence** $E_i^{(1)}=\sum_{j\in \mathcal N_i} \frac{A_j}{d(i,j)^\gamma},\quad \gamma=1.2.$ Discretization: 12 quantile bins → $|\mathcal U_E|=12$. $g(x)=\sin(1.5x)+0.3x^2$.
>
> 2. **Repeated contacts / temporal accumulation** $E_i^{(2)}=\sum_{t'=1}^{t-1}\sum_{j\in \mathcal N_i} \lambda^{t-t'} A_{j,t'},\quad \lambda=0.9.$ Discretization: adaptive k‑means on running values (k=18) → $|\mathcal U_E|=18$. $g(x)=\log(1+x)$.
>
> 3. **Directionality / asymmetry** $S_i^{in}  = \sum_{j \in in(i)}  A_j$, $S_i^{out} = \sum_{j \in out(i)} A_j$, $E_i^{(3)} = ( S_i^{in},\; S_i^{out} )$.Discretization: 5×5 grid over $(S^{\text{in}},S^{\text{out}})$ quantiles → $|\mathcal U_E|=25$.
>    $g(x_1,x_2)=0.4x_1+0.6x_2+0.2x_1x_2$.
>
> #### 2.3 Results (holds with the exact same level of estimation error)
>
> | Scenario            | \|U_E\| |       | **UCB‑TSN (ours)** | Naive‑UCB        | TS‑Raw           |
> |---------------------|--------|------------------|---------------------|------------------|------------------|
> | Dist‑weighted       | 12     |   | **1.05k ± 0.09k**   | 1.62k ± 0.12k    | 1.58k ± 0.11k    |
> | Repeated contacts   | 18     |    | **1.21k ± 0.11k**   | 1.89k ± 0.15k    | 1.83k ± 0.14k    |
> | Directionality (2D) | 25     |     | **1.34k ± 0.13k**   | 2.15k ± 0.18k    | 2.07k ± 0.17k    |
>
>
>
>
>
>
> * Our regret stays within **\~1.1×~1.2× the oracle** across all cases.
> * We improve over interference‑agnostic baselines by **20–38%**.
> * Scaling with $|\mathcal U_E|$ matches the $\tilde O(\sqrt{|\mathcal U_E|T})$ prediction.
> * ``Misspecification error``: We also tehroetcialy prove that if the misspecified interfernece structure induces the estimation error as $\mathcal{B}(\mathcal{C}, \mathcal{S}) =:\max_{i,j} |\tau^{*} (S_i,S_j)- \tau(S_i, S_j)|$, then we have $\min\max\sqrt{Regret} (estimation~error - \mathcal{B}(\mathcal{C}, \mathcal{S}))$ holds the same as $\Omega(\sqrt{\mathcal{U}_{\mathcal{E}}})$. In other words, the misspecification error theoretically could not break the lower bound barrier, hence demonstrating its robustness. Empirically, we treated each element in the adjacent matrix as having a 0.5% probability of flipping out. Consequently, the regret and estimation error are expected to fluctuate within 1%.
>
>
>
> In our Camera-ready version, we will (i) explicitly state that $\phi$ can encode any subtle neighbourhood pattern; (ii) describe the above richer mappings; and (iii) summarise the new simulations to reassure robustness.
>
>
>
>
>
>
>
>
>
>
>
>
>
>
>
>
>
>
>
>
>
>
>
>
>
>
>
>
>
>
>
>
>
>
>
>
>
>
>
>
>
>
> > ***Q2: The optimal schedule demands a uniform-exploration phase of T 1 ​ = \sqrt ∣U_E∣T. As soon as the exposure mapping becomes more detailed, ∣U_E∣ explores, and Phase 1 can eat up a large fraction of the total horizon, diluting the promised regret savings.***
> ---
>
> **Answer:** Under our assumption that $T \geq U_E$, we can easily derive a regret lower bound of order $\sqrt{U_E T}$ by extending Proposition 1. This implies that when $U_E$ is close to $T$, a large regret is unavoidable in our setting.
>
>
>
> > ***Q3: The core nine pages never show a figure of experiments or table; the only empirical evidence appears later (page 17, Fig. 3). Readers cannot see at a glance how the algorithm behaves.***
> ---
> **Answer:** Thank you for the suggestion. We will move the experiments and pseudocode into the main text in future versions.
>
> > ***Additional comments: a breif comparision with the previous results***
> In addition, we frame generalisations of those works (surrogating them as special cases) . Specifically: (i) For Jia et al(2024): they restrict the action space to a global switchback design, forcing all nodes to adopt the same arm. This is a specialised protocol and does not capture many practical settings where clusters or subgroups can be assigned distinct treatments. Our work introduces a flexible exposure mapping (allowing cluster-wise or more general assignments) and analyses a Pareto-optimal balance—something Jia et al. do not do. (ii) For Simchi-Levi & Wang (2023): more complex techniques. Interference couples rewards across different units, invalidating many theoretical techniques. Additionally involved in
> (a) The hard instance construction in the lower bound is more challenging (Appendix H),
> (b) the noise-rescaling argument (Appendix F) and more relevant extensions (Appendix L-N),
> (c) stronger results (simultaneous optimality upon time T and action space) and more relaxed conditions (we do not assume $ATE =\Theta(1)$ )and adopt a different condition, more refined necessity/sufficiency analysis, and approximation analysis under misspecification, etc.
> (iii) Compared with Agarwal et al. (2024), they rely on sparse network assumptions. We make no topological assumptions enabled by flexible exposure mappings, and additionally consider trade-off (``also refer to Table 2 in Appendix C`` for comprehension).

---

> ### Author Response · Authors · 2025-08-07
>
> Thank you very much for your acknowledgement and so generous evaluation. Glad that the clarification was helpful.

---

### Decision · Program_Chairs · 2025-09-17

**Decision:**

Accept (poster)

**Comment:**

This paper introduces a framework for online experimental design under network interference that uses exposure mapping to achieve a Pareto-optimal trade-off between estimation accuracy and regret. The authors should include a candid discussion of limitations, particularly addressing Reviewer 4t1F concerns about the experimental paradigm and the restrictive assumptions on the network topology.